# Dynamic Assortment Selection and Pricing with Censored Preference Feedback

**Jung-hun Kim**
Seoul National University
junghunkim@snu.ac.kr

**Min-hwan Oh**
Seoul National University
minoh@snu.ac.kr

## Abstract

In this study, we investigate the problem of dynamic multi-product selection and pricing by introducing a novel framework based on a *censored multinomial logit* (C-MNL) choice model. In this model, sellers present a set of products with prices, and buyers filter out products priced above their valuation, purchasing at most one product from the remaining options based on their preferences. The goal is to maximize seller revenue by dynamically adjusting product offerings and prices, while learning both product valuations and buyer preferences through purchase feedback. To achieve this, we propose a Lower Confidence Bound (LCB) pricing strategy. By combining this pricing strategy with either an Upper Confidence Bound (UCB) or Thompson Sampling (TS) product selection approach, our algorithms achieve regret bounds of $\widetilde{O}(d^{\frac{3}{2}}\sqrt{T/\kappa})$ and $\widetilde{O}(d^2\sqrt{T/\kappa})$, respectively. Finally, we demonstrate the performance of our methods through simulations.

## 1 Introduction

The rapid growth of online markets has underscored the critical importance of developing strategies for dynamic pricing to maximize revenue. In these markets, sellers have the flexibility to adjust the prices of products sequentially in response to buyer behavior. However, optimizing prices is not a trivial task. To effectively set prices, sellers must learn the underlying demand parameters, as buyers make purchasing decisions based on their preferences and willingness to pay, as modeled by demand functions (Bertsimas & Perakis, 2006; Cheung et al., 2017; den Boer & Zwart, 2015; Javanmard & Nazerzadeh, 2019; Cohen et al., 2020; Javanmard & Nazerzadeh, 2019; Luo et al., 2022; Fan et al., 2024; Shah et al., 2019; Xu & Wang, 2021; Choi et al., 2023). While the prior work has focused on dynamically adjusting prices for single products, real-world applications such as e-commerce, hotel reservations, and air travel often involve multiple products, further complicating the pricing strategy (Den Boer, 2014; Ferreira et al., 2018; Javanmard et al., 2020; Goyal & Perivier, 2021).

In practice, sellers must do more than just set prices—they also need to determine which products to offer. Buyers purcahse a product based on their preferences for available items, and this purchasing process is influenced by the price. Higher prices reduce the likelihood of a purchase, as buyers filter out products priced above their perceived value. This dynamic interplay between pricing and buyer preferences is a fundamental aspect of real-world online markets, making it essential to model both product selection and pricing together.

In this work, we tackle the problem of dynamic multi-product pricing and selection by developing a novel framework that captures the censored behavior of buyers—where buyers consider only those products priced below their valuation and purchase one product from the remaining options. To model this behavior, we extend the widely used multinomial logit (MNL) choice model (Agrawal et al., 2017a;b; Oh & Iyengar, 2021; 2019) to a censored MNL (C-MNL) model. This model allows us to capture buyer behavior more accurately in scenarios where product prices impact buyer choices. In our framework, sellers dynamically learn both the product valuations and buyer preferences, all while facing the challenge of not receiving feedback on which products buyers filtered out due to high prices, reflecting real-world conditions.

To address the inherent uncertainty in buyer behavior, we propose a novel Lower Confidence Bound (LCB) pricing strategy, which sets lower initial prices to encourage exploration and avoid price

censorship. In combination with Upper Confidence Bound (UCB) or Thompson Sampling (TS) strategies for product assortment selection, we provide algorithms that not only maximize revenue but also efficiently balance exploration and exploitation in the face of censored feedback. Through theoretical analysis, we derive regret bounds for our algorithms, and we validate their performance using synthetic datasets.

**Summary of Our Contributions.**

- We propose a novel framework for dynamic multi-product selection and pricing that incorporates a censored version of the multinomial logit (C-MNL) model. In this model, buyers filter out overpriced products and choose from the remaining options based on their preferences.
- We introduce a Lower Confidence Bound (LCB)-based pricing strategy to promote exploration by setting lower prices, avoiding buyer censorship, and facilitating the learning of buyer preferences and product valuations.
- We develop two algorithms that combine LCB pricing with Upper Confidence Bound (UCB) and Thompson Sampling (TS) for assortment selection, achieving regret bounds of $\widetilde{O}(d^{\frac{3}{2}}\sqrt{T/\kappa})$ and $\widetilde{O}(d^2\sqrt{T/\kappa})$, respectively.
- We provide extensive theoretical analysis, including regret bounds, and validate the effectiveness of our algorithms using synthetic datasets, demonstrating their superiority over existing approaches.

## 2 RELATED WORK

**Dynamic Pricing and Learning** Dynamic pricing with learning demand functions or market values has been widely studied (Bertsimas & Perakis, 2006; Cheung et al., 2017; den Boer & Zwart, 2015; Javanmard & Nazerzadeh, 2019; Cohen et al., 2020; Luo et al., 2022; Xu & Wang, 2021; Fan et al., 2024; Shah et al., 2019; Choi et al., 2023; Den Boer, 2014; Ferreira et al., 2018; Javanmard et al., 2020; Goyal & Perivier, 2021). However, previous work typically assumes that products are introduced arbitrarily or stochastically, meaning the products themselves are given rather than being part of the decision-making process. In contrast, our study incorporates a preference model for dynamic selection and pricing, where the agent must determine the assortment of products to offer with prices.

We note that Javanmard et al. (2020); Goyal & Perivier (2021); Erginbas et al. (2023) considered MNL structure for dynamic pricing, which was widely considered in the assortment bandits literature (Agrawal et al., 2017a;b; Oh & Iyengar, 2021; 2019). Based on the MNL structure, the previous pricing strategies have focused solely on optimizing revenue function. Notably, Javanmard et al. (2020); Perivier & Goyal (2022) examined scenarios where the assortment is predetermined rather than chosen by the agent under the dynamic pricing problems, and Erginbas et al. (2023) directly extended Goyal & Perivier (2021) by considering assortment selection under the same MNL structure. Moreover, Javanmard et al. (2020) consider i.i.d feature vectors fixed over time.

In our study, we utilize the MNL model with arbitrary features at each time to capture buyer preferences. Inspired by real-world scenarios, we further incorporate activation functions to address the non-continuous nature of buyer behavior, specifically their acceptable price thresholds. The presence of activation functions in our MNL model prevents a direct conversion to the standard MNL structure, distinguishing our work from that of Javanmard et al. (2020); Goyal & Perivier (2021); Erginbas et al. (2023). Furthermore, we address a multi-product setting where the agent not only prices but also selects products at each time. As a result, we must develop a novel strategy for both pricing and assortment selection to address this challenge.

Notably, while activation functions for buyer demand have been considered in Javanmard & Nazerzadeh (2019); Cohen et al. (2020); Luo et al. (2022); Xu & Wang (2021); Fan et al. (2024); Shah et al. (2019); Choi et al. (2023), these studies focused on single-product offered by the environment with single binary feedback at each time indicating whether the product was purchased or not. In contrast, we examine a multi-product setting where the agent must both select and price multiple products while receiving preference feedback, a scenario commonly observed in real-world online markets.

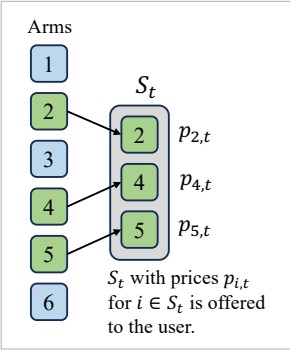 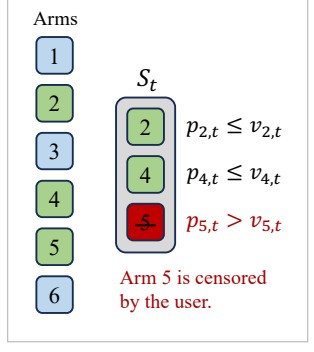 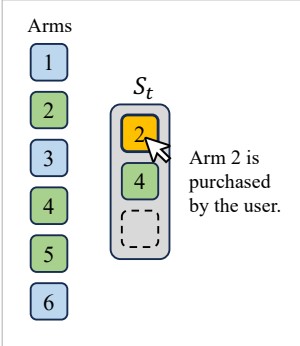

(a) The agent offer an assortment of arms $S_t$ with price $p_{i,t}$ for $i \in S_t$.

(b) Arms $i \in S_t$ satisfying $p_{i,t} > v_{i,t}$ are censored by the user.

(c) The user purchases an arm from the remaining arms in $S_t$ based on preference.

Figure 1: The illustration describes the process involved in making a purchase.

## 3 PROBLEM STATEMENT

There are $N$ arms (products) in the market. As illustrated in Figure 1, at each time $t \in [T]$, **(a)** an agent (seller) selects a set of arms $S_t \subseteq [N]$, referred to as 'assortment,' to a user (buyer) with a size constraint $|S_t| \leq K (\leq N)$. At the same time, the agent prices each arm $i \in S_t$ as $p_{i,t} \in \mathbb{R}_{\geq 0}$ and suggests the assortment with the corresponding prices to the user. **(b)** Then, based on the valuation $v_{i,t}$ and price $p_{i,t}$ for each arm $i \in S_t$, the user filters out any arms $i \in S_t$ where the price exceeds their valuation, i.e., $v_{i,t} < p_{i,t}$. **(c)** Finally, the user purchases at most one arm from the remaining options based on preference. In what follows, we describe our models for the user behavior and the revenue of the agent in more detail.

There are latent parameters $\theta_v$ and $\theta_\alpha \in \mathbb{R}^d$ (unknown to the agent) for valuation and price sensitivity, respectively. At each time $t$, each arm $i \in [N]$ has known feature information $x_{i,t}$ and $w_{i,t} \in \mathbb{R}^d$ for its valuation and price sensitivity, respectively. Then the (latent) valuation of each arm $i$ for the user is defined as $v_{i,t} := x_{i,t}^\top \theta_v \geq 0$. We also consider that there are (latent) price sensitivity parameters as $\alpha_{i,t} := w_{i,t}^\top \theta_\alpha \geq 0$. In this work, we propose a modification of the conventional MNL choice model with threshold-based activation functions, which we name as the *censored multinomial logit* (C-MNL) choice model.

**Definition 1 (Censored multinomial logit choice model)** *Let set of prices* $p_t := \{p_{i,t}\}_{i \in S_t}$. *Then, given $S_t$ and $p_t$, the user purchases an arm $i \in S_t$ by paying $p_{i,t}$ according to the probability defined as follows:*

$$\mathbb{P}_t(i|S_t, p_t) := \frac{\exp(v_{i,t} - \alpha_{i,t}p_{i,t})\mathbb{1}(p_{i,t} \leq v_{i,t})}{1 + \sum_{j \in S_t} \exp(v_{j,t} - \alpha_{j,t}p_{j,t})\mathbb{1}(p_{j,t} \leq v_{j,t})}. \quad (1)$$

From the activation function in the above definition, the user considers purchasing only the arms $i \in S_t$ satisfying that its price is lower than the user's valuation (or willingness to pay) as $p_{i,t} \leq v_{i,t}$. We also note that a higher price for an arm decreases the user's preference for it, while a higher valuation indicates a stronger preference. For notation simplicity, we use $\theta^* := [\theta_v; \theta_\alpha] \in \mathbb{R}^{2d}$ and $z_{i,t}(p) := [x_{i,t}; -pw_{i,t}] \in \mathbb{R}^{2d}$. Then the C-MNL of (1) can be represented as

$$\mathbb{P}_t(i|S_t, p_t) = \frac{\exp(x_{i,t}^\top \theta_v - w_{i,t}^\top \theta_\alpha p_{i,t})\mathbb{1}(p_{i,t} \leq x_{i,t}^\top \theta_v)}{1 + \sum_{j \in S_t} \exp(x_{j,t}^\top \theta_v - w_{j,t}^\top \theta_\alpha p_{j,t})\mathbb{1}(p_{j,t} \leq x_{j,t}^\top \theta_v)}$$

$$= \frac{\exp(z_{i,t}(p_{i,t})^\top \theta^*)\mathbb{1}(p_{i,t} \leq x_{i,t}^\top \theta_v)}{1 + \sum_{j \in S_t} \exp(z_{j,t}(p_{j,t})^\top \theta^*)\mathbb{1}(p_{j,t} \leq x_{j,t}^\top \theta_v)}.$$

As in the previous literature for MNL, it is allowed for each user to choose an outside option ($i_0$), or not to choose any, as $\mathbb{P}_t(i_0|S_t, p_t) = \frac{1}{1 + \sum_{j \in S_t} \exp(z_{j,t}(p_{j,t})^\top \theta^*)\mathbb{1}(p_{j,t} \leq x_{j,t}^\top \theta_v)}$. Importantly, at each

time $t$, the agent only observes feedback of chosen arm $i_t$ (at most one) but does *not* observe feedback on which arms are censored from the activation function based on the latent user's valuation. This makes it challenging to learn the valuation from the preference feedback, and the naive pricing strategies for maximizing revenue (Javanmard et al., 2020; Goyal & Perivier, 2021; Erginbas et al., 2023) do not work properly for our model.

The expected revenue from chosen arm $i \in S_t$ is represented as $R_{i,t}(S_t) = p_{i,t}\mathbb{P}_t(i|S_t, p_t)$. Then the overall expected revenue for the agent is formulated as

$$R_t(S_t, p_t) = \sum_{i \in S_t} R_{i,t}(S_t) = \sum_{i \in S_t} \frac{p_{i,t} \exp(z_{i,t}(p_{i,t})^\top \theta^*)\mathbb{1}(p_{i,t} \leq x_{i,t}^\top \theta_v)}{1 + \sum_{j \in S_t} \exp(z_{j,t}(p_{j,t})^\top \theta^*)\mathbb{1}(p_{j,t} \leq x_{j,t}^\top \theta_v)}.$$

For notation simplicity, we use $p = \{p_i\}_{i \in [N]}$. Then we define an oracle policy (with prior knowledge of $\theta^*$) regarding assortment and prices such that

$$(S_t^*, p_t^*) \in \underset{S \subseteq [N], p \in \mathbb{R}_{\geq 0}^N : |S| \leq K,}{\arg\max} R_t(S, p).$$

Then given $S_t$ and $p_t$ for all $t$ from a policy $\pi$, regret is defined as

$$R^\pi(T) = \sum_{t \in [T]} \mathbb{E}\left[R_t(S_t^*, p_t^*) - R_t(S_t, p_t)\right].$$

The goal of this problem is to find a policy $\pi$ that minimizes regret.

## 4 ALGORITHMS AND REGRET ANALYSES

### 4.1 UCB-BASED ASSORTMENT-SELECTION WITH LCB PRICING: UCBA-LCBP

Here we propose a UCB-based assortment-selection with LCB pricing algorithm (Algorithm 1) as follows. We denote by $P_{t,\theta}(i|S, p) := \frac{\exp(z_{i,t}(p_i)^\top \theta)}{1 + \sum_{j \in S} \exp(z_{j,t}(p_j)^\top \theta)}$ the choice probability without the activation functions. Let the negative log-likelihood $f_t(\theta) := -\sum_{i \in S_t \cup \{i_0\}} y_{i,t} \log P_{t,\theta}(i|S_t, p_t)$ where $y_{i,t} \in \{0,1\}$ is observed preference feedback (1 denotes a choice, and 0 otherwise) and define the gradient of the likelihood as

$$g_t(\theta) := \nabla_\theta f_t(\theta) = \sum_{i \in S_t} (P_{t,\theta}(i|S_t, p_t) - y_{i,t}) z_{i,t}(p_{i,t}). \tag{2}$$

We also define gram matrices from $\nabla_\theta^2 f(\theta)$ as follows:

$$G_t(\theta) := \sum_{i \in S_t} P_{t,\theta}(i|S_t, p_t) z_{i,t}(p_{i,t}) z_{i,t}(p_{i,t})^\top - \sum_{i,j \in S_t} P_{t,\theta}(i|S_t, p_t) P_{t,\theta}(j|S_t, p_t) z_{i,t}(p_{i,t}) z_{j,t}(p_{j,t})^\top,$$

$$G_{v,t}(\theta) := \sum_{i \in S_t} P_{t,\theta}(i|S_t, p_t) x_{i,t} x_{i,t}^\top - \sum_{i,j \in S_t} P_{t,\theta}(i|S_t, p_t) P_{t,\theta}(j|S_t, p_t) x_{i,t} x_{j,t}^\top. \tag{3}$$

Then we construct the estimator of $\widehat{\theta}_t \in \mathbb{R}^{2d}$ for $\theta^*$ from the online mirror descent with (2) and (3), as studied by Zhang & Sugiyama (2024); Lee & Oh (2024), within the range of $\Theta = \{\theta \in \mathbb{R}^{2d} : \|\theta^{1:d}\|_2 \leq 1 \text{ and } \|\theta^{d+1:2d}\|_2 \leq 1\}$, which is described in Line 5.

Now we explain the details regarding the strategy for the decision of price and assortment. For the price strategy, we construct the lower confidence bound (LCB) of the valuation of arms. Let $\beta_\tau = C_1\sqrt{d\tau}\log(T)\log(K)$ where $\tau$ is the number of estimator updates for price, $H_t = \lambda I_{2d} + \sum_{s=1}^{t-1} G_s(\widehat{\theta}_s)$, and $H_{v,t} = \lambda I_d + \sum_{s=1}^{t-1} G_{v,s}(\widehat{\theta}_s)$ for some constant $C_1 > 0$ and $\lambda > 0$. We use $\theta^{n:m}$ for representing a vector consisting of elements from index $n$ to $m$ in $\theta \in \mathbb{R}^{2d}$. Then we denote the estimator regarding valuation by $\widehat{\theta}_{v,t} := \widehat{\theta}_t^{1:d}$. Let $t_\tau$ be the time step when $\tau$-th update of the estimation for price occurs and we use $\widehat{\theta}_{v,(\tau)} := \widehat{\theta}_{v,t_\tau}$ for the pricing strategy. Then with a constant $C > 1$, for the time steps $t$ corresponding to the $\tau$-th update, we construct the lower confidence bound (LCB) of the valuation of arm $i \in [N]$ as

$$\underline{v}_{i,t} := x_{i,t}^\top \widehat{\theta}_{v,(\tau)} - \sqrt{C}\beta_\tau \|x_{i,t}\|_{H_{v,t}^{-1}}.$$

---

**Algorithm 1** UCB-based Assortment-selection with LCB Pricing (UCBA–LCBP)

**Input:** $\lambda, \eta, \beta_\tau, C > 1$

**Init:** $\tau \leftarrow 1, t_1 \leftarrow 1, \widehat{\theta}_{v,(1)} \leftarrow \mathbf{0}_d$

1: **for** $t = 1, \ldots, T$ **do**

2:    $\widetilde{H}_t \leftarrow \lambda I_{2d} + \sum_{s=1}^{t-2} G_s(\widehat{\theta}_s) + \eta G_{t-1}(\widehat{\theta}_{t-1})$ with (3)

3:    $H_t \leftarrow \lambda I_{2d} + \sum_{s=1}^{t-1} G_s(\widehat{\theta}_s)$ with (3)

4:    $H_{v,t} \leftarrow \lambda I_d + \sum_{s=1}^{t-1} G_{v,s}(\widehat{\theta}_s)$ with (3)

5:    $\widehat{\theta}_t \leftarrow \arg\min_{\theta \in \Theta} g_{t-1}(\widehat{\theta}_{t-1})^\top \theta + \frac{1}{2\eta}\|\theta - \widehat{\theta}_{t-1}\|^2_{\widetilde{H}_t^{-1}}$ with (2) ;        ▷ Estimation

6:    **if** $\det(H_t) > C \det(H_{t_\tau})$ **then**

7:      $\tau \leftarrow \tau + 1; t_\tau \leftarrow t$

8:      $\widehat{\theta}_{v,(\tau)} \leftarrow \widehat{\theta}_{v,t_\tau}(= \widehat{\theta}_{t_\tau}^{1:d})$

9:    **for** $i \in [N]$ **do**

10:      $\underline{v}_{i,t} \leftarrow x_{i,t}^\top \widehat{\theta}_{v,(\tau)} - \sqrt{C}\beta_\tau\|x_{i,t}\|_{H_{v,t}^{-1}}$ ;        ▷ LCB for valuation

11:      $p_{i,t} \leftarrow \underline{v}_{i,t}^+$ ;        ▷ **Price selection w/ LCB**

12:      $\overline{v}_{i,t} \leftarrow x_{i,t}^\top \widehat{\theta}_{v,t} + \beta_\tau\|x_{i,t}\|_{H_{v,t}^{-1}}$ ;        ▷ UCB for valuation

13:      $\overline{u}_{i,t} \leftarrow z_{i,t}(p_{i,t})^\top \widehat{\theta}_t + \beta_\tau\|z_{i,t}(p_{i,t})\|_{H_t^{-1}} + 2\sqrt{C}\beta_\tau\|x_{i,t}\|_{H_{v,t}^{-1}}$ ;        ▷ UCB for utility

14:    $S_t \in \arg\max_{S \subseteq [N]:|S|\leq L} \sum_{i\in S} \frac{\overline{v}_{i,t}\exp(\overline{u}_{i,t})}{1+\sum_{j\in S}\exp(\overline{u}_{j,t})}$;        ▷ **Assortment selection w/ UCB**

15:    Offer $S_t$ with prices $p_t = \{p_{i,t}\}_{i\in S_t}$

16:    Observe preference (purchase) feedback $y_{i,t} \in \{0,1\}$ for $i \in S_t$

---

We use notation $x^+ = \max\{x, 0\}$ for $x \in \mathbb{R}$. Then, for the LCB pricing strategy, we set the price of arm $i$ using its LCB as

$$p_{i,t} = \underline{v}_{i,t}^+.$$

Importantly, from this pricing strategy, the algorithm can effectively explore arms avoiding censorship because the arm having a small price is likely to be activated from the user's threshold in the C-MNL choice model. In the analysis, under the condition of a favorable event regarding the LCB, we can appropriately handle the preference feedback from C-MNL for estimating $\theta^*$ with $\widehat{\theta}_t$. However, the conditional analysis for estimation introduces regret with each update. To solve this issue, we periodically update the estimator $\widehat{\theta}_{v,(\tau)}$ for LCB with constant $C > 1$, as described in Line 6, without hurting regret (in order) from estimation error.

Next, for the assortment selection, we construct upper confidence bounds (UCB) for valuation $v_{i,t}$ and preference utility $u_{i,t}$ as $\overline{v}_{i,t}$ and $\overline{u}_{i,t}$, respectively. We construct UCB for the valuation as

$$\overline{v}_{i,t} := x_{i,t}^\top \widehat{\theta}_{v,t} + \beta_\tau\|x_{i,t}\|_{H_{v,t}^{-1}}.$$

Interestingly, when constructing $\overline{u}_{i,t}$ regarding utility $u_{i,t} = z_{i,t}(p_{i,t}^*)^\top\theta^*$, it is required to consider enhanced-exploration under the uncertainty regarding both $\widehat{\theta}_t$ and $p_{i,t}$ (in $z_{i,t}(p_{i,t})$). We construct

$$\overline{u}_{i,t} := z_{i,t}(p_{i,t})^\top\widehat{\theta}_t + \beta_\tau\|z_{i,t}(p_{i,t})\|_{H_t^{-1}} + 2\sqrt{C}\beta_\tau\|x_{i,t}\|_{H_{v,t}^{-1}},$$

where $\beta_\tau\|z_{i,t}(p_{i,t})\|_{H_t^{-1}}$ comes from uncertainty of $\widehat{\theta}_t$ and $2\sqrt{C}\beta_\tau\|x_{i,t}\|_{H_{v,t}^{-1}}$ comes from that of $p_{i,t}$ in $z_{i,t}(p_{i,t})$. Then, using the UCB indexes, the assortment is chosen from

$$S_t \in \arg\max_{S \subseteq [N]:|S|\leq K} \sum_{i\in S} \frac{\overline{v}_{i,t}\exp(\overline{u}_{i,t})}{1+\sum_{j\in S}\exp(\overline{u}_{j,t})}.$$

We set $\eta = \frac{1}{2}\log(K+1) + 3$ and $\lambda = \max\{84d\eta, 192\sqrt{2}\eta\}$ for the algorithm.

## 4.2 REGRET ANALYSIS OF ALGORITHM 1 (`UCBA-LCBP`)

Similar to previous work for logistic and MNL bandit (Oh & Iyengar, 2019; 2021; Lee & Oh, 2024; Goyal & Perivier, 2021; Erginbas et al., 2023; Faury et al., 2020; Abeille et al., 2021), we consider the following regularity condition and definition for regret analysis.

**Assumption 1** $\|\theta_v\|_2 \le 1$, $\|\theta_\alpha\|_2 \le 1$, $\|x_{i,t}\|_2 \le 1$, and $\|w_{i,t}\|_2 \le 1$ for all $i \in [N]$, $t \in [T]$

Recall $\Theta = \{\theta \in \mathbb{R}^{2d} : \|\theta^{1:d}\|_2 \le 1 \text{ and } \|\theta^{d+1:2d}\|_2 \le 1\}$. Then we define a problem-dependent quantity regarding non-linearlity of the MNL structure as follows.

$$\kappa := \inf_{t \in [T], \theta \in \Theta, i \in S \subseteq [N], p \in [0,1]^N} P_{t,\theta}(i|S,p) P_{t,\theta}(i_0|S,p).$$

We note that in the worst-case, $1/\kappa = O(K^2)$ from the definition of $P_{t,\theta}(\cdot|S,p)$ with Assumption 1. Then Algorithm 1 achieves the regret bound in the following.

**Theorem 1** *Under Assumption 1, the policy $\pi$ of Algorithm 1 achieves a regret bound of*

$$R^\pi(T) = \widetilde{O}\left(d^{\frac{3}{2}}\sqrt{T/\kappa} + \frac{d^3}{\kappa}\right).$$

**Proof** The full version of the proof is provided in Appendix A.2. Here we provide a proof sketch. We first define event $E_t = \{\|\widehat{\theta}_s - \theta^*\|_{H_s} \le \beta_{\tau_s}, \forall s \le t\}$ and $E_T$ holds with a high probability. In what follows, we assume that $E_t$ holds at each time $t$.

For notation simplicity, we use $v_{i,t} := x_{i,t}^\top \theta_v$, $u_{i,t} := z_{i,t}(p_{i,t}^*)^\top \theta^*$, and $u_{i,t}^p := z_{i,t}(p_{i,t})^\top \theta^*$. Then we can show that for all $i \in [N]$ and $t \in [T]$, we have

$$\underline{v}_{i,t}^+ \le v_{i,t} \le \overline{v}_{i,t} \text{ and } u_{i,t} \le \overline{u}_{i,t}. \tag{4}$$

For the regret analysis, we need to obtain a bound for

$$R_t(S_t^*, p_t^*) - R_t(S_t, p_t)$$
$$= \sum_{i \in S_t^*} \frac{p_{i,t}^* \exp(u_{i,t}) \mathbb{1}(p_{i,t}^* \le v_{i,t})}{1 + \sum_{j \in S_t^*} \exp(u_{j,t}) \mathbb{1}(p_{j,t}^* \le v_{j,t})} - \sum_{i \in S_t} \frac{p_{i,t} \exp(u_{i,t}^p) \mathbb{1}(p_{i,t} \le v_{i,t})}{1 + \sum_{j \in S_t} \exp(u_{j,t}^p) \mathbb{1}(p_{j,t} \le v_{j,t})}. \tag{5}$$

For the purpose of analysis, we define $\overline{u}_{i,t}' = z_{i,t}(p_{i,t})^\top \theta^* + 2\beta_{\tau_t}\|z_{i,t}(p_{i,t})\|_{H_t^{-1}} + 2\sqrt{C}\beta_{\tau_t}\|x_{i,t}\|_{H_{v,t}^{-1}}$ so that $\overline{u}_{i,t} \le \overline{u}_{i,t}'$. For the first term in (5), with (4) and the UCB-based assortment selection policy, we can show that

$$\sum_{i \in S_t^*} \frac{p_{i,t}^* \exp(u_{i,t}) \mathbb{1}(p_{i,t}^* \le v_{i,t})}{1 + \sum_{j \in S_t^*} \exp(u_{j,t}) \mathbb{1}(p_{j,t}^* \le v_{j,t})} \le \frac{\sum_{i \in S_t} \overline{v}_{i,t} \exp(\overline{u}_{i,t}')}{1 + \sum_{i \in S_t} \exp(\overline{u}_{i,t}')}. \tag{6}$$

For the second term in (5), with (4) and the LCB-based pricing, we have

$$\sum_{i \in S_t} \frac{p_{i,t} \exp(u_{i,t}^p) \mathbb{1}(p_{i,t} \le v_{i,t})}{1 + \sum_{j \in S_t} \exp(u_{j,t}^p) \mathbb{1}(p_{j,t} \le v_{j,t})} = \frac{\sum_{i \in S_t} \underline{v}_{i,t}^+ \exp(u_{i,t}^p)}{1 + \sum_{i \in S_t} \exp(u_{i,t}^p)}. \tag{7}$$

From (5), (6), and (7), we have

$$R_t(S_t^*, p_t^*) - R_t(S_t, p_t) \le \frac{\sum_{i \in S_t} \overline{v}_{i,t} \exp(\overline{u}_{i,t}')}{1 + \sum_{i \in S_t} \exp(\overline{u}_{i,t}')} - \frac{\sum_{i \in S_t} \underline{v}_{i,t}^+ \exp(u_{i,t}^p)}{1 + \sum_{i \in S_t} \exp(u_{i,t}^p)}$$
$$= \frac{\sum_{i \in S_t} \overline{v}_{i,t} \exp(\overline{u}_{i,t}')}{1 + \sum_{i \in S_t} \exp(\overline{u}_{i,t}')} - \frac{\sum_{i \in S_t} \underline{v}_{i,t}^+ \exp(\overline{u}_{i,t}')}{1 + \sum_{i \in S_t} \exp(\overline{u}_{i,t}')} + \frac{\sum_{i \in S_t} \underline{v}_{i,t}^+ \exp(\overline{u}_{i,t}')}{1 + \sum_{i \in S_t} \exp(\overline{u}_{i,t}')} - \frac{\sum_{i \in S_t} \underline{v}_{i,t}^+ \exp(u_{i,t}^p)}{1 + \sum_{i \in S_t} \exp(u_{i,t}^p)}. \tag{8}$$

Let $\tau_t$ be the value of $\tau$ at the time step $t$. We can show that $\mathbb{E}[\beta_{\tau_T}] = \widetilde{O}(d)$ and $\mathbb{E}[\beta_{\tau_T}^2] = \widetilde{O}(d^2)$. Then, for a bound of the first two terms in (8), with expectation bounds for $\beta_{\tau_T}$ and $\beta_{\tau_T}^2$ in the above and elliptical potential bounds, we show that

$$\sum_{t \in [T]} \mathbb{E}\left[\left(\frac{\sum_{i \in S_t} \overline{v}_{i,t} \exp(\overline{u}'_{i,t})}{1 + \sum_{i \in S_t} \exp(\overline{u}'_{i,t})} - \frac{\sum_{i \in S_t} v_{i,t}^+ \exp(\overline{u}'_{i,t})}{1 + \sum_{i \in S_t} \exp(\overline{u}'_{i,t})}\right) \mathbb{1}(E_t)\right]$$

$$= O\left(\sum_{t \in [T]} \mathbb{E}\left[\left(\beta_{\tau_t} \max_{i \in S_t} \|x_{i,t}\|_{H_{v,t}^{-1}} \mathbb{1}(E_t)\right)\right] = \widetilde{O}\left(d^{\frac{3}{2}}\sqrt{T/\kappa}\right). \tag{9}$$

Likewise, for the bound of the last two terms in (8), we can show that

$$\sum_{t \in [T]} \mathbb{E}\left[\left(\frac{\sum_{i \in S_t} v_{i,t}^+ \exp(\overline{u}_{i,t})}{1 + \sum_{i \in S_t} \exp(\overline{u}_{i,t})} - \frac{\sum_{i \in S_t} v_{i,t}^+ \exp(u_{i,t}^p)}{1 + \sum_{i \in S_t} \exp(u_{i,t}^p)}\right) \mathbb{1}(E_t)\right] = \widetilde{O}\left(d^{\frac{3}{2}}\sqrt{T} + \frac{d^3}{\kappa}\right), \tag{10}$$

which conclude the proof with (8), (9), and the fact that $E_T$ holds with a high probability. ∎

Under the C-MNL model, our algorithm can achieve the tight regret bound with respect to $T$ as those established in standard MNL bandits (Oh & Iyengar, 2021) and dynamic pricing under MNL with arbitrary features (Goyal & Perivier, 2021; Erginbas et al., 2023). The regret bounds of Goyal & Perivier (2021); Erginbas et al. (2023) for the MNL dynamic pricing problems include $1/\kappa$ in the leading term where, in their work, $\kappa$ was assumed to be a constant term. In the worst case where $1/\kappa = O(K^2)$, their regret bounds become $\widetilde{O}(K^2\sqrt{T})$. Our regret bound contains only $\sqrt{1/\kappa}$ in the leading term, allowing it to remain $\widetilde{O}(K\sqrt{T})$ for large enough $T$ in the worst case. Moreover, the previous works (Goyal & Perivier, 2021; Erginbas et al., 2023) assumed that $x_{i,t}^\top \theta_\alpha \geq L$ with a positive constant $L > 0$ and their regret bounds include $1/L^n$ for $n \geq 1$. This leads to trivial regret bounds in the worst case when $L$ is small, whereas our regret bound does not depend on $L$. Regarding the dimensionality, the analysis of our new censored MNL model is significantly more challenging and involved due to the presence of activation functions, which adds complexity. As a result, our regret bound scales with $d^{\frac{3}{2}}$. However, whether this dependency can be improved remains an open question.

We now discuss the algorithmic differences between our method and the one proposed in Goyal & Perivier (2021); Erginbas et al. (2023). In the prior work of Goyal & Perivier (2021); Erginbas et al. (2023), the price is determined by maximizing revenue at each time. However, in our C-MNL framework, we cannot estimate $\theta^*$ using the revenue-maximizing price due to the hidden nature of non-purchased feedback regarding whether it is due to stochastic preference or elimination by an activation function. To address this issue, we employ an LCB pricing strategy that enhances exploration across all arms by adhering to acceptable user prices. Since our pessimistic pricing strategy introduces a gap from the optimal price, we further incorporate an exploration-enhanced strategy for choosing assortments.

Additionally, our algorithm is computationally more efficient since it does not require solving an optimization problem for pricing decisions, which was necessary in the previous work. We also note that regarding the computational costs of assortment selection, which is common in all MNL bandit literature, the assortment optimization can be computed by solving an LP (Davis et al., 2013).

### 4.3 TS-BASED ASSORTMENT-SELECTION WITH LCB PRICING: `TSA-LCBP`

Here we propose a Thompson sampling (TS)-based assortment-selection with LCB pricing algorithm (Algorithm 2). As in Algorithm 1, we first estimate $\widehat{\theta}_t$ using the online mirror descent within the range of $\Theta = \{\theta \in \mathbb{R}^{2d} : \|\theta^{1:d}\|_2 \leq 1$ and $\|\theta^{d+1:2d}\|_2 \leq 1\}$. For determining price, we utilize the LCB pricing as $p_{i,t} = v_{i,t}^+$, where, recall, $v_{i,t} = x_{i,t}^\top \widehat{\theta}_{v,(\tau)} - \beta_\tau \|x_{i,t}\|_{H_{v,t}^{-1}}$ with $\beta_\tau = C_1 \sqrt{d\tau} \log(T) \log(K)$.

For choosing the assortment, we sample two different types of instances from Gaussian distributions; one is for valuation and the other is for preference utility, each of which is sampled for $M$ times as $\widetilde{\theta}_{v,t}^{(m)} \in \mathbb{R}^d$ and $\widetilde{\theta}_t^{(m)} \in \mathbb{R}^{2d}$ for $m \in [M]$, respectively. We set $M = \lceil 1 - \frac{\log(2N)}{\log(1 - 1/4\sqrt{e\pi})} \rceil$. Then we

---

**Algorithm 2** TS-based Assortment-selection with LCB Pricing (TSA-LCBP)

---

**Input:** $\lambda, \eta, M, \beta_\tau, C > 1$

**Init:** $\tau \leftarrow 1, t_1 \leftarrow 1, \widehat{\theta}_{v,(1)} \leftarrow \mathbf{0}_d$

**for** $t = 1, \ldots, T$ **do**

$\quad \widetilde{H}_t \leftarrow \lambda I_{2d} + \sum_{s=1}^{t-2} G_s(\widehat{\theta}_s) + \eta G_{t-1}(\widehat{\theta}_{t-1})$ with (3)

$\quad H_t \leftarrow \lambda I_{2d} + \sum_{s=1}^{t-1} G_s(\widehat{\theta}_s)$ with (3)

$\quad H_{v,t} \leftarrow \lambda I_d + \sum_{s=1}^{t-1} G_{v,s}(\widehat{\theta}_s)$ with (3)

$\quad \widehat{\theta}_t \leftarrow \arg\min_{\theta \in \Theta} g_t(\widehat{\theta}_{t-1})^\top \theta + \frac{1}{2\eta} \|\theta - \widehat{\theta}_{t-1}\|^2_{\widetilde{H}_t^{-1}}$ with (2) ; $\qquad\qquad$ ▷ Estimation

$\quad$ Sample $\{\widetilde{\theta}_{v,t}^{(m)}\}_{m \in [M]}$ independently from $\mathcal{N}(\widehat{\theta}_{v,t}(= \widehat{\theta}_t^{1:d}), \beta_\tau^2 H_{v,t}^{-1})$

$\quad$ Sample $\{\widetilde{\theta}_t^{(m)}\}_{m \in [M]}$ independently from $\mathcal{N}(\widehat{\theta}_t, 2\beta_\tau^2 H_t^{-1})$

$\quad$ **if** $\det(H_t) > C \det(H_{t_\tau})$ **then**

$\quad\quad \tau \leftarrow \tau + 1; t_\tau \leftarrow t$

$\quad\quad \widehat{\theta}_{v,(\tau)} \leftarrow \widehat{\theta}_{v,t_\tau}(= \widehat{\theta}_{t_\tau}^{1:d})$

$\quad$ **for** $i \in [N]$ **do**

$\quad\quad \underline{v}_{i,t} \leftarrow x_{i,t}^\top \widehat{\theta}_{v,(\tau)} - \sqrt{C}\beta_\tau \|x_{i,t}\|_{H_{v,t}^{-1}}$ ; $\qquad\qquad$ ▷ LCB for valuation

$\quad\quad p_{i,t} \leftarrow \underline{v}_{i,t}^+$ ; $\qquad\qquad\qquad\qquad\qquad$ ▷ **Price selection w/ LCB**

$\quad\quad \widetilde{v}_{i,t} \leftarrow \arg\max_{m \in [M]} x_{i,t}^\top \widetilde{\theta}_{v,t}^{(m)}$ ; $\qquad\qquad\qquad$ ▷ TS for valuation

$\quad\quad \widetilde{\eta}_{i,t} \leftarrow \widetilde{v}_{i,t} - x_{i,t}^\top \widehat{\theta}_{v,t}$

$\quad\quad \widetilde{u}_{i,t} \leftarrow \arg\max_{m \in [M]} z_{i,t}(p_{i,t})^\top \widetilde{\theta}_t^{(m)} + 8C\widetilde{\eta}_{i,t}$ ; $\qquad$ ▷ TS for utility

$\quad S_t \in \arg\max_{S \subseteq [N]:|S| \leq K} \sum_{i \in S} \frac{\widetilde{v}_{i,t} \exp(\widetilde{u}_{i,t})}{1 + \sum_{j \in S} \exp(\widetilde{u}_{j,t})}$ ; $\qquad$ ▷ **Assortment selection w/ TS**

$\quad$ Offer $S_t$ with prices $p_t = \{p_{i,t}\}_{i \in S_t}$

$\quad$ Observe preference (purchase) feedback $y_{i,t} \in \{0, 1\}$ for $i \in S_t$

---

construct TS indexes regarding the valuation and utility as

$$\widetilde{v}_{i,t} := \arg\max_{m \in [M]} x_{i,t}^\top \widetilde{\theta}_{v,t}^{(m)} \text{ and } \widetilde{u}_{i,t} := \arg\max_{m \in [M]} z_{i,t}(p_{i,t})^\top \widetilde{\theta}_t^{(m)} + 16\widetilde{\eta}_{i,t}, \text{ respectively,}$$

where $\widetilde{\eta}_{i,t} = \widetilde{v}_{i,t} - x_{i,t}^\top \widehat{\theta}_{v,t}$. For the utility of $\widetilde{u}_{i,t}$, we have to consider the uncertainty regarding $p_{i,t}$ as well as $\widehat{\theta}_t$, which leads to requiring an additional exploration term $\widetilde{\eta}_{i,t}$. Then the assortment is determined from

$$S_t \in \arg\max_{S \subseteq [N]:|S| \leq K} \sum_{i \in S} \frac{\widetilde{v}_{i,t} \exp(\widetilde{u}_{i,t})}{1 + \sum_{j \in S} \exp(\widetilde{u}_{j,t})}.$$

In the following, we provide a regret bound of the algorithm by setting $\eta = \frac{1}{2}\log(K+1) + 3$ and $\lambda = \max\{84d\eta, 192\sqrt{2}\eta\}$.

## 4.4 REGRET ANALYSIS OF ALGORITHM 2 (TSA-LCBP)

**Theorem 2** *Under Assumption 1, the policy $\pi$ of Algorithm 2 achieves a regret bound of*

$$R^\pi(T) = \widetilde{O}\left(d^2\sqrt{T/\kappa} + \frac{d^4}{\kappa}\right)$$

**Proof** The full version of the proof is provided in Appendix A.3. Here we provide some key components of the proof. We first define event $E_t = \{\|\widehat{\theta}_s - \theta^*\|_{H_s} \leq \beta_t, \forall s \leq t\}$ and $E_T$ holds with a high probability. Let $A_t^* = \{i \in S_t^* : p_{i,t}^* \leq v_{i,t}\}$ and, recall, $v_{i,t} = x_{i,t}^\top \theta_v$, $u_{i,t} = z_{i,t}(p_{i,t}^*)^\top \theta^*$, and $u_{i,t}^p = z_{i,t}(p_{i,t})^\top \theta^*$. Then under $E_t$, from the pricing and assortment selection strategies, we can show that

$$R_t(S_t^*, p_t^*) - R_t(S_t, p_t) \leq \frac{\sum_{i \in A_t^*} v_{i,t} \exp(u_{i,t})}{1 + \sum_{i \in A_t^*} \exp(u_{i,t})} - \frac{\sum_{i \in S_t} \underline{v}_{i,t}^+ \exp(u_{i,t}^p)}{1 + \sum_{i \in S_t} \exp(u_{i,t}^p)}. \tag{11}$$

We define event $\widetilde{E}_t^{(a)}$ such that for all $i \in [N]$, we have

$$|\widetilde{v}_{i,t} - x_{i,t}^\top \widehat{\theta}_{v,t}| \le \gamma_t \|x_{i,t}\|_{H_{v,t}^{-1}} \text{ and } |\widetilde{u}_{i,t} - z_{i,t}(p_{i,t})^\top \widehat{\theta}_t| \le 8C\gamma_t(\|z_{i,t}(p_{i,t})\|_{H_t^{-1}} + \|x_{i,t}\|_{H_{v,t}^{-1}}),$$

which is shown to hold with a high probability. We also define event $\widetilde{E}_t^{(b)}$ such that for all $i \in [N]$, we have $\widetilde{v}_{i,t} \ge v_{i,t}$ and $\widetilde{u}_{i,t} \ge u_{i,t}$, which is shown to holds at least a positive constant. Let $\widetilde{E}_t = \widetilde{E}_t^{(a)} \cap \widetilde{E}_t^{(b)}$. Then we can show that $\mathbb{P}(\widetilde{E}_t|\mathcal{F}_{t-1}, E_t) \ge 1/8\sqrt{e\pi}$ where $\mathcal{F}_{t-1}$ is the filtration containing information before $t$.

Let $\widetilde{z}_{i,t} = z_{i,t}(p_{i,t}) - \mathbb{E}_{j \sim P_{t,\widehat{\theta}_t}(\cdot|S_t, p_t)}[z_{i,t}(p_{i,t})]$ and $\widetilde{x}_{i,t} = x_{i,t} - \mathbb{E}_{j \sim P_{t,\widehat{\theta}_t}(\cdot|S_t, p_t)}[x_{i,t}]$ and $\gamma_t = \beta_{\tau_t}\sqrt{8d\log(Mt)}$ where $\tau_t$ is the value of $\tau$ at time $t$. For the ease of presentation, we use

$$L_t = \gamma_t^2(\max_{i \in S_t} \|z_{i,t}(p_{i,t})\|_{H_t^{-1}}^2 + \max_{i \in S_t} \|x_{i,t}\|_{H_{v,t}^{-1}}^2) + \gamma_t^2(\max_{i \in S_t} \|\widetilde{z}_{i,t}\|_{H_t^{-1}}^2 + \max_{i \in S_t} \|\widetilde{x}_{i,t}\|_{H_{v,t}^{-1}}^2)$$

$$+ \gamma_t \sum_{i \in S_t} P_{t,\widehat{\theta}_t}(i|S_t, p_t)(\|\widetilde{z}_{i,t}\|_{H_t^{-1}} + \|\widetilde{x}_{i,t}\|_{H_{v,t}^{-1}}) + \gamma_t \sum_{i \in S_t} \|x_{i,t}\|_{H_{v,t}^{-1}}).$$

With a constant lower bound for $\mathbb{P}(\widetilde{E}_t|\mathcal{F}_{t-1}, E_t)$ and elliptical potential bounds, by omitting some details, we can show that

$$\mathbb{E}\left[\mathbb{E}\left[\left(\frac{\sum_{i \in A_t^*} v_{i,t}\exp(u_{i,t})}{1 + \sum_{i \in A_t^*}\exp(u_{i,t})} - \frac{\sum_{i \in S_t} \underline{v}_{i,t}^+ \exp(u_{i,t}^p)}{1 + \sum_{i \in S_t}\exp(u_{i,t}^p)}\right)\mathbb{1}(E_t) \mid \mathcal{F}_{t-1}\right]\right]$$

$$= O\left(\mathbb{E}\left[\mathbb{E}\left[L_t \mid \mathcal{F}_{t-1}, \widetilde{E}_t, E_t\right]\mathbb{P}(E_t|\mathcal{F}_{t-1})\right]\right) = \widetilde{O}\left(d^2\sqrt{T/\kappa} + \frac{d^4}{\kappa}\right),$$

which concludes the proof with (11) and the fact that $E_T$ holds with a high probability. ∎

To the best of our knowledge, this is the first work to apply Thompson Sampling (TS) to dynamic pricing under MNL functions, whereas the previous related works focused on UCB method (Erginbas et al., 2023) (or did not consider assortment selection (Goyal & Perivier, 2021)). Additionally, prior work on TS for MNL bandits (Oh & Iyengar, 2019) includes $1/\kappa$ in the regret bound so that $\widetilde{O}((1/\kappa)\sqrt{T})$ and requires computationally intensive estimation with an $O(t)$ cost at each time step $t$. In contrast, by using online mirror descent updates, our TS algorithm reduces the $\kappa$ dependency in the main term of the regret bound with $\widetilde{O}(\sqrt{T/\kappa})$ for large enough $T$ and provides computationally efficient online updates with an $O(1)$ cost for estimation in MNL bandits. It is also worth noting that our TS regret bound has an additional $\sqrt{d}$ term compared to the UCB algorithm (Algorithm 1). This phenomenon of increased regret with respect to $d$, compared to that of UCB, is consistent with observations from previous TS-based bandit literature (Oh & Iyengar, 2019; Agrawal & Goyal, 2013; Abeille & Lazaric, 2017). Furthermore, achieving a regret bound for TS, even after UCB, is widely considered an established contribution in bandit literature (Agrawal & Goyal, 2013; Abeille & Lazaric, 2017; Kim & Oh, 2024).

## 5 EXPERIMENTS

Here, we present numerical results using synthetic datasets with varying numbers of products $N$.[1] For the experiments, we generate each element in $\theta_v$ and $\theta_\alpha$ from the uniform distribution $(0, 1)$ and normalize them. We also generate features in the same way. We set $K = 5$ and $d = 4$. Unfortunately, there is no algorithm that can be directly applied to our novel setting. Therefore, for the benchmarks, we utilize previous algorithms proposed for dynamic pricing under MNL model such as `DASP-MNL` proposed in Erginbas et al. (2023) and `ONM` (online newton method) in Goyal & Perivier (2021). We note that `ONM` works under a given assortment rather than selecting one, so we adjust the method by adopting the method for the assortment optimization in Erginbas et al. (2023). We also utilize the method of Explore-then-commit (`ETC`) (Lattimore & Szepesvári, 2020) as a benchmark, which conducts exploration over the first $T^{2/3}$ time steps and then exploits for the remainder of the time. In Figure 2, we can observe other benchmarks do not work properly in our setting and our algorithms outperform the benchmarks with sublinear regret.

---

[1]Source code: https://github.com/junghunkim7786/DynamicAssortmentSelectionPricing

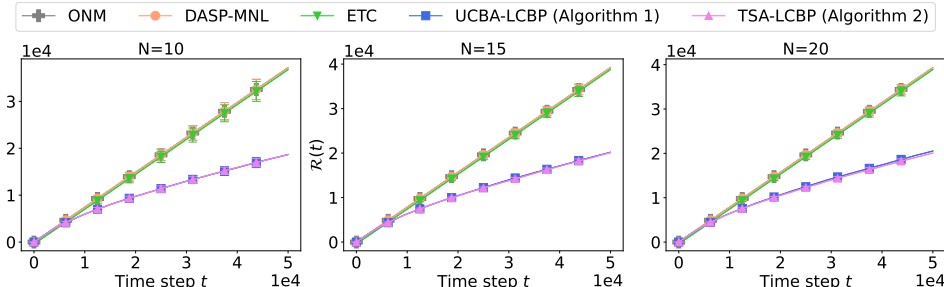

Figure 2: Experimental results for the regret of algorithms

## 6  RANDOMNESS IN ACTIVATION FUNCTION.

We further investigate the presence of randomness in the activation function in C-MNL. Let $\zeta_{i,t}$ be a zero-mean random noise drawn from the range of $[-c, c]$ for some $0 < c \leq 1$. we consider

$$\widetilde{\mathbb{P}}_t(i|S_t, p_t) = \frac{\exp(z_{i,t}(p_{i,t})^\top \theta^*)\mathbb{1}(p_{i,t} \leq (x_{i,t}^\top \theta_v + \zeta_{i,t})^+)}{1 + \sum_{j \in S_t} \exp(z_{j,t}(p_{j,t})^\top \theta^*)\mathbb{1}(p_{j,t} \leq (x_{j,t}^\top \theta_v + \zeta_{j,t})^+)}.$$

We propose a variant of Algorithm 1 (Algorithm 3 in Appendix A.4) using a robust LCB pricing strategy, which achieves $\widetilde{O}(d^{\frac{3}{2}}\sqrt{T/\kappa})$ when $c = O(1/\sqrt{T})$. Further details on the algorithm and theorem can be found in Appendix A.4.

## 7  CONCLUSION

In this study, we explore dynamic multi-product selection and pricing within a new framework of the censored multi-nomial logit choice model. We introduce algorithms that incorporate an LCB pricing strategy along with either a UCB or TS product selection strategy. These algorithms achieve regret bounds of $\widetilde{O}(d^{\frac{3}{2}}\sqrt{T/\kappa})$ and $\widetilde{O}(d^2\sqrt{T/\kappa})$, respectively. Lastly, we validate our algorithms through experiments with synthetic datasets.

**Reproducibility Statement.**  Complete proofs of the theorems are included in the appendix.

## 8  ACKNOWLEDGMENTS

The authors thank Joongkyu Lee for providing useful code. This work was supported by the Global-LAMP Program of the National Research Foundation of Korea (NRF) grant funded by the Ministry of Education (No. RS-2023-00301976) and also supported by the Korea government (MSIT) (No. RS-2022-NR071853 and RS-2023-00222663) and AI-Bio Research Grant through Seoul National University.

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

# A APPENDIX

## A.1 NOTATION TABLE FOR THE PROOFS

Table 1: We provide definitions of notations for the proofs.

$$v_{i,t} := x_{i,t}^\top \theta_v$$

$$\alpha_{i,t} := w_{i,t}^\top \theta_\alpha$$

$$\theta^* := [\theta_v; \theta_\alpha]$$

$$z_{i,t}(p) := [x_{i,t}; -pw_{i,t}]$$

$$\mathbb{P}_t(i|S_t, p_t) := \frac{\exp(v_{i,t} - \alpha_{i,t} p_{i,t})\mathbb{1}(p_{i,t} \le v_{i,t})}{1 + \sum_{j \in S_t} \exp(v_{j,t} - \alpha_{j,t} p_{j,t})\mathbb{1}(p_{j,t} \le v_{j,t})}$$
$$= \frac{\exp(x_{i,t}^\top \theta_v - w_{i,t}^\top \theta_\alpha p_{i,t})\mathbb{1}(p_{i,t} \le x_{i,t}^\top \theta_v)}{1 + \sum_{j \in S_t} \exp(x_{j,t}^\top \theta_v - w_{j,t}^\top \theta_\alpha p_{j,t})\mathbb{1}(p_{j,t} \le x_{j,t}^\top \theta_v)}$$
$$= \frac{\exp(z_{i,t}(p_{i,t})^\top \theta^*)\mathbb{1}(p_{i,t} \le x_{i,t}^\top \theta_v)}{1 + \sum_{j \in S_t} \exp(z_{j,t}(p_{j,t})^\top \theta^*)\mathbb{1}(p_{j,t} \le x_{j,t}^\top \theta_v)}$$

$$R_{i,t}(S_t) := p_{i,t}\mathbb{P}_t(i|S_t, p_t)$$

$$R_t(S_t, p_t) := \sum_{i \in S_t} R_{i,t}(S_t)$$

$$P_{t,\theta}(i|S, p) := \frac{\exp(z_{i,t}(p_i)^\top \theta)}{1 + \sum_{j \in S} \exp(z_{j,t}(p_j)^\top \theta)}$$

$$\widehat{\theta}_{v,t} := \widehat{\theta}_t^{1:d}$$

$$v_{i,t} := x_{i,t}^\top \theta_v$$

$$\overline{u}'_{i,t} := z_{i,t}(p_{i,t})^\top \theta^* + 2\beta_{\tau_t}\|z_{i,t}(p_{i,t})\|_{H_t^{-1}} + 2\sqrt{C}\beta_{\tau_t}\|x_{i,t}\|_{H_{v,t}^{-1}}$$

$$u_{i,t} := z_{i,t}(p_{i,t}^*)^\top \theta^*$$

$$x_{i,t}^o := [x_{i,t}; \mathbf{0}_d]$$

$$\widehat{u}_{i,t} := z_{i,t}(p_{i,t})^\top \widehat{\theta}_t$$

$$x_{i_0,t} := \mathbf{0}_d$$

$$z_{i_0,t} := \mathbf{0}_{2d}$$

$$Q(u) := \frac{\sum_{i \in S_t} v_{i,t}^+ \exp(u_i)}{1 + \sum_{i \in S_t} \exp(u_i)}$$

$$\widetilde{x}_{i,t} := x_{i,t} - \mathbb{E}_{j \sim P_{t,\widehat{\theta}_t}(\cdot|S_t, p_t)}[x_{j,t}]$$

$$\widetilde{z}_{i,t} := z_{i,t}(p_{i,t}) - \mathbb{E}_{j \sim P_{t,\widehat{\theta}_t}(\cdot|S_t, p_t)}[z_{j,t}(p_{j,t})]$$

$$\widetilde{G}_t(\widehat{\theta}_t) := \sum_{i \in S_t} P_{t,\widehat{\theta}_t}(i|S_t, p_t)z_{i,t}(p_{i,t})z_{i,t}(p_{i,t})^\top \mathbb{1}(E_t)$$
$$- \sum_{i \in S_t} \sum_{j \in S_t} P_{t,\widehat{\theta}_t}(i|S_t, p_t)P_{t,\widehat{\theta}_t}(j|S_t, p_t)z_{i,t}(p_{i,t})z_{j,t}(p_{j,t})^\top \mathbb{1}(E_t)$$

$$H_t' := \lambda I_{2d} + \sum_{s=1}^{t-1} \widetilde{G}_s(\widehat{\theta}_s)$$

$$\widetilde{u}'_{i,t} := z_{i,t}(p_{i,t})^\top \theta^* + 9C\gamma_t(\|z_{i,t}(p_{i,t})\|_{H_t^{-1}} + \|x_{i,t}\|_{H_{v,t}^{-1}})$$

## A.2 PROOF OF THEOREM 1

Let $\tau_t$ be the value of $\tau$ at time $t$ according to the update procedure in the algorithm. We first define event $E_t = \{\|\widehat{\theta}_s - \theta^*\|_{H_s} \le \beta_{\tau_s}, \forall s \le t\}$. Then we have $E_T \subset E_{T-1}, \dots, \subset E_1$ and $E_T$ holds with a high probability (to be shown). In what follows, we first assume that $E_t$ holds for each $t$. Under this event, we provide inequalities regarding the upper and lower bounds of valuation and utility function in the following. For notation simplicity, we use $v_{i,t} := x_{i,t}^\top \theta_v$, $u_{i,t} := z_{i,t}(p_{i,t}^*)^\top \theta^*$, and $x_{i,t}^o := [x_{i,t}; \mathbf{0}_d]$.

**Lemma 1** *For $t > 0$, under $E_t$, for all $i \in [N]$ we have*

$$\underline{v}_{i,t}^+ \le v_{i,t} \le \overline{v}_{i,t} \text{ and } u_{i,t} \le \overline{u}_{i,t}.$$

**Proof** For $t_\tau \leq t \leq t_{\tau+1} - 1$ for $\tau \geq 1$, under $E_t$, we have

$$
\begin{aligned}
|x_{i,t}^\top \theta_v - x_{i,t}^\top \widehat{\theta}_{v,(\tau)}| &= |{x_{i,t}^o}^\top \theta^* - {x_{i,t}^o}^\top \widehat{\theta}_{t_\tau}| \\
&\leq \|x_{i,t}^o\|_{H_t^{-1}} \|\theta^* - \widehat{\theta}_{t_\tau}\|_{H_t} \\
&\leq \|x_{i,t}^o\|_{H_t^{-1}} \sqrt{\frac{\det(H_t)}{\det(H_{t_\tau})}} \|\theta^* - \widehat{\theta}_{t_\tau}\|_{H_{t_\tau}} \\
&\leq \|x_{i,t}^o\|_{H_t^{-1}} \sqrt{C} \|\theta^* - \widehat{\theta}_{t_\tau}\|_{H_{t_\tau}} \\
&\leq \|x_{i,t}\|_{H_{v,t}^{-1}} \sqrt{C} \beta_{\tau_t},
\end{aligned}
$$

where the second inequality is obtained from Lemma 12 with the update procedure of $\widehat{\theta}_{v,(\tau)}$ in the algorithm. This implies $\underline{v}_{i,t} \leq v_{i,t}$. Then with $v_{i,t} \geq 0$, we have

$$
\underline{v}_{i,t}^+ \leq v_{i,t}.
$$

Under $E_t$, we also have

$$
|x_{i,t}^\top \theta_v - x_{i,t}^\top \widehat{\theta}_{v,t}| = |{x_{i,t}^o}^\top \theta^* - {x_{i,t}^o}^\top \widehat{\theta}_t| \leq \|x_{i,t}^o\|_{H_t^{-1}} \|\theta^* - \widehat{\theta}_t\|_{H_t} \leq \|x_{i,t}\|_{H_{v,t}^{-1}} \beta_{\tau_t},
$$

which implies

$$
v_{i,t} \leq \overline{v}_{i,t}.
$$

Now we provide the proof for the upper bound of $u_{i,t}$. Under $E_t$, we have

$$
\begin{aligned}
z_{i,t}(p_{i,t}^*)^\top \theta^* - z_{i,t}(p_{i,t})^\top \widehat{\theta}_t &= z_{i,t}(p_{i,t}^*)^\top \theta^* - z_{i,t}(p_{i,t})^\top \theta^* + z_{i,t}(p_{i,t})^\top \theta^* - z_{i,t}(p_{i,t})^\top \widehat{\theta}_t \\
&\leq z_{i,t}(p_{i,t}^*)^\top \theta^* - z_{i,t}(p_{i,t})^\top \theta^* + |z_{i,t}(p_{i,t})^\top \widehat{\theta}_t - z_{i,t}(p_{i,t})^\top \theta^*| \\
&\leq p_{i,t}^* w_{i,t}^\top \theta_\alpha - p_{i,t} w_{i,t}^\top \theta_\alpha + \|z_{i,t}(p_{i,t})\|_{H_t^{-1}} \|\widehat{\theta}_t - \theta^*\|_{H_t} \\
&\leq (p_{i,t}^* - p_{i,t}) w_{i,t}^\top \theta_\alpha + \beta_{\tau_t} \|z_{i,t}(p_{i,t})\|_{H_t^{-1}} \\
&\leq (v_{i,t} - \underline{v}_{i,t}^+) + \beta_{\tau_t} \|z_{i,t}(p_{i,t})\|_{H_t^{-1}} \\
&\leq (v_{i,t} - \underline{v}_{i,t}) + \beta_{\tau_t} \|z_{i,t}(p_{i,t})\|_{H_t^{-1}} \\
&\leq 2\sqrt{C} \beta_{\tau_t} \|x_{i,t}\|_{H_{v,t}^{-1}} + \beta_{\tau_t} \|z_{i,t}(p_{i,t})\|_{H_t^{-1}},
\end{aligned}
$$

where the third last inequality comes from $p_{i,t}^* \leq v_{i,t}$, $p_{i,t} = \underline{v}_{i,t}^+$, $v_{i,t} \geq \underline{v}_{i,t}^+$, and (positive sensitivity) $0 \leq w_{i,t}^\top \theta_\alpha \leq 1$. This concludes the proof.

∎

We have

$$
\begin{aligned}
&R_t(S_t^*, p_t^*) - R_t(S_t, p_t) \\
&= \sum_{i \in S_t^*} \frac{p_{i,t}^* \exp(z_{i,t}(p_{i,t}^*)^\top \theta^*) \mathbb{1}(p_{i,t}^* \leq x_{i,t}^\top \theta_v)}{1 + \sum_{j \in S_t^*} \exp(z_{j,t}(p_{j,t}^*)^\top \theta^*) \mathbb{1}(p_{j,t}^* \leq x_{j,t}^\top \theta_v)} \\
&\quad - \sum_{i \in S_t} \frac{p_{i,t} \exp(z_{i,t}(p_{i,t})^\top \theta^*) \mathbb{1}(p_{i,t} \leq x_{i,t}^\top \theta_v)}{1 + \sum_{j \in S_t} \exp(z_{j,t}(p_{j,t})^\top \theta^*) \mathbb{1}(p_{j,t} \leq x_{j,t}^\top \theta_v)}.
\end{aligned}
\tag{12}
$$

Let $\overline{u}_{i,t}' = z_{i,t}(p_{i,t})^\top \theta^* + 2\sqrt{C} \beta_{\tau_t} \|z_{i,t}(p_{i,t})\|_{H_t^{-1}} + 2\sqrt{C} \beta_{\tau_t} \|x_{i,t}\|_{H_{v,t}^{-1}}$. Then under $E_t$, we have $z_{i,t}(p_{i,t})^\top \widehat{\theta}_t - \beta_{\tau_t} \|z_{i,t}(p_{i,t})\|_{H_t^{-1}} \leq z_{i,t}(p_{i,t})^\top \theta^*$, which implies $\overline{u}_{i,t} \leq \overline{u}_{i,t}'$. In what follows, we provide lemmas for the bounds of each term in the above instantaneous regret. For notation simplicity, we use $u_{i,t}^p := z_{i,t}(p_{i,t})^\top \theta^*$.

**Lemma 2** *For $t > 0$, under $E_t$ we have*

$$
\sum_{i \in S_t^*} \frac{p_{i,t}^* \exp(z_{i,t}(p_{i,t}^*)^\top \theta^*) \mathbb{1}(p_{i,t}^* \leq x_{i,t}^\top \theta_v)}{1 + \sum_{j \in S_t^*} \exp(z_{j,t}(p_{j,t}^*)^\top \theta^*) \mathbb{1}(p_{j,t}^* \leq x_{j,t}^\top \theta_v)} \leq \frac{\sum_{i \in S_t} \overline{v}_{i,t} \exp(\overline{u}_{i,t}')}{1 + \sum_{i \in S_t} \exp(\overline{u}_{i,t}')}
$$

*and*

$$\sum_{i \in S_t} \frac{p_{i,t} \exp(z_{i,t}(p_{i,t})^\top \theta^*) \mathbb{1}(p_{i,t} \le x_{i,t}^\top \theta_v)}{1 + \sum_{j \in S_t} \exp(z_{j,t}(p_{j,t})^\top \theta^*) \mathbb{1}(p_{j,t} \le x_{j,t}^\top \theta_v)} = \frac{\sum_{i \in S_t} \underline{v}_{i,t}^+ \exp(u_{i,t}^p)}{1 + \sum_{i \in S_t} \exp(u_{i,t}^p)}.$$

**Proof** First, we provide a proof for the inequality in this lemma. We define $A_t^* = \{i \in S_t^* : p_{i,t}^* \le v_{i,t}\}$. We observe that $A_t^* = \arg\max_{S \subseteq [N]: |S| \le K} \frac{\sum_{i \in S} p_{i,t}^* \exp(u_{i,t})}{1 + \sum_{i \in S} \exp(u_{i,t})}$. Then, from Lemma A.3 in Agrawal et al. (2017a) and $u_{i,t} \le \overline{u}_{i,t}$ from Lemma 1, we can show that

$$\frac{\sum_{i \in A_t^*} p_{i,t}^* \exp(u_{i,t})}{1 + \sum_{i \in A_t^*} \exp(u_{i,t})} \le \frac{\sum_{i \in A_t^*} p_{i,t}^* \exp(\overline{u}_{i,t})}{1 + \sum_{i \in A_t^*} \exp(\overline{u}_{i,t})}. \tag{13}$$

From the above, under $E_t$, we have

$$R_t(S_t^*, p_t^*) = \frac{\sum_{i \in A_t^*} p_{i,t}^* \exp(u_{i,t})}{1 + \sum_{i \in A_t^*} \exp(u_{i,t})} \le \frac{\sum_{i \in A_t^*} p_{i,t}^* \exp(\overline{u}_{i,t})}{1 + \sum_{i \in A_t^*} \exp(\overline{u}_{i,t})} \le \frac{\sum_{i \in A_t^*} v_{i,t} \exp(\overline{u}_{i,t})}{1 + \sum_{i \in A_t^*} \exp(\overline{u}_{i,t})}$$

$$\le \frac{\sum_{i \in A_t^*} \overline{v}_{i,t} \exp(\overline{u}_{i,t})}{1 + \sum_{i \in A_t^*} \exp(\overline{u}_{i,t})} \le \frac{\sum_{i \in S_t} \overline{v}_{i,t} \exp(\overline{u}_{i,t})}{1 + \sum_{i \in S_t} \exp(\overline{u}_{i,t})}, \tag{14}$$

where the first inequality is obtained from (13), the second last inequality is obtained from $v_{i,t} \le \overline{v}_{i,t}$ from Lemma 1, and the last inequality is obtained from the policy $\pi$ of constructing $S_t$. Then from the definition of $S_t$, as in Lemma H.2 in Lee & Oh (2024), we can show that

$$\frac{\sum_{i \in S_t} \overline{v}_{i,t} \exp(\overline{u}_{i,t})}{1 + \sum_{i \in S_t} \exp(\overline{u}_{i,t})} \le \frac{\sum_{i \in S_t} \overline{v}_{i,t} \exp(\overline{u}_{i,t}')}{1 + \sum_{i \in S_t} \exp(\overline{u}_{i,t}')}. \tag{15}$$

Here we provide a proof for the equation in this lemma. Since $p_{i,t} = \underline{v}_{i,t}^+$ from the policy $\pi$ and $\underline{v}_{i,t}^+ \le v_{i,t}$ from Lemma 1, we have

$$R_t(S_t, p_t) = \frac{\sum_{i \in S_t} \underline{v}_{i,t}^+ \exp(u_{i,t}^p) \mathbb{1}(\underline{v}_{i,t}^+ \le v_{i,t})}{1 + \sum_{i \in S_t} \exp(u_{i,t}^p) \mathbb{1}(\underline{v}_{i,t}^+ \le v_{i,t})} = \frac{\sum_{i \in S_t} \underline{v}_{i,t}^+ \exp(u_{i,t}^p)}{1 + \sum_{i \in S_t} \exp(u_{i,t}^p)}, \tag{16}$$

which concludes the proof.

∎

From (12) and Lemma 2, under $E_t$, we have

$$R_t(S_t^*, p_t^*) - R_t(S_t, p_t)$$

$$= \sum_{i \in S_t^*} \frac{p_{i,t}^* \exp(z_{i,t}(p_{i,t})^\top \theta^*) \mathbb{1}(p_{i,t}^* \le x_{i,t}^\top \theta_v)}{1 + \sum_{j \in S_t^*} \exp(z_{j,t}(p_{j,t})^\top \theta^*) \mathbb{1}(p_{j,t}^* \le x_{j,t}^\top \theta_v)}$$

$$\quad - \sum_{i \in S_t} \frac{p_{i,t} \exp(z_{i,t}(p_{i,t})^\top \theta^*) \mathbb{1}(p_{i,t} \le x_{i,t}^\top \theta_v)}{1 + \sum_{j \in S_t} \exp(z_{j,t}(p_{j,t})^\top \theta^*) \mathbb{1}(p_{j,t} \le x_{j,t}^\top \theta_v)}$$

$$\le \frac{\sum_{i \in S_t} \overline{v}_{i,t} \exp(\overline{u}_{i,t}')}{1 + \sum_{i \in S_t} \exp(\overline{u}_{i,t}')} - \frac{\sum_{i \in S_t} \underline{v}_{i,t}^+ \exp(u_{i,t}^p)}{1 + \sum_{i \in S_t} \exp(u_{i,t}^p)}$$

$$= \frac{\sum_{i \in S_t} \overline{v}_{i,t} \exp(\overline{u}_{i,t}')}{1 + \sum_{i \in S_t} \exp(\overline{u}_{i,t}')} - \frac{\sum_{i \in S_t} \underline{v}_{i,t}^+ \exp(\overline{u}_{i,t}')}{1 + \sum_{i \in S_t} \exp(\overline{u}_{i,t}')} + \frac{\sum_{i \in S_t} \underline{v}_{i,t}^+ \exp(\overline{u}_{i,t}')}{1 + \sum_{i \in S_t} \exp(\overline{u}_{i,t}')} - \frac{\sum_{i \in S_t} \underline{v}_{i,t}^+ \exp(u_{i,t}^p)}{1 + \sum_{i \in S_t} \exp(u_{i,t}^p)}. \tag{17}$$

To obtain a bound for the above, we provide the following lemmas.

**Lemma 3** *For $t > 0$, under $E_t$ we have*

$$\frac{\sum_{i \in S_t} \overline{v}_{i,t} \exp(\overline{u}'_{i,t})}{1 + \sum_{i \in S_t} \exp(\overline{u}'_{i,t})} - \frac{\sum_{i \in S_t} \underline{v}^+_{i,t} \exp(\overline{u}'_{i,t})}{1 + \sum_{i \in S_t} \exp(\overline{u}'_{i,t})} = O\left(\beta_{\tau_t} \max_{i \in S_t} \|x_{i,t}\|_{H^{-1}_{v,t}}\right).$$

**Proof** For $\tau \geq 0$ and $t_\tau \leq t \leq t_{\tau+1} - 1$, under $E_t$, we have

$$
\begin{aligned}
\overline{v}_{i,t} - \underline{v}_{i,t} &= x_{i,t}^\top \widehat{\theta}_{v,t} - x_{i,t}^\top \widehat{\theta}_{v,(\tau_t)} + (\sqrt{C} + 1)\beta_{\tau_t} \|x_{i,t}\|_{H^{-1}_{v,t}} \\
&= x_{i,t}^\top \widehat{\theta}_{v,t} - x_{i,t}^\top \theta_v + x_{i,t}^\top \theta_v - x_{i,t}^\top \widehat{\theta}_{v,(\tau_t)} + (\sqrt{C} + 1)\beta_{\tau_t} \|x_{i,t}\|_{H^{-1}_{v,t}} \\
&= x_{i,t}^{o\top} \widehat{\theta}_t - x_{i,t}^{o\top} \theta^* + x_{i,t}^{o\top} \theta^* - x_{i,t}^{o\top} \widehat{\theta}_{t_\tau} + (\sqrt{C} + 1)\beta_{\tau_t} \|x_{i,t}\|_{H^{-1}_{v,t}} \\
&\leq \|\widehat{\theta}_t - \theta^*\|_{H_t} \|x_{i,t}^o\|_{H^{-1}_t} + \|\widehat{\theta}_{t_\tau} - \theta^*\|_{H_t} \|x_{i,t}^o\|_{H^{-1}_t} + (\sqrt{C} + 1)\beta_{\tau_t} \|x_{i,t}\|_{H^{-1}_{v,t}} \\
&\leq \beta_{\tau_t} \|x_{i,t}\|_{H^{-1}_{v,t}} + \sqrt{\frac{\det(H_t)}{\det(H_{t_\tau})}} \|\widehat{\theta}_{t_\tau} - \theta^*\|_{H_{t_\tau}} \|x_{i,t}^o\|_{H^{-1}_t} + (\sqrt{C} + 1)\beta_{\tau_t} \|x_{i,t}\|_{H^{-1}_{v,t}} \\
&\leq 2(\sqrt{C} + 1)\beta_{\tau_t} \|x_{i,t}\|_{H^{-1}_{v,t}},
\end{aligned}
$$

where the second inequality is obtained from Lemma 12.

Let $\widehat{u}_{i,t} = z_{i,t}(p_{i,t})^\top \widehat{\theta}_t$. Using the above inequality, under $E_t$, we have

$$
\begin{aligned}
\frac{\sum_{i \in S_t} \overline{v}_{i,t} \exp(\overline{u}'_{i,t})}{1 + \sum_{i \in S_t} \exp(\overline{u}'_{i,t})} - \frac{\sum_{i \in S_t} \underline{v}^+_{i,t} \exp(\overline{u}'_{i,t})}{1 + \sum_{i \in S_t} \exp(\overline{u}'_{i,t})} &= \frac{\sum_{i \in S_t} (\overline{v}_{i,t} - \underline{v}^+_{i,t}) \exp(\overline{u}'_{i,t})}{1 + \sum_{i \in S_t} \exp(\overline{u}'_{i,t})} \\
&\leq \frac{\sum_{i \in S_t} (\overline{v}_{i,t} - \underline{v}_{i,t}) \exp(\overline{u}'_{i,t})}{1 + \sum_{i \in S_t} \exp(\overline{u}'_{i,t})} \\
&= \frac{\sum_{i \in S_t} 2(\sqrt{C} + 1)\beta_{\tau_t} \|x_{i,t}\|_{H^{-1}_{v,t}} \exp(\overline{u}'_{i,t})}{1 + \sum_{i \in S_t} \exp(\overline{u}'_{i,t})} \\
&\leq 2(\sqrt{C} + 1)\beta_{\tau_t} \max_{i \in S_t} \|x_{i,t}\|_{H^{-1}_{v,t}}. \qquad (18)
\end{aligned}
$$

∎

Let $\widetilde{z}_{i,t} = z_{i,t}(p_{i,t}) - \mathbb{E}_{j \sim P_{t,\widehat{\theta}_t}(\cdot|S_t,p_t)}[z_{j,t}(p_{j,t})]$ and $\widetilde{x}_{i,t} = x_{i,t} - \mathbb{E}_{j \sim P_{t,\widehat{\theta}_t}(\cdot|S_t,p_t)}[x_{j,t}]$.

**Lemma 4** *For $t > 0$, under $E_t$ we have*

$$
\begin{aligned}
&\frac{\sum_{i \in S_t} \underline{v}^+_{i,t} \exp(\overline{u}'_{i,t})}{1 + \sum_{i \in S_t} \exp(\overline{u}'_{i,t})} - \frac{\sum_{i \in S_t} \underline{v}^+_{i,t} \exp(u^p_{i,t})}{1 + \sum_{i \in S_t} \exp(u^p_{i,t})} \\
&= O\left( \beta^2_{\tau_t} (\max_{i \in S_t} \|z_{i,t}(p_{i,t})\|^2_{H^{-1}_t} + \max_{i \in S_t} \|x_{i,t}\|^2_{H^{-1}_{v,t}}) + \beta^2_{\tau_t} (\max_{i \in S_t} \|\widetilde{z}_{i,t}\|^2_{H^{-1}_t} + \max_{i \in S_t} \|\widetilde{x}_{i,t}\|^2_{H^{-1}_{v,t}}) \right. \\
&\qquad \left. + \beta_{\tau_t} \sum_{i \in S_t} P_{t,\widehat{\theta}_t}(i|S_t, p_t)(\|\widetilde{z}_{i,t}\|_{H^{-1}_t} + \|\widetilde{x}_{i,t}\|_{H^{-1}_{v,t}}) \right).
\end{aligned}
$$

**Proof** *The proof is provided in Appendix A.5* ∎

In the below, we provide elliptical potential lemmas.

**Lemma 5**

$$\sum_{t=1}^{T} \max_{i \in S_t} \|z_{i,t}(p_{i,t})\|^2_{H_t^{-1}} \mathbb{1}(E_t) \leq (4d/\kappa) \log(1 + (2TK/d\lambda)),$$

$$\sum_{t=1}^{T} \max_{i \in S_t} \|\widetilde{z}_{i,t}\|^2_{H_t^{-1}} \mathbb{1}(E_t) \leq (4d/\kappa) \log(1 + (8TK/d\lambda)),$$

$$\sum_{t=1}^{T} \sum_{i \in S_t} P_{t,\widehat{\theta}_t}(i|S_t, p_t) \|\widetilde{z}_{i,t}\|^2_{H_t^{-1}} \mathbb{1}(E_t) \leq 4d \log(1 + (8TK/d\lambda)).$$

**Proof** Define

$$\widetilde{G}_t(\widehat{\theta}_t)$$
$$:= \sum_{i \in S_t} P_{t,\widehat{\theta}_t}(i|S_t, p_t) z_{i,t}(p_{i,t}) z_{i,t}(p_{i,t})^\top \mathbb{1}(E_t)$$
$$- \sum_{i \in S_t} \sum_{j \in S_t} P_{t,\widehat{\theta}_t}(i|S_t, p_t) P_{t,\widehat{\theta}_t}(j|S_t, p_t) z_{i,t}(p_{i,t}) z_{j,t}(p_{j,t})^\top \mathbb{1}(E_t). \tag{19}$$

Then we first have

$$\widetilde{G}_t(\widehat{\theta}_t)$$
$$= \sum_{i \in S_t} P_{t,\widehat{\theta}_t}(i|S_t, p_t) z_{i,t}(p_{i,t}) z_{i,t}(p_{i,t})^\top \mathbb{1}(E_t)$$
$$\quad - \sum_{i \in S_t} \sum_{j \in S_t} P_{t,\widehat{\theta}_t}(i|S_t, p_t) P_{t,\widehat{\theta}_t}(j|S_t, p_t) z_{i,t}(p_{i,t}) z_{j,t}(p_{j,t})^\top \mathbb{1}(E_t)$$
$$= \sum_{i \in S_t} P_{t,\widehat{\theta}_t}(i|S_t, p_t) z_{i,t}(p_{i,t}) z_{i,t}(p_{i,t})^\top \mathbb{1}(E_t)$$
$$\quad - \frac{1}{2} \sum_{i \in S_t} \sum_{j \in S_t} P_{t,\widehat{\theta}_t}(i|S_t, p_t) P_{t,\widehat{\theta}_t}(j|S_t, p_t) (z_{i,t}(p_{i,t}) z_{j,t}(p_{j,t})^\top + z_{j,t}(p_{j,t}) z_{i,t}(p_{i,t})^\top) \mathbb{1}(E_t)$$
$$\succeq \sum_{i \in S_t} P_{t,\widehat{\theta}_t}(i|S_t, p_t) z_{i,t}(p_{i,t}) z_{i,t}(p_{i,t})^\top \mathbb{1}(E_t)$$
$$\quad - \frac{1}{2} \sum_{i \in S_t} \sum_{j \in S_t} P_{t,\widehat{\theta}_t}(i|S_t, p_t) P_{t,\widehat{\theta}_t}(j|S_t, p_t) (z_{i,t}(p_{i,t}) z_{i,t}(p_{i,t})^\top + z_{j,t}(p_{j,t}) z_{j,t}(p_{j,t})^\top) \mathbb{1}(E_t)$$
$$= \sum_{i \in S_t} P_{t,\widehat{\theta}_t}(i|S_t, p_t) z_{i,t}(p_{i,t}) z_{i,t}(p_{i,t})^\top \mathbb{1}(E_t)$$
$$\quad - \sum_{i \in S_t} \sum_{j \in S_t} P_{t,\widehat{\theta}_t}(i|S_t, p_t) P_{t,\widehat{\theta}_t}(j|S_t, p_t) z_{i,t}(p_{i,t}) z_{i,t}(p_{i,t})^\top \mathbb{1}(E_t)$$
$$= \sum_{i \in S_t} P_{t,\widehat{\theta}_t}(i|S_t, p_t) \left(1 - \sum_{j \in S_t} P_{t,\widehat{\theta}_t}(j|S_t, p_t)\right) z_{i,t}(p_{i,t}) z_{i,t}(p_{i,t})^\top \mathbb{1}(E_t)$$
$$\succeq \sum_{i \in S_t} P_{t,\widehat{\theta}_t}(i|S_t, p_t) P_{t,\widehat{\theta}_t}(i_0|S_t, p_t) z_{i,t}(p_{i,t}) z_{i,t}(p_{i,t})^\top \mathbb{1}(E_t)$$
$$\succeq \sum_{i \in S_t} \kappa z_{i,t}(p_{i,t}) z_{i,t}(p_{i,t})^\top \mathbb{1}(E_t). \tag{20}$$

Define $H_t' := \lambda I_{2d} + \sum_{s=1}^{t-1} \widetilde{G}_s(\widehat{\theta}_s)$. Then we have

$$H_{t+1}' = H_t' + \widetilde{G}_t(\widehat{\theta}_t) \succeq H_t' + \sum_{i \in S_t} \kappa z_{i,t}(p_{i,t}) z_{i,t}(p_{i,t})^\top \mathbb{1}(E_t), \tag{21}$$

which implies that

$$
\begin{aligned}
\det(H'_{t+1}) &= \det(H'_t + \widetilde{G}_t(\widehat{\theta}_t)) \\
&\geq \det\Big(H'_t + \sum_{i \in S_t} \kappa z_{i,t}(p_{i,t}) z_{i,t}(p_{i,t})^\top \mathbb{1}(E_t)\Big) \\
&= \det(H'_t)\det\Big(I_{2d} + \sum_{i \in S_t} \kappa H'^{-1/2}_t z_{i,t}(p_{i,t})(H'^{-1/2}_t z_{i,t}(p_{i,t}))^\top \mathbb{1}(E_t)\Big) \\
&= \det(H'_t)(1 + \sum_{i \in S_t} \kappa \|z_{i,t}(p_{i,t})\|^2_{H'^{-1}_t} \mathbb{1}(E_t)) \\
&\geq \det(\lambda I_{2d}) \prod_{s=1}^{t} \Big(1 + \sum_{i \in S_s} \kappa \|z_{i,s}(p_{i,s})\|^2_{H'^{-1}_s} \mathbb{1}(E_s))\Big) \\
&\geq \lambda^{2d} \prod_{s=1}^{t} \Big(1 + \max_{i \in S_s} \kappa \|z_{i,s}(p_{i,s})\|^2_{H'^{-1}_s} \mathbb{1}(E_s))\Big) \\
&\geq \lambda^{2d} \prod_{s=1}^{t} \Big(1 + \max_{i \in S_s} \kappa \|z_{i,s}(p_{i,s})\|^2_{H'^{-1}_s} \mathbb{1}(E_s))\Big).
\end{aligned}
\tag{22}
$$

Since $p_{i,t} = \underline{v}^+_{i,t} \leq v_{i,t} \leq 1$ under $E_t$, we have $\|z_{i,t}(p_{i,t})\|^2_2 \leq (\|x_{i,t}\|_2 + \|w_{i,t}\|_2)^2 \leq 4$. Then under $E_t$, from the above inequality, $\lambda \geq 4$, and $0 < \kappa \leq 1$, using the fact that $x \leq 2\log(1+x)$ for any $x \in [0,1]$ and $\kappa \max_{i \in S_t} \|z_{i,t}(p_{i,t})\|^2_{H'^{-1}_t} \mathbb{1}(E_t) \leq \max_{i \in S_t} \|z_{i,t}(p_{i,t})\|^2_2 \mathbb{1}(E_t)/\lambda \leq 1$, we have

$$
\begin{aligned}
\sum_{t \in [T]} \kappa \max_{i \in S_t} \|z_{i,t}(p_{i,t})\|^2_{H'^{-1}_t} \mathbb{1}(E_t) &\leq 2 \sum_{t \in [T]} \log\Big(1 + \kappa \max_{i \in S_t} \|z_{i,t}(p_{i,t})\|^2_{H'^{-1}_t} \mathbb{1}(E_t)\Big) \\
&= 2 \log \prod_{t \in [T]} \Big(1 + \kappa \max_{i \in S_t} \|z_{i,t}(p_{i,t})\|^2_{H'^{-1}_t} \mathbb{1}(E_t)\Big) \\
&\leq 2 \log\Big(\frac{\det(H'_{t+1})}{\lambda^{2d}}\Big).
\end{aligned}
\tag{23}
$$

Using Lemma 13, $|S_t| \leq K$, $H'_t \preceq \lambda I_{2d} + \sum_{s=1}^{t-1} z_{i,s}(p_{i,s}) z_{i,s}(p_{i,s})^\top \mathbb{1}(E_t)$, $\|z_{i,t}(p_{i,t})\|_2 \leq 2$ under $E_t$, and $z_{i,t}(p_{i,t}) \in \mathbb{R}^{2d}$, we can show that

$$
\det(H'_{t+1}) \leq (\lambda + (2TK/d))^{2d}.
$$

Then from the above inequality, (23), and using the fact that $0 \prec H'_t \preceq H_t$ from $G_t(\theta) \succeq 0$, we can conclude

$$
\sum_{t=1}^{T} \max_{i \in S_t} \|z_{i,t}(p_{i,t})\|^2_{H^{-1}_t} \mathbb{1}(E_t) \leq \sum_{t=1}^{T} \max_{i \in S_t} \|z_{i,t}(p_{i,t})\|^2_{H'^{-1}_t} \mathbb{1}(E_t) \leq (4d/\kappa) \log(1 + (2TK/d\lambda)).
$$

Now we provide a proof for the second inequality of this lemma. Let $x_{i_0,t} = \mathbf{0}_d$ and $w_{i_0,t} = \mathbf{0}_d$ which implies $z_{i_0,t} = \mathbf{0}_{2d}$. Then we have

$$
\begin{aligned}
&\widetilde{G}_t(\widehat{\theta}_t) \\
&:= \sum_{i \in S_t} P_{t,\widehat{\theta}_t}(i|S_t, p_t) z_{i,t}(p_{i,t}) z_{i,t}(p_{i,t})^\top \mathbb{1}(E_t) \\
&\qquad - \sum_{i \in S_t} \sum_{j \in S_t} P_{t,\widehat{\theta}_t}(i|S_t, p_t) P_{t,\widehat{\theta}_t}(j|S_t, p_t) z_{i,t}(p_{i,t}) z_{j,t}(p_{j,t})^\top \mathbb{1}(E_t) \\
&= \sum_{i \in S_t} P_{t,\widehat{\theta}_t}(i|S_t, p_t) z_{i,t}(p_{i,t}) z_{i,t}(p_{i,t})^\top \mathbb{1}(E_t) \\
&\qquad - \sum_{i \in S_t \cup \{i_0\}} \sum_{j \in S_t \cup \{i_0\}} P_{t,\widehat{\theta}_t}(i|S_t, p_t) P_{t,\widehat{\theta}_t}(j|S_t, p_t) z_{i,t}(p_{i,t}) z_{j,t}(p_{j,t})^\top \mathbb{1}(E_t) \\
&= \mathbb{E}_{i \sim P_{t,\widehat{\theta}_t}(\cdot|S_t,p_t)}[z_{i,t}(p_{i,t}) z_{i,t}(p_{i,t})^\top] \mathbb{1}(E_t) - \mathbb{E}_{i \sim P_{t,\widehat{\theta}_t}(\cdot|S_t,p_t)}[z_{i,t}(p_{i,t})] \mathbb{E}_{i \sim P_{t,\widehat{\theta}_t}(\cdot|S_t,p_t)}[z_{i,t}(p_{i,t})]^\top \mathbb{1}(E_t) \\
&= \mathbb{E}_{i \sim P_{t,\widehat{\theta}_t}(\cdot|S_t,p_t)}[\widetilde{z}_{i,t} \widetilde{z}_{i,t}^\top] \mathbb{1}(E_t) \\
&\succeq \sum_{i \in S_t} P_{t,\widehat{\theta}_t}(i|S_t, p_t) \widetilde{z}_{i,t} \widetilde{z}_{i,t}^\top \mathbb{1}(E_t) \\
&\succeq \sum_{i \in S_t} \kappa \widetilde{z}_{i,t} \widetilde{z}_{i,t}^\top \mathbb{1}(E_t).
\end{aligned}
\tag{24}
$$

Define $H'_t := \lambda I_{2d} + \sum_{s=1}^{t-1} \widetilde{G}_s(\widehat{\theta}_s)$. Then by following the same proof steps of the first inequality of this lemma, we can show that

$$
\det(H'_{t+1}) \geq \lambda^{2d} \prod_{s=1}^{t} \left( 1 + \kappa \max_{i \in S_s} \|\widetilde{z}_{i,s}\|_{H'^{-1}_s} \mathbb{1}(E_s) \right)
\tag{25}
$$

Since, under $E_t$, we have $\|z_{i,t}(p_{i,t})\|_2 \leq \|x_{i,t}\|_2 + \|w_{i,t}\|_2 \leq 2$ implying that $\|\widetilde{z}_{i,t}\|_2^2 \leq 16$. Then, from the above inequality and $\lambda \geq 16$, using the fact that $x \leq 2\log(1+x)$ for any $x \in [0,1]$ and $\kappa \max_{i \in S_t} \|\widetilde{z}_{i,t}\|_{H'^{-1}_t}^2 \mathbb{1}(E_t) \leq \max_{i \in S_t} \|\widetilde{z}_{i,t}\|_2^2 \mathbb{1}(E_t)/\lambda \leq 1$, we have

$$
\begin{aligned}
\sum_{t \in [T]} \kappa \max_{i \in S_t} \|\widetilde{z}_{i,t}\|_{H'^{-1}_t}^2 \mathbb{1}(E_t) &\leq 2 \sum_{t \in [T]} \log \left( 1 + \kappa \max_{i \in S_t} \|\widetilde{z}_{i,t}\|_{H'^{-1}_t}^2 \mathbb{1}(E_t) \right) \\
&= 2 \log \prod_{t \in [T]} \left( 1 + \kappa \max_{i \in S_t} \|\widetilde{z}_{i,t}\|_{H'^{-1}_t}^2 \mathbb{1}(E_t) \right) \\
&\leq 2 \log \left( \frac{\det(H'_{t+1})}{\lambda^{2d}} \right).
\end{aligned}
\tag{26}
$$

Since we have $\det(H'_{t+1}) \leq (\lambda + (8TK/d))^{2d}$ and $0 \prec H'_t \preceq H_t$, from the above inequality and (26), we can conclude

$$
\sum_{t=1}^{T} \max_{i \in S_t} \|\widetilde{z}_{i,t}\|_{H_t^{-1}}^2 \mathbb{1}(E_t) \leq \sum_{t=1}^{T} \max_{i \in S_t} \|\widetilde{z}_{i,t}\|_{H'^{-1}_t}^2 \mathbb{1}(E_t) \leq (4d/\kappa) \log(1 + (8TK/d\lambda)).
$$

Now we provide a proof for the third inequality in this lemma. Then we have

$$
\begin{aligned}
\widetilde{G}_t(\widehat{\theta}_t) \\
:= & \sum_{i \in S_t} P_{t,\widehat{\theta}_t}(i|S_t, p_t) z_{i,t}(p_{i,t}) z_{i,t}(p_{i,t})^\top \mathbb{1}(E_t) \\
& - \sum_{i \in S_t} \sum_{j \in S_t} P_{t,\widehat{\theta}_t}(i|S_t, p_t) P_{t,\widehat{\theta}_t}(j|S_t, p_t) z_{i,t}(p_{i,t}) z_{j,t}(p_{j,t})^\top \mathbb{1}(E_t) \\
= & \sum_{i \in S_t} P_{t,\widehat{\theta}_t}(i|S_t, p_t) z_{i,t}(p_{i,t}) z_{i,t}(p_{i,t})^\top \mathbb{1}(E_t) \\
& - \sum_{i \in S_t \cup \{i_0\}} \sum_{j \in S_t \cup \{i_0\}} P_{t,\widehat{\theta}_t}(i|S_t, p_t) P_{t,\widehat{\theta}_t}(j|S_t, p_t) z_{i,t}(p_{i,t}) z_{j,t}(p_{j,t})^\top \mathbb{1}(E_t) \\
= & \mathbb{E}_{i \sim P_{t,\widehat{\theta}_t}(\cdot|S_t, p_t)}[z_{i,t}(p_{i,t}) z_{i,t}(p_{i,t})^\top] \mathbb{1}(E_t) \\
& - \mathbb{E}_{i \sim P_{t,\widehat{\theta}_t}(\cdot|S_t, p_t)}[z_{i,t}(p_{i,t})] \mathbb{E}_{i \sim P_{t,\widehat{\theta}_t}(\cdot|S_t, p_t)}[z_{i,t}(p_{i,t})]^\top \mathbb{1}(E_t) \\
= & \mathbb{E}_{i \sim P_{t,\widehat{\theta}_t}(\cdot|S_t, p_t)}[\widetilde{z}_{i,t} \widetilde{z}_{i,t}^\top] \mathbb{1}(E_t) \\
\succeq & \sum_{i \in S_t} P_{t,\widehat{\theta}_t}(i|S_t, p_t) \widetilde{z}_{i,t} \widetilde{z}_{i,t}^\top \mathbb{1}(E_t).
\end{aligned} \tag{27}
$$

Define $H'_t := \lambda I_{2d} + \sum_{s=1}^{t-1} \widetilde{G}_s(\widehat{\theta}_s)$. Then by following the same proof steps, we can show that

$$
\det(H'_{t+1}) \geq (2\lambda)^{2d} \prod_{s=1}^{t} \left( 1 + \sum_{i \in S_s} P_{s,\widehat{\theta}_s}(i|S_s, p_s) \|\widetilde{z}_{i,s}\|_{H_s'^{-1}} \mathbb{1}(E_s) \right) \tag{28}
$$

Since, under $E_t$, we have $\|z_{i,t}(p_{i,t})\|_2 \leq \|x_{i,t}\|_2 + \|w_{i,t}\|_2 \leq 2$ implying that $\|\widetilde{z}_{i,t}\|_2^2 \leq 16$. Then, from the above inequality and $\lambda \geq 16$, using the fact that $x \leq 2 \log(1 + x)$ for any $x \in [0, 1]$ and $\sum_{i \in S_t} P_{t,\widehat{\theta}_t}(i|S_t, p_t) \|\widetilde{z}_{i,t}\|_{H_t'^{-1}}^2 \mathbb{1}(E_t) \leq \max_{i \in S_t} \|\widetilde{z}_{i,t}\|_2^2 \mathbb{1}(E_t)/\lambda \leq 1$, we have

$$
\begin{aligned}
\sum_{t \in [T]} \sum_{i \in S_t} P_{t,\widehat{\theta}_t}(i|S_t, p_t) \|\widetilde{z}_{i,t}\|_{H_t'^{-1}}^2 \mathbb{1}(E_t) \leq & 2 \sum_{t \in [T]} \log \left( 1 + \sum_{i \in S_t} P_{t,\widehat{\theta}_t}(i|S_t, p_t) \|\widetilde{z}_{i,t}\|_{H_t'^{-1}}^2 \mathbb{1}(E_t) \right) \\
= & 2 \log \prod_{t \in [T]} \left( 1 + \sum_{i \in S_t} P_{t,\widehat{\theta}_t}(i|S_t, p_t) \|\widetilde{z}_{i,t}\|_{H_t'^{-1}}^2 \mathbb{1}(E_t) \right) \\
\leq & 2 \log \left( \frac{\det(H'_{t+1})}{\lambda^{2d}} \right).
\end{aligned} \tag{29}
$$

Since we have $\det(H'_{t+1}) \leq (\lambda + (8TK/d))^{2d}$ and $0 \prec H'_t \preceq H_t$, from the above inequality and (29), we can conclude

$$
\begin{aligned}
\sum_{t=1}^{T} \sum_{i \in S_t} P_{t,\widehat{\theta}_t}(i|S_t, p_t) \|\widetilde{z}_{i,t}\|_{H_t^{-1}}^2 \mathbb{1}(E_t) \leq & \sum_{t=1}^{T} \sum_{i \in S_t} P_{t,\widehat{\theta}_t}(i|S_t, p_t) \|\widetilde{z}_{i,t}\|_{H_t'^{-1}}^2 \mathbb{1}(E_t) \\
& \leq 4d \log(1 + (8TK/d\lambda)).
\end{aligned}
$$

$\blacksquare$

**Lemma 6**

$$\sum_{t=1}^{T} \max_{i \in S_t} \|x_{i,t}\|_{H_{v,t}^{-1}}^2 \leq (2d/\kappa) \log(1 + (TK/d\lambda)),$$

$$\sum_{t=1}^{T} \max_{i \in S_t} \|\widetilde{x}_{i,t}\|_{H_{v,t}^{-1}}^2 \leq (2d/\kappa) \log(1 + (4TK/d\lambda)),$$

$$\sum_{t=1}^{T} \max_{i \in S_t} P_{t,\widehat{\theta}_t}(i|S_t, p_t) \|\widetilde{x}_{i,t}\|_{H_{v,t}^{-1}} \leq 2d \log(1 + (4TK/d\lambda)).$$

**Proof** By following the proof steps in Lemma 5, we can prove the inequalities. ∎

Here we provide a lemma regarding the probability of the good event $E_t$. We define

$$\beta_1^2 := \eta(6\log(1 + (K+1)t) + 6)\left(\frac{17}{16}\lambda + 2\sqrt{\lambda}\log\left(2\sqrt{1+2t}T^2\right) + 16\left(\log(2\sqrt{1+2t}T^2)\right)^2\right) + 4\eta$$

$$+ 2\eta\sqrt{6}cd\log(1 + (t+1)/2\lambda) + 16\lambda$$

and for $\tau \geq 1$,

$$\beta_{\tau+1}^2 := \eta(6\log(1 + (K+1)t) + 6)\left(\frac{17}{16}\lambda + 2\sqrt{\lambda}\log\left(2\sqrt{1+2t}T^2\right) + 16\left(\log(2\sqrt{1+2t}T^2)\right)^2\right) + 4\eta$$

$$+ 2\eta\sqrt{6}cd\log(1 + (t+1)/2\lambda) + \beta_\tau^2.$$

**Lemma 7** *Let $c = 2\eta$, $\lambda \geq \max\{192\sqrt{2}\eta, 84d\eta\}$, and $\eta = \frac{1}{2}\log(K+1) + 3$. Then for $1 \leq t \leq t_2$, we have*

$$\mathbb{P}(E_t) \geq 1 - \frac{1}{T^2},$$

*and for $\tau \geq 2$ and $t_\tau + 1 \leq t \leq t_{\tau+1}$, we have*

$$\mathbb{P}(E_t|E_{t_\tau}) \geq 1 - \frac{1}{T^2}.$$

**Proof** The proof is provided in Appendix A.6 ∎

**Lemma 8**

$$\mathbb{P}(E_T) \geq 1 - \frac{2}{T}.$$

**Proof** Recall $E_t = \{\|\widehat{\theta}_s - \theta^*\|_{H_s} \leq \beta_s, \forall s \leq t\}$. For the time step $t_\tau + 1 \leq t \leq t_{\tau+1}$ for $\tau \geq 2$, since $E_1 \subseteq E_2, \ldots, \subseteq E_T$, from Lemma 7 we have $\mathbb{P}(E_t|E_{t_\tau}) = \mathbb{P}(E_t)/\mathbb{P}(E_{t_\tau}) \geq 1 - \frac{1}{T^2}$ implying $\mathbb{P}(E_t) \geq \left(1 - \frac{1}{T^2}\right)\mathbb{P}(E_{t_\tau})$. Likewise, we have $\mathbb{P}(E_{t_\tau}) \geq \left(1 - \frac{1}{T^2}\right)\mathbb{P}(E_{t_{\tau-1}})$. We also have $\mathbb{P}(E_t) \geq 1 - \frac{1}{T^2}$ for $1 \leq t \leq t_2$.

Therefore, from $\tau_T \leq T$, we can obtain

$$\mathbb{P}(E_T) \geq \left(1 - \frac{1}{T^2}\right)\mathbb{P}(E_{t_{\tau_T}}) \geq \left(1 - \frac{1}{T^2}\right)^{T-1}\mathbb{P}(E_{t_2}) \geq \left(1 - \frac{1}{T^2}\right)^T.$$

Let $X = \left(1 - \frac{1}{T^2}\right)^T$. By using the fact that $1 - \frac{1}{x} \leq \log(x) \leq x - 1$ for $x > 0$, we have

$$X - 1 \geq \log(X) = T\log\left(1 - \frac{1}{T^2}\right) \geq T\left(1 - \frac{1}{1 - \frac{1}{T^2}}\right) = \frac{-T}{T^2 - 1},$$

which conclude that $\mathbb{P}(E_T) \geq 1 - \frac{T}{T^2-1} \geq 1 - \frac{2}{T}$.

∎

Now we provide a bound for the total number of estimation updates, $\tau_T$. Using Lemma 13, under $E_T$, with $\|z_{i,t}(p_{i,t})\|_2 \leq 2$ and $z_{i,t}(p_{i,t}) \in \mathbb{R}^{2d}$, we can show that $\det(H_{T+1}) \leq (\lambda + (2TK/d))^{2d}$. Therefore, from the update procedure in the algorithm, $\tau_T$ satisfies $2^{\tau_T} \leq 2(\lambda + (TK/2d))^{2d}$, which implies $\tau_T = O(d \log(TK))$. Then we have

$$
\begin{aligned}
\mathbb{E}[\beta_{\tau_T}] &= \mathbb{E}[\beta_{\tau_T}|E_T]\mathbb{P}(E_T) + \mathbb{E}[\beta_{\tau_T}|E_T^c]\mathbb{P}(E_T^c) \\
&\leq C_1 d\sqrt{\log(KT)} \log(T) \log(K) + \mathbb{E}[\beta_{\tau_T}|E_T^c]\mathbb{P}(E_T^c) \\
&\leq C_1 d\sqrt{\log(KT)} \log(T) \log(K) + C_1\sqrt{dT}\log(T)\log(K)(2/T) \\
&= \widetilde{O}(d),
\end{aligned}
\tag{30}
$$

where the second inequality is obtained from $\mathbb{P}(E_T^c) \leq \frac{2}{T}$ and $\tau_T \leq T$. Likewise, we have

$$
\begin{aligned}
\mathbb{E}[\beta_{\tau_T}^2] &= \mathbb{E}[\beta_{\tau_T}^2|E_T]\mathbb{P}(E_T) + \mathbb{E}[\beta_{\tau_T}^2|E_T^c]\mathbb{P}(E_T^c) \\
&\leq C_1^2 d^2 \log(KT) \log(T)^2 \log(K)^2 + \mathbb{E}[\beta_{\tau_T}^2|E_T^c]\mathbb{P}(E_T^c) \\
&\leq C_1^2 d^2 \log(KT) \log(T)^2 \log(K)^2 + C_1^2 dT \log(T)^2 \log(K)^2(2/T) \\
&= \widetilde{O}(d^2),
\end{aligned}
\tag{31}
$$

Then from Lemmas 3, 4, 5, 8, and (17), (30), (31), using the fact that $E_1^c \subseteq E_2^c, \ldots, \subseteq E_T^c$, we obtain

$$
\begin{aligned}
R^\pi(T) &= \sum_{t \in [T]} \mathbb{E}[R_t(S_t^*, p_t^*) - R_t(S_t, p_t)] \\
&= \sum_{t \in [T]} \mathbb{E}[(R_t(S_t^*, p_t^*) - R_t(S_t, p_t))\mathbb{1}(E_t)] + \sum_{t \in [T]} \mathbb{E}[(R_t(S_t^*, p_t^*) - R_t(S_t, p_t))\mathbb{1}(E_t^c)] \\
&\leq \sum_{t \in [T]} \mathbb{E}[(R_t(S_t^*, p_t^*) - R_t(S_t, p_t))\mathbb{1}(E_t)] + \sum_{t \in [T]} \mathbb{P}(E_T^c) \\
&\leq \sum_{t \in [T]} \mathbb{E}\left[\left(\frac{\sum_{i \in S_t} \overline{v}_{i,t} \exp(\overline{u}_{i,t})}{1 + \sum_{i \in S_t} \exp(\overline{u}_{i,t})} - \frac{\sum_{i \in S_t} v_{i,t}^+ \exp(u_{i,t}^p)}{1 + \sum_{i \in S_t} \exp(u_{i,t}^p)}\right)\mathbb{1}(E_t)\right] + O(1) \\
&\leq \sum_{t \in [T]} \mathbb{E}\left[\left(\frac{\sum_{i \in S_t} \overline{v}_{i,t} \exp(\overline{u}_{i,t})}{1 + \sum_{i \in S_t} \exp(\overline{u}_{i,t})} - \frac{\sum_{i \in S_t} v_{i,t}^+ \exp(\overline{u}_{i,t})}{1 + \sum_{i \in S_t} \exp(\overline{u}_{i,t})}\right.\right. \\
&\qquad\qquad \left.\left. + \frac{\sum_{i \in S_t} v_{i,t}^+ \exp(\overline{u}_{i,t})}{1 + \sum_{i \in S_t} \exp(\overline{u}_{i,t})} - \frac{\sum_{i \in S_t} v_{i,t}^+ \exp(u_{i,t}^p)}{1 + \sum_{i \in S_t} \exp(u_{i,t}^p)}\right)\mathbb{1}(E_t)\right] + O(1)
\end{aligned}
$$

$$= O\left(\mathbb{E}\left[\beta_{\tau_T} \sum_{t\in[T]}\left(\max_{i\in S_t}\|x_{i,t}\|_{H_{v,t}^{-1}} + \sum_{i\in S_t} P_{t,\widehat{\theta}_t}(i|S_t,p_t)\left(\|\widetilde{x}_{i,t}\|_{H_{v,t}^{-1}} + \|\widetilde{z}_{i,t}\|_{H_t^{-1}}\right)\right)\mathbb{1}(E_t)\right]\right.$$

$$\left. + \mathbb{E}\left[\beta_{\tau_T}^2 \sum_{t\in[T]}\left(\max_{i\in S_t}\|x_{i,t}\|_{H_{v,t}^{-1}}^2 + \max_{i\in S_t}\|z_{i,t}(p_{i,t})\|_{H_t^{-1}}^2 + \max_{i\in S_t}\|\widetilde{x}_{i,t}\|_{H_{v,t}^{-1}}^2 + \max_{i\in S_t}\|\widetilde{z}_{i,t}\|_{H_t^{-1}}^2\right)\mathbb{1}(E_t)\right]\right)$$

$$= \widetilde{O}\left(\mathbb{E}\left[\beta_{\tau_T}\sqrt{T}\sqrt{\sum_{t\in[T]}\max_{i\in S_t}\|x_{i,t}\|_{H_{v,t}^{-1}}^2} + \beta_{\tau_T}\sqrt{\sum_{t\in[T]}\sum_{i\in S_t}P_{t,\widehat{\theta}_t}(i|S_t,p_t)}\times\right.\right.$$

$$\left.\left.\left(\sqrt{\sum_{t\in[T]}\sum_{i\in S_t}P_{t,\widehat{\theta}_t}(i|S_t,p_t)\|\widetilde{x}_{i,t}\|_{H_{v,t}^{-1}}^2} + \sqrt{\sum_{t\in[T]}\sum_{i\in S_t}P_{t,\widehat{\theta}_t}(i|S_t,p_t)\|\widetilde{z}_{i,t}\|_{H_t^{-1}}^2}\right)\mathbb{1}(E_t)\right] + \frac{d}{\kappa}\mathbb{E}[\beta_{\tau_T}^2]\right)$$

$$= \widetilde{O}\left(\mathbb{E}[\beta_{\tau_T}]\sqrt{dT/\kappa} + \frac{d^3}{\kappa}\right)$$

$$= \widetilde{O}\left(d^{3/2}\sqrt{\frac{T}{\kappa}} + \frac{d^3}{\kappa}\right).$$

### A.3 PROOF OF THEOREM 2

Let $\tau_t$ be the value of $\tau$ at time $t$ according to the update procedure in the algorithm. We first define event $E_t = \{\|\widehat{\theta}_s - \theta^*\|_{H_s} \leq \beta_{\tau_s}, \forall s \leq t\}$. Then we can observe $E_T \subset E_{T-1},\ldots,\subset E_1$ and $\mathbb{P}(E_T) \geq 1 - 1/T$ from Lemma 8. From Lemma 1, under $E_t$, we have

$$\underline{v}_{i,t}^+ \leq v_{i,t}. \tag{32}$$

We let $\gamma_t = \beta_{\tau_t}\sqrt{8d\log(Mt)}$ and filtration $\mathcal{F}_{t-1}$ be the $\sigma$-algebra generated by random variables before time $t$. In the following, we provide a lemma for error bounds of TS indexes.

**Lemma 9** *For any given $\mathcal{F}_{t-1}$, with probability at least $1 - \mathcal{O}(1/t^2)$, for all $i \in [N]$, we have*

$$|\widetilde{v}_{i,t} - x_{i,t}^\top\widehat{\theta}_{v,t}| \leq \gamma_t\|x_{i,t}\|_{H_{v,t}^{-1}} \text{ and } |\widetilde{u}_{i,t} - z_{i,t}(p_{i,t})^\top\widehat{\theta}_t| \leq 8C\gamma_t(\|z_{i,t}(p_{i,t})\|_{H_t^{-1}} + \|x_{i,t}\|_{H_{v,t}^{-1}}).$$

**Proof** We can show this lemma by adopting proof techniques of Lemma 10 in Oh & Iyengar (2019). We first provide a proof of the first inequality in this lemma. Given $\mathcal{F}_{t-1}$, Gaussian random variable $x_{i,t}^\top\widetilde{\theta}_{v,t}^{(m)}$ has mean $x_{i,t}^\top\widehat{\theta}_t$ and standard deviation $\beta_{\tau_t}\|x_{i,t}\|_{H_t^{-1}}$. Let $m' = \arg\max_{m\in M}x_{i,t}^\top\widetilde{\theta}_{v,t}^{(m)}$. Then we have

$$|\max_{m\in[M]}x_{i,t}^\top\widetilde{\theta}_{v,t}^{(m)} - x_{i,t}^\top\widehat{\theta}_t| = |x_{i,t}^\top(\widetilde{\theta}_{v,t}^{(m')} - \widehat{\theta}_t)|$$

$$= |x_{i,t}^\top H_{v,t}^{-1/2}H_{v,t}^{1/2}(\widetilde{\theta}_{v,t}^{(m')} - \widehat{\theta}_t)|$$

$$\leq \beta_{\tau_t}\|x_{i,t}\|_{H_{v,t}^{-1}}\|\beta_{\tau_t}^{-1}H_{v,t}^{1/2}(\widetilde{\theta}_{v,t}^{(m')} - \widehat{\theta}_t)\|_2$$

$$\leq \beta_{\tau_t}\|x_{i,t}\|_{H_{v,t}^{-1}}\max_{m\in[M]}\|\beta_{\tau_t}^{-1}H_{v,t}^{1/2}(\widetilde{\theta}_{v,t}^{(m)} - \widehat{\theta}_t)\|_2$$

$$= \beta_{\tau_t}\|x_{i,t}\|_{H_{v,t}^{-1}}\max_{m\in[M]}\|\xi_{v,m}\|_2,$$

where each element in $\xi_{v,m}$ is a standard normal random variable, which concludes the proof of the last inequality in this lemma from $\max_{m\in[M]}\|\xi_{v,m}\|_2 \leq \sqrt{4d\log(Mt)}$ with probability at least $1 - \frac{1}{t^2}$.

Now we provide a proof for the second inequality in this lemma. Let $m^* = \arg\max_{m\in[M]} x_{i,t}^\top \widetilde{\theta}_t^{(m)}$. Then we have

$$|\max_{m\in[M]} z_{i,t}(p_{i,t})^\top \widetilde{\theta}_t^{(m)} - z_{i,t}(p_{i,t})^\top \widehat{\theta}_t + 8C\widetilde{\eta}_{i,t}|$$

$$\leq |z_{i,t}(p_{i,t})^\top(\widetilde{\theta}_t^{(m^*)} - \widehat{\theta}_t)| + 8C|x_{i,t}^\top(\widetilde{\theta}_{v,t}^{(m')} - \widehat{\theta}_{v,t})|$$

$$= |z_{i,t}(p_{i,t})^\top H_t^{-1/2} H_t^{1/2}(\widetilde{\theta}_t^{(m^*)} - \widehat{\theta}_t)| + 8C|x_{i,t}^\top H_{v,t}^{-1/2} H_{v,t}^{1/2}(\widetilde{\theta}_{v,t}^{(m')} - \widehat{\theta}_{v,t})|$$

$$\leq \sqrt{2}\beta_{\tau_t}\|z_{i,t}(p_{i,t})\|_{H_t^{-1}}\|(\sqrt{2}\beta_{\tau_t})^{-1}H_t^{1/2}(\widetilde{\theta}_t^{(m^*)} - \widehat{\theta}_t)\|_2 + 8C\beta_{\tau_t}\|x_{i,t}\|_{H_{v,t}^{-1}}\|\beta_{\tau_t}^{-1}H_{v,t}^{1/2}(\widetilde{\theta}_{v,t}^{(m')} - \widehat{\theta}_{v,t})\|_2$$

$$\leq \sqrt{2}\beta_{\tau_t}\|z_{i,t}(p_{i,t})\|_{H_t^{-1}} \max_{m\in[M]}\|(\sqrt{2}\beta_{\tau_t})^{-1}H_t^{1/2}(\widetilde{\theta}_t^{(m)} - \widehat{\theta}_t)\|_2$$

$$+ 8C\beta_{\tau_t}\|x_{i,t}\|_{H_{v,t}^{-1}} \max_{m\in[M]}\|\beta_{\tau_t}^{-1}H_{v,t}^{1/2}(\widetilde{\theta}_{v,t}^{(m)} - \widehat{\theta}_{v,t})\|_2$$

$$= \sqrt{2}\beta_{\tau_t}\|z_{i,t}(p_{i,t})\|_{H_t^{-1}} \max_{m\in[M]}\|\xi_m\|_2 + 8C\beta_{\tau_t}\|x_{i,t}\|_{H_{v,t}^{-1}} \max_{m\in[M]}\|\xi_{v,m}\|_2,$$

where each element in $\xi_m$ and $\xi_{v,m}$ is a standard normal random variable. We use the fact that $\|\xi_m\|_2 \leq \sqrt{8d\log(t)}$ and $\|\xi_{v,m}\|_2 \leq \sqrt{4d\log(t)}$ with probability at least $1 - \frac{2}{t^2}$. By using union bound for all $m\in[M]$, with probability at least $1 - O(1/t^2)$, we have

$$\left|\max_{m\in[M]} z_{i,t}(p_{i,t})^\top \widetilde{\theta}_t^{(m)} - z_{i,t}(p_{i,t})^\top \widehat{\theta}_t\right| \leq \left(\sqrt{8d\log(Mt)}\right)\beta_{\tau_t}(\sqrt{2}\|z_{i,t}(p_{i,t})\|_{H_t^{-1}} + 8C\|x_{i,t}\|_{H_{v,t}^{-1}})$$

$$\leq 8C\gamma_t(\|z_{i,t}(p_{i,t})\|_{H_t^{-1}} + \|x_{i,t}\|_{H_{v,t}^{-1}}),$$

which concludes the proof.

∎

For notation simplicity, we use $u_{i,t}^p = z_{i,t}(p_{i,t})^\top \theta^*$. We define $A_t^* = \{i \in S_t^* : p_{i,t}^* \leq v_{i,t}\}$. As in (14) and (16), under $E_t$, we have

$$R_t(S_t^*, p_t^*) - R_t(S_t, p_t)$$

$$= \frac{\sum_{i\in A_t^*} p_{i,t}^* \exp(u_{i,t})}{1 + \sum_{i\in A_t^*} \exp(u_{i,t})} - \frac{\sum_{i\in S_t} v_{i,t}^+ \exp(u_{i,t}^p)\mathbb{1}(v_{i,t}^+ \leq v_{i,t})}{1 + \sum_{i\in S_t} \exp(u_{i,t}^p)\mathbb{1}(v_{i,t}^+ \leq v_{i,t})}$$

$$\leq \frac{\sum_{i\in A_t^*} v_{i,t} \exp(u_{i,t})}{1 + \sum_{i\in A_t^*} \exp(u_{i,t})} - \frac{\sum_{i\in S_t} v_{i,t}^+ \exp(u_{i,t}^p)\mathbb{1}(v_{i,t}^+ \leq v_{i,t})}{1 + \sum_{i\in S_t} \exp(u_{i,t}^p)\mathbb{1}(v_{i,t}^+ \leq v_{i,t})}$$

$$= \frac{\sum_{i\in A_t^*} v_{i,t} \exp(u_{i,t})}{1 + \sum_{i\in A_t^*} \exp(u_{i,t})} - \frac{\sum_{i\in S_t} v_{i,t}^+ \exp(u_{i,t}^p)}{1 + \sum_{i\in S_t} \exp(u_{i,t}^p)}. \tag{33}$$

In what follows, we provide several definitions of sets and events for the analysis of Thompson sampling. Regarding the valuation, we first define $\widetilde{v}_{i,t}(\Theta_v) = \max_{m\in[M]} x_{i,t}^\top \theta_v^{(m)}$ for $\Theta_v = \{\theta_v^{(m)} \in \mathbb{R}^d\}_{m\in[M]}$ and define sets

$$\widetilde{\Theta}_{v,t} = \left\{\Theta_v \in \mathbb{R}^{d\times M} : \left|\widetilde{v}_{i,t}(\Theta_v) - x_{i,t}^\top \widehat{\theta}_{v,t}\right| \leq \gamma_t\|x_{i,t}\|_{H_{v,t}^{-1}} \forall i \in [N]\right\} \text{ and}$$

$$\widetilde{\Theta}'_{v,t} = \left\{\Theta_v \in \mathbb{R}^{d\times M} : \widetilde{v}_{i,t}(\Theta) \geq v_{i,t} \forall i \in [N]\right\} \cap \widetilde{\Theta}_t.$$

Then we define event $\widetilde{E}_{v,t} = \{\{\widetilde{\theta}_{v,t}^{(m)}\}_{m\in[M]} \in \widetilde{\Theta}'_{v,t}\}$.

Regarding the utility, we define $\widetilde{u}_{i,t}(\Theta_u, \Theta_v) = \max_{m\in[M]} z_{i,t}(p_{i,t})^\top \theta^{(m)} + \max_{m\in[M]} z_{i,t}(p_{i,t})^\top(\theta_v^{(m)} - \widehat{\theta}_{v,t})$ for $\Theta_u = \{\theta^{(m)} \in \mathbb{R}^{2d}\}_{m\in[M]}$ and $\Theta_v = \{\theta_v^{(m)} \in \mathbb{R}^d\}_{m\in[M]}$,

and define sets

$$
\widetilde{\Theta}_t = \left\{ \Theta_u \times \Theta_v \in \mathbb{R}^{2d \times M} \times \mathbb{R}^{d \times M} : \left| \widetilde{u}_{i,t}(\Theta_u, \Theta_v) - z_{i,t}(p_{i,t})^\top \widehat{\theta}_t \right| \right.
$$

$$
\left. \leq 8C\gamma_t(\|z_{i,t}(p_{i,t})\|_{H_t^{-1}} + \|x_{i,t}\|_{H_{v,t}^{-1}}) \; \forall i \in [N] \right\}
$$

$$
\text{and } \widetilde{\Theta}'_t = \left\{ \Theta_u \times \Theta_v \in \mathbb{R}^{2d \times M} \times \mathbb{R}^{d \times M} : \widetilde{u}_{i,t}(\Theta_u, \Theta_v) \geq u_{i,t} \; \forall i \in [N] \right\} \cap \widetilde{\Theta}_t
$$

Then we define event $\widetilde{E}_{u,t} = \{\{\widetilde{\theta}_t^{(m)}\}_{m \in [M]} \times \{\widetilde{\theta}_{v,t}^{(m)}\}_{m \in [M]} \in \widetilde{\Theta}'_t\}$. For the ease of presentation, we define $\widetilde{E}_t = \widetilde{E}_{v,t} \cap \widetilde{E}_{u,t}$. In the following, we provide a lemma that will be used for following regret analysis. Let $\widetilde{z}_{i,t} = z_{i,t}(p_{i,t}) - \mathbb{E}_{j \sim P_{t,\widehat{\theta}_t}(\cdot|S_t,p_t)}[z_{i,t}(p_{i,t})]$ and $\widetilde{x}_{i,t} = x_{i,t} - \mathbb{E}_{j \sim P_{t,\widehat{\theta}_t}(\cdot|S_t,p_t)}[x_{i,t}]$.

**Lemma 10** *For $t \in [T]$, under $\widetilde{E}_{u,t}$ and $E_t$, we have*

$$
\sup_{\Theta_u \times \Theta_v \in \widetilde{\Theta}_t} \left( \frac{\sum_{i \in S_t} \widetilde{v}_{i,t} \exp(\widetilde{u}_{i,t})}{1 + \sum_{i \in S_t} \exp(\widetilde{u}_{i,t})} - \frac{\sum_{i \in S_t} \underline{v}_{i,t}^+ \exp(\widetilde{u}_{i,t}(\Theta_u, \Theta_v))}{1 + \sum_{i \in S_t} \exp(\widetilde{u}_{i,t}(\Theta_u, \Theta_v))} \right)
$$

$$
= O\left( \gamma_t^2 (\max_{i \in S_t} \|z_{i,t}(p_{i,t})\|_{H_t^{-1}}^2 + \max_{i \in S_t} \|x_{i,t}\|_{H_{v,t}^{-1}}^2) + \gamma_t^2 (\max_{i \in S_t} \|\widetilde{z}_{i,t}\|_{H_t^{-1}}^2 + \max_{i \in S_t} \|\widetilde{x}_{i,t}\|_{H_{v,t}^{-1}}^2) \right.
$$

$$
\left. + \gamma_t \sum_{i \in S_t} P_{t,\widehat{\theta}_t}(i|S_t,p_t)(\|\widetilde{z}_{i,t}\|_{H_t^{-1}} + \|\widetilde{x}_{i,t}\|_{H_{v,t}^{-1}}) + \gamma_t \max_{i \in S_t} \|x_{i,t}\|_{H_{v,t}^{-1}} \right).
$$

**Proof** We define $\widetilde{u}'_{i,t} = z_{i,t}(p_{i,t})^\top \theta^* + 9C\gamma_t(\|z_{i,t}(p_{i,t})\|_{H_t^{-1}} + \|x_{i,t}\|_{H_{v,t}^{-1}})$. Then from $\widetilde{E}_{u,t}$ and $E_t$, we have

$$
\begin{aligned}
\widetilde{u}_{i,t} &\leq z_{i,t}(p_{i,t})^\top \widehat{\theta}_t + 8C\gamma_t(\|z_{i,t}(p_{i,t})\|_{H_t^{-1}} + \|x_{i,t}\|_{H_{v,t}^{-1}}) \\
&\leq z_{i,t}(p_{i,t})^\top \theta^* + \beta_{\tau_t} \|z_{i,t}(p_{i,t})\|_{H_t^{-1}} + 8C\gamma_t(\|z_{i,t}(p_{i,t})\|_{H_t^{-1}} + \|x_{i,t}\|_{H_{v,t}^{-1}}) \\
&\leq \widetilde{u}'_{i,t}.
\end{aligned}
$$

From the definition of $S_t$, we have $\widetilde{v}_{i,t} \geq 0$ for $i \in S_t$. This is because if $\widetilde{v}_{i,t} < 0$ for some $i \in [N]$ then $i \notin S_t$. Then as in (15), we can show that

$$
\frac{\sum_{i \in S_t} \widetilde{v}_{i,t} \exp(\widetilde{u}_{i,t})}{1 + \sum_{i \in S_t} \exp(\widetilde{u}_{i,t})} \leq \frac{\sum_{i \in S_t} \widetilde{v}_{i,t} \exp(\widetilde{u}'_{i,t})}{1 + \sum_{i \in S_t} \exp(\widetilde{u}'_{i,t})}.
$$

Then we have

$$
\begin{aligned}
&\frac{\sum_{i \in S_t} \widetilde{v}_{i,t} \exp(\widetilde{u}_{i,t})}{1 + \sum_{i \in S_t} \exp(\widetilde{u}_{i,t})} - \frac{\sum_{i \in S_t} \underline{v}_{i,t}^+ \exp(\widetilde{u}_{i,t}(\Theta_u, \Theta_v))}{1 + \sum_{i \in S_t} \exp(\widetilde{u}_{i,t}(\Theta_u, \Theta_v))} \\
&\leq \frac{\sum_{i \in S_t} \widetilde{v}_{i,t} \exp(\widetilde{u}'_{i,t})}{1 + \sum_{i \in S_t} \exp(\widetilde{u}'_{i,t})} - \frac{\sum_{i \in S_t} \underline{v}_{i,t}^+ \exp(\widetilde{u}_{i,t}(\Theta_u, \Theta_v))}{1 + \sum_{i \in S_t} \exp(\widetilde{u}_{i,t}(\Theta_u, \Theta_v))} \\
&\leq \frac{\sum_{i \in S_t} \widetilde{v}_{i,t} \exp(\widetilde{u}'_{i,t})}{1 + \sum_{i \in S_t} \exp(\widetilde{u}'_{i,t})} - \frac{\sum_{i \in S_t} \underline{v}_{i,t}^+ \exp(\widetilde{u}'_{i,t})}{1 + \sum_{i \in S_t} \exp(\widetilde{u}'_{i,t})} + \frac{\sum_{i \in S_t} \underline{v}_{i,t}^+ \exp(\widetilde{u}'_{i,t})}{1 + \sum_{i \in S_t} \exp(\widetilde{u}'_{i,t})} - \frac{\sum_{i \in S_t} \underline{v}_{i,t}^+ \exp(\widetilde{u}_{i,t}(\Theta_u, \Theta_v))}{1 + \sum_{i \in S_t} \exp(\widetilde{u}_{i,t}(\Theta_u, \Theta_v))}.
\end{aligned}
$$

$$(34)$$

We define $\widehat{u}_{i,t} = z_{i,t}(p_{i,t})^\top \widehat{\theta}_t$. Then, for the first two terms in the above, we have

$$\frac{\sum_{i\in S_t}\widetilde{v}_{i,t}\exp(\widetilde{u}'_{i,t})}{1+\sum_{i\in S_t}\exp(\widetilde{u}'_{i,t})} - \frac{\sum_{i\in S_t}\underline{v}^+_{i,t}\exp(\widetilde{u}'_{i,t})}{1+\sum_{i\in S_t}\exp(\widetilde{u}'_{i,t})} = \frac{\sum_{i\in S_t}(\widetilde{v}_{i,t}-\underline{v}^+_{i,t})\exp(\widetilde{u}'_{i,t})}{1+\sum_{i\in S_t}\exp(\widetilde{u}'_{i,t})}$$

$$\leq \frac{\sum_{i\in S_t}(\widetilde{v}_{i,t}-\underline{v}_{i,t})\exp(\widetilde{u}'_{i,t})}{1+\sum_{i\in S_t}\exp(\widetilde{u}'_{i,t})} \leq \frac{\sum_{i\in S_t}(|\widetilde{v}_{i,t}-x^\top_{i,t}\widehat{\theta}_{v,t}|+|x^\top_{i,t}\widehat{\theta}_{v,t}-\underline{v}_{i,t}|)\exp(\widetilde{u}'_{i,t})}{1+\sum_{i\in S_t}\exp(\widetilde{u}'_{i,t})}$$

$$= \frac{\sum_{i\in S_t}(\gamma_t+\beta_t)\|x_{i,t}\|_{H^{-1}_{v,t}}\exp(\widetilde{u}'_{i,t})}{1+\sum_{i\in S_t}\exp(\widetilde{u}'_{i,t})} \leq \frac{\sum_{i\in S_t}2\gamma_t\|x_{i,t}\|_{H^{-1}_{v,t}}\exp(\widetilde{u}'_{i,t})}{1+\sum_{i\in S_t}\exp(\widetilde{u}'_{i,t})} \leq 2\gamma_t\max_{i\in S_t}\|x_{i,t}\|_{H^{-1}_{v,t}}.$$

$$(35)$$

For the latter two terms in (34), by following the same proof technique in Lemma 4 and using the fact that $|\widetilde{u}'_{i,t}-\widetilde{u}_{i,t}(\Theta_u,\Theta_v)| \leq |\widetilde{u}'_{i,t}-z_{i,t}(p_{i,t})^\top\widehat{\theta}_t|+|z_{i,t}(p_{i,t})^\top\widehat{\theta}_t-\widetilde{u}_{i,t}(\Theta_u,\Theta_v)| = O(\gamma_t(\|z_{i,t}(p_{i,t})\|_{H^{-1}_t}+\|x_{i,t}\|_{H^{-1}_{v,t}}))$ from $E_t$ and $\Theta_u\times\Theta_v \in \widetilde{\Theta}_t$ with $\beta_t\leq\gamma_t$, we can show that

$$\sup_{\Theta_u\times\Theta_v\in\widetilde{\Theta}_t}\left(\frac{\sum_{i\in S_t}\underline{v}^+_{i,t}\exp(\widetilde{u}'_{i,t})}{1+\sum_{i\in S_t}\exp(\widetilde{u}'_{i,t})} - \frac{\sum_{i\in S_t}\underline{v}^+_{i,t}\exp(\widetilde{u}_{i,t}(\Theta_u,\Theta_v))}{1+\sum_{i\in S_t}\exp(\widetilde{u}_{i,t}(\Theta_u,\Theta_v))}\right)$$

$$= O\left(\gamma^2_t(\max_{i\in S_t}\|z_{i,t}(p_{i,t})\|^2_{H^{-1}_t}+\max_{i\in S_t}\|x_{i,t}\|^2_{H^{-1}_{v,t}})+\gamma^2_t(\max_{i\in S_t}\|\widetilde{z}_{i,t}\|^2_{H^{-1}_t}+\max_{i\in S_t}\|\widetilde{x}_{i,t}\|^2_{H^{-1}_{v,t}})\right.$$

$$\left.+\gamma_t\sum_{i\in S_t}P_{t,\widehat{\theta}_t}(i|S_t,p_t)(\|\widetilde{z}_{i,t}\|_{H^{-1}_t}+\|\widetilde{x}_{i,t}\|_{H^{-1}_{v,t}})\right),$$

$$(36)$$

We can conclude the proof from (34), (35), and (36). ■

Then, for a bound of instantaneous regret of (33), we have

$$\mathbb{E}\left[\mathbb{E}\left[\left(\frac{\sum_{i\in A^*_t}v_{i,t}\exp(u_{i,t})}{1+\sum_{i\in A^*_t}\exp(u_{i,t})}-\frac{\sum_{i\in S_t}\underline{v}^+_{i,t}\exp(u^p_{i,t})}{1+\sum_{i\in S_t}\exp(u^p_{i,t})}\right)\mathbb{1}(E_t)\mid\mathcal{F}_{t-1}\right]\right]$$

$$\leq \mathbb{E}\left[\mathbb{E}\left[\left(\frac{\sum_{i\in A^*_t}v_{i,t}\exp(u_{i,t})}{1+\sum_{i\in A^*_t}\exp(u_{i,t})}-\inf_{\Theta_u\times\Theta_v\in\widetilde{\Theta}_t}\max_{S\subseteq[N]:|S|\leq K}\frac{\sum_{i\in S}\underline{v}^+_{i,t}\exp(\widetilde{u}_{i,t}(\Theta_u,\Theta_v))}{1+\sum_{i\in S}\exp(\widetilde{u}_{i,t}(\Theta_u,\Theta_v))}\right)\mathbb{1}(E_t)\mid\mathcal{F}_{t-1}\right]\right]$$

$$= \mathbb{E}\left[\mathbb{E}\left[\left(\frac{\sum_{i\in A^*_t}v_{i,t}\exp(u_{i,t})}{1+\sum_{i\in A^*_t}\exp(u_{i,t})}-\inf_{\Theta_u\times\Theta_v\in\widetilde{\Theta}_t}\max_{S\subseteq[N]:|S|\leq K}\frac{\sum_{i\in S}\underline{v}^+_{i,t}\exp(\widetilde{u}_{i,t}(\Theta_u,\Theta_v))}{1+\sum_{i\in S}\exp(\widetilde{u}_{i,t}(\Theta_u,\Theta_v))}\right)\mathbb{1}(E_t)\mid\mathcal{F}_{t-1},\widetilde{E}_t\right]\right]$$

$$\leq \mathbb{E}\left[\mathbb{E}\left[\left(\frac{\sum_{i\in A^*_t}v_{i,t}\exp(\widetilde{u}_{i,t})}{1+\sum_{i\in A^*_t}\exp(\widetilde{u}_{i,t})}-\inf_{\Theta_u\times\Theta_v\in\widetilde{\Theta}_t}\frac{\sum_{i\in S_t}\underline{v}^+_{i,t}\exp(\widetilde{u}_{i,t}(\Theta_u,\Theta_v))}{1+\sum_{i\in S_t}\exp(\widetilde{u}_{i,t}(\Theta_u,\Theta_v))}\right)\mathbb{1}(E_t)\mid\mathcal{F}_{t-1},\widetilde{E}_t\right]\right]$$

$$\leq \mathbb{E}\left[\mathbb{E}\left[\left(\frac{\sum_{i\in A^*_t}\widetilde{v}_{i,t}\exp(\widetilde{u}_{i,t})}{1+\sum_{i\in A^*_t}\exp(\widetilde{u}_{i,t})}-\inf_{\Theta_u\times\Theta_v\in\widetilde{\Theta}_t}\frac{\sum_{i\in S_t}\underline{v}^+_{i,t}\exp(\widetilde{u}_{i,t}(\Theta_u,\Theta_v))}{1+\sum_{i\in S_t}\exp(\widetilde{u}_{i,t}(\Theta_u,\Theta_v))}\right)\mathbb{1}(E_t)\mid\mathcal{F}_{t-1},\widetilde{E}_t\right]\right]$$

$$\leq \mathbb{E}\left[\mathbb{E}\left[\left(\frac{\sum_{i\in S_t}\widetilde{v}_{i,t}\exp(\widetilde{u}_{i,t})}{1+\sum_{i\in S_t}\exp(\widetilde{u}_{i,t})}-\inf_{\Theta_u\times\Theta_v\in\widetilde{\Theta}_t}\frac{\sum_{i\in S_t}\underline{v}^+_{i,t}\exp(\widetilde{u}_{i,t}(\Theta_u,\Theta_v))}{1+\sum_{i\in S_t}\exp(\widetilde{u}_{i,t}(\Theta_u,\Theta_v))}\right)\mathbb{1}(E_t)\mid\mathcal{F}_{t-1},\widetilde{E}_t\right]\right]$$

$$= \mathbb{E}\left[\mathbb{E}\left[\sup_{\Theta_u\times\Theta_v\in\widetilde{\Theta}_t}\left(\frac{\sum_{i\in S_t}\widetilde{v}_{i,t}\exp(\widetilde{u}_{i,t})}{1+\sum_{i\in S_t}\exp(\widetilde{u}_{i,t})}-\frac{\sum_{i\in S_t}\underline{v}^+_{i,t}\exp(\widetilde{u}_{i,t}(\Theta_u,\Theta_v))}{1+\sum_{i\in S_t}\exp(\widetilde{u}_{i,t}(\Theta_u,\Theta_v))}\right)\mathbb{1}(E_t)\mid\mathcal{F}_{t-1},\widetilde{E}_t\right]\right]$$

$$= O\left(\mathbb{E}\left[\mathbb{E}\left[\left(\gamma^2_t(\max_{i\in S_t}\|z_{i,t}(p_{i,t})\|^2_{H^{-1}_t}+\max_{i\in S_t}\|x_{i,t}\|^2_{H^{-1}_{v,t}})+\gamma^2_t(\max_{i\in S_t}\|\widetilde{z}_{i,t}\|^2_{H^{-1}_t}+\max_{i\in S_t}\|\widetilde{x}_{i,t}\|^2_{H^{-1}_{v,t}})\right.\right.\right.\right.$$

$$\left.\left.\left.\left.+\gamma_t\sum_{i\in S_t}P_{t,\widehat{\theta}_t}(i|S_t,p_t)(\|\widetilde{z}_{i,t}\|_{H^{-1}_t}+\|\widetilde{x}_{i,t}\|_{H^{-1}_{v,t}})+\gamma_t\max_{i\in S_t}\|x_{i,t}\|_{H^{-1}_{v,t}}\right)\mathbb{1}(E_t)\mid\mathcal{F}_{t-1},\widetilde{E}_t\right]\right]\right)$$

$$= O\left(\mathbb{E}\left[\mathbb{E}\left[\gamma_t^2(\max_{i \in S_t} \|z_{i,t}(p_{i,t})\|_{H_t^{-1}}^2 + \max_{i \in S_t} \|x_{i,t}\|_{H_{v,t}^{-1}}^2) + \gamma_t^2(\max_{i \in S_t} \|\widetilde{z}_{i,t}\|_{H_t^{-1}}^2 + \max_{i \in S_t} \|\widetilde{x}_{i,t}\|_{H_{v,t}^{-1}}^2)\right.\right.\right.$$

$$\left.\left.\left. + \gamma_t \sum_{i \in S_t} P_{t,\widehat{\theta}_t}(i|S_t, p_t)(\|\widetilde{z}_{i,t}\|_{H_t^{-1}} + \|\widetilde{x}_{i,t}\|_{H_{v,t}^{-1}}) + \gamma_t \max_{i \in S_t} \|x_{i,t}\|_{H_{v,t}^{-1}} \mid \mathcal{F}_{t-1}, \widetilde{E}_t, E_t\right] \times \mathbb{P}(E_t|\widetilde{E}_t, \mathcal{F}_{t-1})\right]\right)$$

$$= O\left(\mathbb{E}\left[\mathbb{E}\left[\gamma_t^2(\max_{i \in S_t} \|z_{i,t}(p_{i,t})\|_{H_t^{-1}}^2 + \max_{i \in S_t} \|x_{i,t}\|_{H_{v,t}^{-1}}^2) + \gamma_t^2(\max_{i \in S_t} \|\widetilde{z}_{i,t}\|_{H_t^{-1}}^2 + \max_{i \in S_t} \|\widetilde{x}_{i,t}\|_{H_{v,t}^{-1}}^2)\right.\right.\right.$$

$$\left.\left.\left. + \gamma_t \sum_{i \in S_t} P_{t,\widehat{\theta}_t}(i|S_t, p_t)(\|\widetilde{z}_{i,t}\|_{H_t^{-1}} + \|\widetilde{x}_{i,t}\|_{H_{v,t}^{-1}}) + \gamma_t \max_{i \in S_t} \|x_{i,t}\|_{H_{v,t}^{-1}} \mid \mathcal{F}_{t-1}, \widetilde{E}_t, E_t\right] \mathbb{P}(E_t|\mathcal{F}_{t-1})\right]\right),$$

$$\tag{37}$$

where the first equality comes from the independency of $\widetilde{E}_t$ given $\mathcal{F}_{t-1}$, the second inequality is obtained from $u_{i,t} \leq \widetilde{u}_{i,t}$ under the event $\widetilde{E}_t$ and from the definition of $S_t$, the third inequality is obtained from the fact that $v_{i,t}^+ \leq \widetilde{v}_{i,t}^+$ under $\widetilde{E}_t$, the third last equality is obtained from Lemma 10, and the last equality comes from independence between $E_t$ and $\widetilde{E}_t$ given $\mathcal{F}_{t-1}$.

We provide a lemma below for further analysis.

**Lemma 11** *For all $t \in [T]$, we have*

$$\mathbb{P}\left(\widetilde{v}_{i,t} \geq v_{i,t} \text{ and } \widetilde{u}_{i,t} \geq u_{i,t} \; \forall i \in [N] \mid \mathcal{F}_{t-1}, E_t\right) \geq \frac{1}{4\sqrt{e\pi}}.$$

**Proof** Given $\mathcal{F}_{t-1}$, $x_{i,t}^\top \widetilde{\theta}_{v,t}^{(m)}$ follows Gaussian distribution with mean $x_{i,t}^\top \widehat{\theta}_{v,t}$ and standard deviation $\beta_{\tau_t} \|x_{i,t}\|_{H_{v,t}^{-1}}$. Then we have

$$\mathbb{P}\left(\max_{m \in [M]} x_{i,t}^\top \widetilde{\theta}_{v,t}^{(m)} \geq x_{i,t}^\top \theta_v \; \forall i \in [N]|\mathcal{F}_{t-1}, E_t\right)$$

$$\geq 1 - N\mathbb{P}\left(x_{i,t}^\top \widetilde{\theta}_{v,t}^{(m)} < x_{i,t}^\top \theta_v \; \forall m \in [M]|\mathcal{F}_{t-1}, E_t\right)$$

$$\geq 1 - N\mathbb{P}\left(Z_m < \frac{x_{i,t}^\top \theta_v - x_{i,t}^\top \widehat{\theta}_{v,t}}{\beta_{\tau_t} \|x_{i,t}\|_{H_{v,t}^{-1}}} \; \forall m \in [M]|\mathcal{F}_{t-1}, E_t\right)$$

$$\geq 1 - N\mathbb{P}\left(Z < 1\right)^M,$$

where $Z_m$ and $Z$ are standard normal random variables. Likewise, we have

$$\mathbb{P}\left(\max_{m_1 \in [M]} z_{i,t}(p_{i,t})^\top \widetilde{\theta}_t^{(m_1)} + 8C \max_{m_2 \in [M]}(x_{i,t}^\top \widetilde{\theta}_{v,t}^{(m_2)} - x_{i,t}^\top \widehat{\theta}_{v,t}) \geq z_{i,t}(p_{i,t}^*)^\top \theta^* \; \forall i \in [N] \mid \mathcal{F}_{t-1}, E_t\right)$$

$$\geq \mathbb{P}\left(\max_{m \in [M]} z_{i,t}(p_{i,t})^\top \widetilde{\theta}_t^{(m)} + 8C(x_{i,t}^\top \widetilde{\theta}_{v,t}^{(m)} - x_{i,t}^\top \widehat{\theta}_{v,t}) \geq z_{i,t}(p_{i,t}^*)^\top \theta^* \; \forall i \in [N] \mid \mathcal{F}_{t-1}, E_t\right)$$

$$\geq 1 - N\mathbb{P}\left(z_{i,t}(p_{i,t})^\top \widetilde{\theta}_t^{(m)} + 8C(x_{i,t}^\top \widetilde{\theta}_{v,t}^{(m)} - x_{i,t}^\top \widehat{\theta}_{v,t}) < z_{i,t}(p_{i,t}^*)^\top \theta^* \; \forall m \in [M] \mid \mathcal{F}_{t-1}, E_t\right)$$

$$= 1 - N\mathbb{P}\left(\frac{z_{i,t}(p_{i,t})^\top \widetilde{\theta}_t^{(m)} - z_{i,t}(p_{i,t})^\top \widehat{\theta}_t + 8C(x_{i,t}^\top \widetilde{\theta}_{v,t}^{(m)} - x_{i,t}^\top \widehat{\theta}_{v,t})}{\beta_{\tau_t}\sqrt{2\|z_{i,t}(p_{i,t})\|_{H_t^{-1}}^2 + 8C\|x_{i,t}\|_{H_{v,t}^{-1}}^2}}\right.$$

$$\times \frac{\beta_{\tau_t}\sqrt{2\|z_{i,t}(p_{i,t})\|_{H_t^{-1}}^2 + 8C\|x_{i,t}\|_{H_{v,t}^{-1}}^2}}{\beta_{\tau_t}(\|z_{i,t}(p_{i,t})\|_{H_t^{-1}} + 2\sqrt{C}\|x_{i,t}\|_{H_{v,t}^{-1}})}$$

$$\left. < \frac{z_{i,t}(p_{i,t}^*)^\top \theta^* - z_{i,t}(p_{i,t})^\top \widehat{\theta}_t}{\beta_{\tau_t}(\|z_{i,t}(p_{i,t})\|_{V_t^{-1}} + 2\sqrt{C}\|x_{i,t}\|_{V_{v,t}^{-1}})} \; \forall m \in [M] \mid \mathcal{F}_{t-1}, E_t\right)$$

$$\geq 1 - N\mathbb{P}\left(Z_m \frac{\beta_{\tau_t}\sqrt{2\|z_{i,t}(p_{i,t})\|^2_{H_t^{-1}} + 8C\|x_{i,t}\|^2_{H_{v,t}^{-1}})}}{\beta_{\tau_t}(\|z_{i,t}(p_{i,t})\|_{H_t^{-1}} + 2\sqrt{C}\|x_{i,t}\|_{H_{v,t}^{-1}})}\right.$$

$$\left. < \frac{z_{i,t}(p_{i,t}^*)^\top \theta^* - z_{i,t}(p_{i,t})^\top \widehat{\theta}_t}{\beta_{\tau_t}(\|z_{i,t}(p_{i,t})\|_{V_t^{-1}} + 2\sqrt{C}\|x_{i,t}\|_{V_{v,t}^{-1}})}\ \forall m \in [M]\ |\ \mathcal{F}_{t-1}, E_t\right)$$

$$\geq 1 - N\mathbb{P}\left(Z_m < \frac{z_{i,t}(p_{i,t}^*)^\top \theta^* - z_{i,t}(p_{i,t})^\top \widehat{\theta}_t}{\beta_{\tau_t}(\|z_{i,t}(p_{i,t})\|_{V_t^{-1}} + 2\sqrt{C}\|x_{i,t}\|_{V_{v,t}^{-1}})}\ \forall m \in [M]\ |\ \mathcal{F}_{t-1}, E_t\right)$$

$$\geq 1 - N\mathbb{P}\left(Z < 1\right)^M,$$

where the third last inequality is obtained from the fact that the variance of $z_{i,t}(p_{i,t})^\top\widetilde{\theta}_t^{(m)} - z_{i,t}(p_{i,t})^\top\widehat{\theta}_t + 8C(x_{i,t}^\top\widetilde{\theta}_{v,t}^{(m)} - x_{i,t}^\top\widehat{\theta}_{v,t})$ is $\beta_{\tau_t}^2(2\|z_{i,t}(p_{i,t})\|^2_{H_t^{-1}} + 8C\|x_{i,t}\|^2_{H_{v,t}^{-1}})$ and second last inequality is obtained from $\sqrt{2(a^2+b^2)} \geq (a+b)$, and the last inequality is obtained from $u_{i,t} \leq \overline{u}_{i,t}$ in Lemma 1 and independency for $M$ samples.

Then using union bound, we have

$$\mathbb{P}\left(\widetilde{v}_{i,t} \geq v_{i,t}\ \text{and}\ \widetilde{u}_{i,t} \geq u_{i,t}\ \forall i \in [N]|\mathcal{F}_{t-1}, E_t\right)$$
$$\geq 1 - 2N\mathbb{P}\left(Z < 1\right)^M.$$
$$\geq 1 - 2N(1 - \frac{1}{4\sqrt{e\pi}})^M$$
$$\geq \frac{1}{4\sqrt{e\pi}},$$

where the second last inequality is obtained from $\mathbb{P}(Z \leq 1) \leq 1 - 1/4\sqrt{e\pi}$ using the anti-concentration of standard normal distribution, and the last inequality comes from $M = \lceil 1 - \frac{\log 2N}{\log(1-1/4\sqrt{e\pi})}\rceil$. This concludes the proof. ∎

From Lemmas 9 and 11, for $t \geq t_0$ for some constant $t_0 > 0$, we have

$$\mathbb{P}(\widetilde{E}_t|\mathcal{F}_{t-1}, E_t)$$
$$= \mathbb{P}\left(\widetilde{u}_{i,t} \geq u_{i,t}, \widetilde{v}_{i,t} \geq v_{i,t}\ \forall i \in [N]\ \text{and}\ \{\widetilde{\theta}_{v,t}^{(m)}\}_{m\in[M]} \in \widetilde{\Theta}_{v,t}, \{\widetilde{\theta}_t^{(m)}\}_{m\in[M]} \times \{\widetilde{\theta}_{v,t}^{(m)}\}_{m\in[M]} \in \widetilde{\Theta}_t|\mathcal{F}_{t-1}, E_t\right)$$
$$= \mathbb{P}\left(\widetilde{u}_{i,t} \geq u_{i,t}, \widetilde{v}_{i,t} \geq v_{i,t}\ \forall i \in [N]|\mathcal{F}_{t-1}, E_t\right)$$
$$\qquad - \mathbb{P}\left(\{\widetilde{\theta}_{v,t}^{(m)}\}_{m\in[M]} \notin \widetilde{\Theta}_{v,t}, \{\widetilde{\theta}_t^{(m)}\}_{m\in[M]} \times \{\widetilde{\theta}_{v,t}^{(m)}\}_{m\in[M]} \notin \widetilde{\Theta}_t|\mathcal{F}_{t-1}, E_t\right)$$
$$\geq 1/4\sqrt{e\pi} - \mathcal{O}(1/t^2)$$
$$\geq 1/8\sqrt{e\pi}.$$

For simplicity of the proof, we ignore the time steps before (constant) $t_0$, which does not affect our final result. For simplicity, we also use

$$L_t = \gamma_t^2(\max_{i\in S_t}\|z_{i,t}(p_{i,t})\|^2_{H_t^{-1}} + \max_{i\in S_t}\|x_{i,t}\|^2_{H_{v,t}^{-1}}) + \gamma_t^2(\max_{i\in S_t}\|\widetilde{z}_{i,t}\|^2_{H_t^{-1}} + \max_{i\in S_t}\|\widetilde{x}_{i,t}\|^2_{H_{v,t}^{-1}})$$
$$+ \gamma_t \sum_{i\in S_t} P_{t,\widehat{\theta}_t}(i|S_t, p_t)(\|\widetilde{z}_{i,t}\|_{H_t^{-1}} + \|\widetilde{x}_{i,t}\|_{H_{v,t}^{-1}}) + \gamma_t \max_{i\in S_t}\|x_{i,t}\|_{H_{v,t}^{-1}}.$$

Hence, we have

$$\mathbb{E}\left[L_t\ |\ \mathcal{F}_{t-1}, E_t\right] \geq \mathbb{E}\left[L_t\ |\ \mathcal{F}_{t-1}, E_t, \widetilde{E}_t\right]\mathbb{P}(\widetilde{E}_t|\mathcal{F}_{t-1}, E_t)$$
$$\geq \mathbb{E}\left[L_t\ |\ \mathcal{F}_{t-1}, E_t, \widetilde{E}_t\right]1/8\sqrt{e\pi}. \tag{38}$$

With (37) and (38), we have

$$\mathbb{E}\left[\left(\frac{\sum_{i \in A_t^*} v_{i,t} \exp(u_{i,t})}{1 + \sum_{i \in A_t^*} \exp(u_{i,t})} - \frac{\sum_{i \in S_t} \underline{v}_{i,t}^+ \exp(u_{i,t}^p)}{1 + \sum_{i \in S_t} \exp(u_{i,t}^p)}\right) \mathbb{1}(E_t) \mid \mathcal{F}_{t-1}\right]$$

$$= O\left(\mathbb{E}\left[L_t \mid \mathcal{F}_{t-1}, \widetilde{E}_t, E_t\right] \mathbb{P}(E_t \mid \mathcal{F}_{t-1})\right)$$

$$= O\left(\mathbb{E}\left[L_t \mid \mathcal{F}_{t-1}, E_t\right] \mathbb{P}(E_t \mid \mathcal{F}_{t-1})\right). \tag{39}$$

Then from (33), (39), (30), (31) and Lemma 5, 6, 8, with $E_T^c \supset E_{T-1}^c, \ldots, \supset E_1^c$, we have

$$R^\pi(T) = \sum_{t \in [T]} \mathbb{E}[R_t(S_t^*, p_t^*) - R_t(S_t, p_t)\mathbb{1}(E_t)] + \sum_{t \in [T]} \mathbb{E}[R_t(S_t^*, p_t^*) - R_t(S_t, p_t)\mathbb{1}(E_t^c)]$$

$$\leq \sum_{t \in [T]} \mathbb{E}\left[\left(\frac{\sum_{i \in A_t^*} p_{i,t}^* \exp(u_{i,t})}{1 + \sum_{i \in A_t^*} \exp(u_{i,t})} - \frac{\sum_{i \in S_t} \underline{v}_{i,t}^+ \exp(u_{i,t}^p)\mathbb{1}(\underline{v}_{i,t}^+ \leq v_{i,t})}{1 + \sum_{i \in S_t} \exp(u_{i,t}^p)\mathbb{1}(\underline{v}_{i,t}^+ \leq v_{i,t})}\right)\mathbb{1}(E_t)\right] + \sum_{t \in [T]} \mathbb{P}[E_T^c]$$

$$= O\left(\sum_{t \in [T]} \mathbb{E}\left[\mathbb{E}\left[L_t \mid \mathcal{F}_{t-1}, E_t\right] \mathbb{P}(E_t \mid \mathcal{F}_{t-1})\right]\right)$$

$$= O\left(\sum_{t \in [T]} \mathbb{E}\left[L_t \mathbb{1}(E_t)\right]\right)$$

$$= \widetilde{O}\left(\mathbb{E}\left[\sqrt{d}\beta_{\tau_T}\sqrt{T}\sqrt{\sum_{t \in [T]} \max_{i \in S_t} \|x_{i,t}\|_{H_{v,t}^{-1}}^2} + +\sqrt{d}\beta_{\tau_T}\sqrt{\sum_{t \in [T]}\sum_{i \in S_t} P_{t,\widehat{\theta}_t}(i|S_t, p_t)\times}\right.\right.$$

$$\left.\left.\left(\sqrt{\sum_{t \in [T]}\sum_{i \in S_t} P_{t,\widehat{\theta}_t}(i|S_t, p_t)\|\widetilde{x}_{i,t}\|_{H_{v,t}^{-1}}^2} + \sqrt{\sum_{t \in [T]}\sum_{i \in S_t} P_{t,\widehat{\theta}_t}(i|S_t, p_t)\|\widetilde{z}_{i,t}\|_{H_t^{-1}}^2\mathbb{1}(E_t)}\right)\right] + \frac{d^2}{\kappa}\mathbb{E}[\beta_{\tau_T^2}]\right)$$

$$= \widetilde{O}\left(\mathbb{E}[\beta_{\tau_T}]d\sqrt{T/\kappa} + \frac{d^4}{\kappa}\right)$$

$$= \widetilde{O}\left(d^2\sqrt{T/\kappa} + \frac{d^4}{\kappa}\right).$$

## A.4 RANDOMNESS IN ACTIVATION FUNCTION

In this section, we study the case where there exists randomness in the activation function of C-MNL. Let $\zeta_{i,t}$ be a zero-mean random noise drawn from the range of $[-c, c]$ for some $0 < c \leq 1$. Then the noisy activation is modeled in C-MNL as

$$\widetilde{\mathbb{P}}_t(i|S_t, p_t) = \frac{\exp(z_{i,t}(p_{i,t})^\top \theta^*)\mathbb{1}(p_{i,t} \leq (x_{i,t}^\top \theta_v + \zeta_{i,t})^+)}{1 + \sum_{j \in S_t} \exp(z_{j,t}(p_{j,t})^\top \theta^*)\mathbb{1}(p_{j,t} \leq (x_{j,t}^\top \theta_v + \zeta_{j,t})^+)}.$$

### A.4.1 ALGORITHM & REGRET ANALYSIS

Here we provide an algorithm (Algorithm 3) for the random activation C-MNL. The different part from Algorithm 1 is in pricing strategy such that $p_{i,t} = (\underline{v}_{i,t} - c)^+$. The remaining parts are the same.

Now we provide a regret bound of the algorithm in the following.

**Theorem 3** *Under Assumption 1, the policy $\pi$ of Algorithm 3 achieves a regret bound of*

$$R^\pi(T) = \widetilde{O}\left(d^{\frac{3}{2}}\sqrt{T/\kappa} + cT\right).$$

---

**Algorithm 3** UCB-based Assortment-selection with Robust-LCB Pricing (UCBA-RLCBP)

**Input:** $\lambda, \eta, \beta_\tau, c$
**Init:** $\tau \leftarrow 1, t_1 \leftarrow 1, \widehat{\theta}_{v,(1)} \leftarrow \mathbf{0}_d$
**for** $t = 1, \ldots, T$ **do**
    $\widetilde{H}_t \leftarrow \lambda I_{2d} + \sum_{s=1}^{t-2} G_s(\widehat{\theta}_s) + \eta G_{t-1}(\widehat{\theta}_{t-1})$ with (3)
    $H_t \leftarrow \lambda I_{2d} + \sum_{s=1}^{t-1} G_s(\widehat{\theta}_s)$ with (3)
    $H_{v,t} \leftarrow \lambda I_d + \sum_{s=1}^{t-1} G_{v,s}(\widehat{\theta}_s)$ with (3)
    $\widehat{\theta}_t \leftarrow \arg\min_{\theta \in \Theta} g_t(\widehat{\theta}_{t-1})^\top \theta + \frac{1}{2\eta}\|\theta - \widehat{\theta}_{t-1}\|_{\widetilde{H}_t^{-1}}^2$ with (2) ;        ▷ `Estimation`
    **if** $\det(H_t) > 2\det(H_{t_\tau})$ **then**
        $\tau \leftarrow \tau + 1; t_\tau \leftarrow t$
        $\widehat{\theta}_{v,(\tau)} \leftarrow \widehat{\theta}_{v,t_\tau}(= \widehat{\theta}_{t_\tau}^{1:d})$
    **for** $i \in [N]$ **do**
        $\underline{v}_{i,t} \leftarrow x_{i,t}^\top \widehat{\theta}_{v,(\tau)} - \sqrt{2}\beta_t\|x_{i,t}\|_{H_{v,t}^{-1}}$ ;        ▷ `LCB for valuation`
        $p_{i,t} \leftarrow (\underline{v}_{i,t} - c)^+$ ;        ▷ **`Price selection w/ LCB`**
        $\overline{v}_{i,t} \leftarrow x_{i,t}^\top \widehat{\theta}_{v,t} + \beta_t\|x_{i,t}\|_{H_{v,t}^{-1}}$ ;        ▷ `UCB for valuation`
        $\overline{u}_{i,t}^c \leftarrow z_{i,t}(p_{i,t})^\top \widehat{\theta}_t + \beta_t\|z_{i,t}(p_{i,t})\|_{H_t^{-1}} + 2\sqrt{2}\beta_t\|x_{i,t}\|_{H_{v,t}^{-1}} + c$ ;  ▷ `UCB for utility`
    $S_t \leftarrow \arg\max_{S \subseteq [N]:|S| \leq L} \sum_{i \in S} \frac{\overline{v}_{i,t}\exp(\overline{u}_{i,t})}{1+\sum_{j \in S}\exp(\overline{u}_{j,t})}$;    ▷ **`Assortment selection w/ UCB`**
    Offer $S_t$ with prices $p_t = \{p_{i,t}\}_{i \in S_t}$
    Observe preference (purchase) feedback $y_{i,t} \in \{0,1\}$ for $i \in S_t$

---

Therefore, if we have $c = O(1/\sqrt{T})$, the regret bound in the above theorem becomes $\widetilde{O}(d^{\frac{3}{2}}\sqrt{T/\kappa})$ same as that in Theorem 1 for the case without the noise in activation functions.

**Proof** Here we provide only the different parts from the proof of Theorem 1. Let $\underline{v}_{i,t}^c = (\underline{v}_{i,t} - c)$ and $\overline{u}_{i,t}^{\prime c} = z_{i,t}(p_{i,t})^\top \theta^* + 2\sqrt{2}\beta_{\tau_t}\|z_{i,t}(p_{i,t})\|_{H_t^{-1}} + 2\sqrt{2}\beta_{\tau_t}\|x_{i,t}\|_{H_{v,t}^{-1}} + c$. Then we can observe that under $E_t$, $p_{i,t} \leq v_{i,t} + \zeta_{i,t}$ and $\overline{u}_{i,t} \leq \overline{u}_{i,t}'$. From (12) and Lemma 2, under $E_t$, we have

$$R_t(S_t^*, p_t^*) - R_t(S_t, p_t)$$
$$\leq \frac{\sum_{i \in S_t}\overline{v}_{i,t}\exp(\overline{u}_{i,t}^{\prime c})}{1 + \sum_{i \in S_t}\exp(\overline{u}_{i,t}^{\prime c})} - \frac{\sum_{i \in S_t}\underline{v}_{i,t}^{c+}\exp(\overline{u}_{i,t}^{\prime c})}{1 + \sum_{i \in S_t}\exp(\overline{u}_{i,t}^{\prime c})} + \frac{\sum_{i \in S_t}\underline{v}_{i,t}^{c+}\exp(\overline{u}_{i,t}^{\prime c})}{1 + \sum_{i \in S_t}\exp(\overline{u}_{i,t}^{\prime c})} - \frac{\sum_{i \in S_t}\underline{v}_{i,t}^{c+}\exp(u_{i,t}^p)}{1 + \sum_{i \in S_t}\exp(u_{i,t}^p)}.$$
(40)

By following the proof of Lemmas 3 and 4, under $E_t$, we can show that

$(a)$   $\dfrac{\sum_{i \in S_t}\overline{v}_{i,t}\exp(\overline{u}_{i,t}')}{1 + \sum_{i \in S_t}\exp(\overline{u}_{i,t}')} - \dfrac{\sum_{i \in S_t}\underline{v}_{i,t}^{c+}\exp(\overline{u}_{i,t}')}{1 + \sum_{i \in S_t}\exp(\overline{u}_{i,t}')}$

      $= O\left(\beta_{\tau_t}^2 \max_{i \in S_t}\|x_{i,t}\|_{H_{v,t}^{-1}}^2 + \beta_{\tau_t}^2 \max_{i \in S_t}\|z_{i,t}(p_{i,t})\|_{H_t^{-1}}^2 + \beta_{\tau_t}\max_{i \in S_t}\|x_{i,t}\|_{H_{v,t}^{-1}} + c\right)$,

$(b)$   $\dfrac{\sum_{i \in S_t}\underline{v}_{i,t}^{c+}\exp(\overline{u}_{i,t}')}{1 + \sum_{i \in S_t}\exp(\overline{u}_{i,t}')} - \dfrac{\sum_{i \in S_t}\underline{v}_{i,t}^{c+}\exp(u_{i,t}^p)}{1 + \sum_{i \in S_t}\exp(u_{i,t}^p)}$

      $= O\left(\beta_{\tau_t}^2(\max_{i \in S_t}\|z_{i,t}(p_{i,t})\|_{H_t^{-1}}^2 + \max_{i \in S_t}\|x_{i,t}\|_{H_{v,t}^{-1}}^2) + \beta_{\tau_t}^2(\max_{i \in S_t}\|\widetilde{z}_{i,t}\|_{H_t^{-1}}^2 + \max_{i \in S_t}\|\widetilde{x}_{i,t}\|_{H_{v,t}^{-1}}^2)\right.$

        $\left. + \beta_{\tau_t}\sum_{i \in S_t}P_{t,\widehat{\theta}_t}(i|S_t, p_t)(\|\widetilde{z}_{i,t}\|_{H_t^{-1}} + \|\widetilde{x}_{i,t}\|_{H_{v,t}^{-1}}) + c\right)$.

Then by following the proof steps of Theorem 1, we can show that

$$R^\pi(T) = \widetilde{O}\left(d^{\frac{3}{2}}\sqrt{T/\kappa} + cT + \frac{d^3}{\kappa}\right).$$

■

### A.5 PROOF OF LEMMA 4

Here we utilize some proof techniques in Lee & Oh (2024). Let $Q(u) = \frac{\sum_{i \in S_t} v_{i,t}^+ \exp(u_i)}{1 + \sum_{i \in S_t} \exp(u_i)}$ and $u_t^p = [u_{i,t}^p : i \in S_t]$. Then by applying a second-order Taylor expansion, there exists $\xi_t' = (1 - c)u_t^p + c\overline{u}_t'$ for some $c \in (0, 1)$ such that

$$
\frac{\sum_{i \in S_t} v_{i,t}^+ \exp(\overline{u}_{i,t}')}{1 + \sum_{i \in S_t} \exp(\overline{u}_{i,t}')} - \frac{\sum_{i \in S_t} v_{i,t}^+ \exp(u_{i,t}^p)}{1 + \sum_{i \in S_t} \exp(u_{i,t}^p)}
$$
$$
= \sum_{i \in S_t} \nabla_i Q(u_t^p)(\overline{u}_{i,t}' - u_{i,t}^p) + \frac{1}{2} \sum_{i \in S_t} \sum_{j \in S_t} (\overline{u}_{i,t}' - u_{i,t}^p) \nabla_{ij} Q(\xi_t')(\overline{u}_{i,t}' - u_{i,t}^p). \tag{41}
$$

Let $x_{i_0,t} = \mathbf{0}_d$ and $w_{i_0,t} = \mathbf{0}_d$ implying $z_{i_0,t} = \mathbf{0}_{2d}$. Then for the first order term in the above, we have

$$
\sum_{i \in S_t} \nabla_i Q(u_t^p)(\overline{u}_{i,t}' - u_{i,t}^p)
$$
$$
= \sum_{i \in S_t} v_{i,t}^+ P_{i,t}(u_t^p)(\overline{u}_{i,t}' - u_{i,t}^p) - \sum_{i,j \in S_t} v_{i,t}^+ P_{i,t}(u_t^p) P_{j,t}(u_t^p)(\overline{u}_{j,t}' - u_{j,t}^p)
$$
$$
= \sum_{i \in S_t} 2\sqrt{C}\beta_t \underline{v}_{i,t}^+ P_{i,t}(u_t^p)(\|z_{i,t}(p_{i,t})\|_{H_t^{-1}} + \|x_{i,t}\|_{H_{v,t}^{-1}})
$$
$$
- \sum_{i,j \in S_t} 2\sqrt{C}\beta_t \underline{v}_{i,t}^+ P_{i,t}(u_t^p) P_{j,t}(u_t^p)(\|z_{j,t}(p_{j,t})\|_{H_t^{-1}} + \|x_{j,t}\|_{H_{v,t}^{-1}})
$$
$$
= \sum_{i \in S_t} 2\sqrt{C}\beta_t \underline{v}_{i,t}^+ P_{i,t}(u_t^p)
$$
$$
\times \left( \|z_{i,t}(p_{i,t})\|_{H_t^{-1}} - \sum_{j \in S_t} P_{j,t}(u_t^p)\|z_{j,t}(p_{j,t})\|_{H_t^{-1}} + \|x_{i,t}\|_{H_{v,t}^{-1}} - \sum_{j \in S_t} P_{j,t}(u_t^p)\|x_{j,t}\|_{H_{v,t}^{-1}} \right).
$$

For the first two terms in the above, we have

$$
\|z_{i,t}(p_{i,t})\|_{H_t^{-1}} - \sum_{j \in S_t} P_{j,t}(u_t^p)\|z_{j,t}(p_{j,t})\|_{H_t^{-1}}
$$
$$
= \|z_{i,t}(p_{i,t})\|_{H_t^{-1}} - \sum_{j \in S_t \cup \{i_0\}} P_{j,t}(u_t^p)\|z_{j,t}(p_{j,t})\|_{H_t^{-1}}
$$
$$
= \|z_{i,t}(p_{i,t})\|_{H_t^{-1}} - \mathbb{E}_{j \sim P_{t,\theta^*}(\cdot | S_t, p_t)} \left[ \|z_{j,t}(p_{j,t})\|_{H_t^{-1}} \right]
$$
$$
\leq \|z_{i,t}(p_{i,t})\|_{H_t^{-1}} - \left\| \mathbb{E}_{j \sim P_{t,\theta^*}(\cdot | S_t, p_t)} [z_{j,t}(p_{j,t})] \right\|_{H_t^{-1}}
$$
$$
\leq \left\| z_{i,t}(p_{i,t}) - \mathbb{E}_{j \sim P_{t,\theta^*}(\cdot | S_t, p_t)} [z_{j,t}(p_{j,t})] \right\|_{H_t^{-1}},
$$

where the first inequality is obtained from Jensen's inequality and the last inequality is from $\|a\| = \|a - b + b\| \leq \|a - b\| + \|b\|$. By following the proof steps in (H.1), (H.2), (H.3), and (H.4) in Lee

& Oh (2024), we can show that

$$\sum_{i \in S_t} \underline{v}^+_{i,t} P_{i,t}(u^p_t) \left\| z_{i,t}(p_{i,t}) - \mathbb{E}_{j \sim P_{t,\theta^*}(\cdot|S_t,p_t)} [z_{j,t}(p_{j,t})] \right\|_{H_t^{-1}}$$

$$\leq \sum_{i \in S_t} P_{i,t}(u^p_t) \left\| z_{i,t}(p_{i,t}) - \mathbb{E}_{j \sim P_{t,\theta^*}(\cdot|S_t,p_t)} [z_{j,t}(p_{j,t})] \right\|_{H_t^{-1}}$$

$$= O \left( \beta_{\tau_t} \max_{i \in S_t} \|z_{i,t}(p_{i,t})\|^2_{H_t^{-1}} + \beta_{\tau_t} \max_{i \in S_t} \|\widetilde{z}_{i,t}\|^2_{H_t^{-1}} + \sum_{i \in S_t} P_{t,\widehat{\theta}_t}(i|S_t,p_t)\|\widetilde{z}_{i,t}\|_{H_t^{-1}} \right),$$

where the first inequality is obtained from $0 \leq \underline{v}^+_{i,t} \leq 1$ under $E_t$.

Then, likewise, we can show that

$$\sum_{i \in S_t} \underline{v}^+_{i,t} P_{i,t}(u^p_t) \left( \|x_{i,t}\|_{H_{v,t}^{-1}} - \sum_{j \in S_t} P_{j,t}(u^p_t)\|x_{j,t}\|_{H_{v,t}^{-1}} \right)$$

$$\leq \sum_{i \in S_t} P_{i,t}(u^p_t) \left\| x_{i,t} - \mathbb{E}_{j \sim P_{t,\theta^*}(\cdot|S_t,p_t)} [x_{j,t}] \right\|_{H_{v,t}^{-1}}$$

$$= O \left( \beta_{\tau_t} \max_{i \in S_t} \|x_{i,t}\|^2_{H_{v,t}^{-1}} + \beta_{\tau_t} \max_{i \in S_t} \|\widetilde{x}_{i,t}\|^2_{H_{v,t}^{-1}} + \sum_{i \in S_t} P_{t,\widehat{\theta}_t}(i|S_t,p_t)\|\widetilde{x}_{i,t}\|_{H_{v,t}^{-1}} \right).$$

Putting the above results together, for the first-order term, we have

$$\sum_{i \in S_t} \nabla_i Q(u_t)(\overline{u}'_{i,t} - u_{i,t})$$

$$= O \left( \beta^2_{\tau_t} (\max_{i \in S_t} \|z_{i,t}(p_{i,t})\|^2_{H_t^{-1}} + \max_{i \in S_t} \|x_{i,t}\|^2_{H_{v,t}^{-1}}) + \beta^2_{\tau_t} (\max_{i \in S_t} \|\widetilde{z}_{i,t}\|^2_{H_t^{-1}} + \max_{i \in S_t} \|\widetilde{x}_{i,t}\|^2_{H_{v,t}^{-1}}) \right.$$

$$\left. + \beta_{\tau_t} \sum_{i \in S_t} P_{t,\widehat{\theta}_t}(i|S_t,p_t)(\|\widetilde{z}_{i,t}\|_{H_t^{-1}} + \|\widetilde{x}_{i,t}\|_{H_{v,t}^{-1}}) \right). \tag{42}$$

Now we provide a bound for the second order term. By following the proof steps in (H.6) in Lee & Oh (2024) with $0 \leq \underline{v}^+_{i,t} \leq 1$ under $E_t$, we can show that

$$\frac{1}{2} \sum_{i,j \in S_t} (\overline{u}'_{i,t} - u_{i,t})\nabla_{ij} Q(\xi'_t)(\overline{u}'_{i,t} - u_{i,t}) = O \left( \beta^2_{\tau_t} (\max_{i \in S_t} \|z_{i,t}(p_{i,t})\|^2_{H_t^{-1}} + \max_{i \in S_t} \|x_{i,t}\|^2_{H_{v,t}^{-1}}) \right). \tag{43}$$

Then we can conclude the proof by (41), (42), and (43).

## A.6 PROOF OF LEMMA 7

For $1 \leq t \leq t_2 - 1$, since $p_{i,t} = 0$ from the algorithm, we have $y_{i,t} \sim \mathbb{P}_t(\cdot|S_t,p_t) = P_{t,\theta^*}(\cdot|S_t,p_t)$. Then from Lemma 1 in Lee & Oh (2024), for $1 \leq t \leq t_2$, we can show that $\mathbb{P}(E_t) \geq 1 - \frac{1}{T^2}$.

Now, we provide a proof for the time steps $t_\tau + 1 \leq t \leq t_{\tau+1}$ for $\tau \geq 2$. We utilize the proof procedure in Lemma 1 in Lee & Oh (2024). The main difference lies in focusing on the *conditional* probability for a good event in our proof. Under $E_{t_\tau}$, for $t_\tau \leq t \leq t_{\tau+1} - 1$, since $\underline{v}_{i,t} \leq v_{i,t}$, we have $y_{i,t} \sim \mathbb{P}_t(\cdot|S_t,p_t) = P_{t,\theta^*}(\cdot|S_t,p_t)$. Then from Lemma F.1 in the previous work, we can show that for $t_\tau + 1 \leq t \leq t_{\tau+1}$, with $\eta = \frac{1}{2}\log(K+1) + 3$ and $\lambda \geq 1$, we have

$$\|\widehat{\theta}_t - \theta^*\|^2_{H_t} \leq 2\eta \left( \sum_{s=t_\tau}^{t-1} f_s(\theta^*) - f_s(\widehat{\theta}_{s+1}) \right) + \|\widehat{\theta}_{t_\tau} - \theta^*\|^2_{H_{t_\tau}} + 96\sqrt{2}\eta \sum_{s=t_\tau}^{t-1} \|\widehat{\theta}_{s+1} - \widehat{\theta}_s\|^2_2$$

$$- \sum_{s=t_\tau}^{t-1} \|\widehat{\theta}_{s+1} - \widehat{\theta}_s\|^2_{H_s}.$$

$$\tag{44}$$

Then from Lemmas 14 and 15, for any $c > 0$ with $\lambda \geq 84d\eta$, we can show that with probability at least $1 - \delta$,

$$\sum_{s=t_\tau}^{t-1} f_s(\theta^*) - f_s(\widehat{\theta}_{s+1})$$

$$\leq (3\log(1 + (K+1)t) + 3)\left(\frac{17}{16}\lambda + 2\sqrt{\lambda}\log\left(2\sqrt{1+2t}/\delta\right) + 16\left(\log(2\sqrt{1+2t}/\delta)\right)^2\right) + 2$$

$$+ \frac{1}{2c}\sum_{s=t_\tau}^{t-1}\|\widehat{\theta}_s - \widehat{\theta}_{s+1}\|_{H_s}^2 + 2\sqrt{6}cd\log(1 + (t+1)/2\lambda). \tag{45}$$

By setting $c = 2\eta$ and with $\lambda \geq 192\sqrt{2}\eta$, we have

$$96\sqrt{2}\eta\sum_{s=t_\tau}^{t-1}\|\widehat{\theta}_{s+1} - \widehat{\theta}_s\|_2^2 + \left(\frac{\eta}{c} - 1\right)\sum_{s=t_\tau}^{t-1}\|\widehat{\theta}_{s+1} - \widehat{\theta}_s\|_{H_s}^2$$

$$= 96\sqrt{2}\eta\sum_{s=t_\tau}^{t-1}\|\widehat{\theta}_{s+1} - \widehat{\theta}_s\|_2^2 + \left(\frac{\eta}{c} - 1\right)\sum_{s=t_\tau}^{t-1}\|\widehat{\theta}_{s+1} - \widehat{\theta}_s\|_{H_s}^2$$

$$\leq \left(96\sqrt{2}\eta - \frac{\lambda}{2}\right)\sum_{s=t_\tau}^{t}\|\widehat{\theta}_{s+1} - \widehat{\theta}_s\|_2^2 \leq 0, \tag{46}$$

where the first inequality comes from $H_s \succeq \lambda I_{2d}$. Set $\delta = 1/T^2$. Then under $E_{t_\tau}$, from (44), (45), (46), with probability at least $1 - 1/T^2$, we obtain

$$\|\widehat{\theta}_t - \theta^*\|_{H_t}^2$$

$$\leq \eta(6\log(1 + (K+1)t) + 6)\left(\frac{17}{16}\lambda + 2\sqrt{\lambda}\log\left(2\sqrt{1+2t}T^2\right) + 16\left(\log(2\sqrt{1+2t}T^2)\right)^2\right) + 4\eta$$

$$+ 4\eta\sqrt{6}cd\log(1 + (t+1)/2\lambda) + \|\widehat{\theta}_{t_\tau} - \theta^*\|_{H_{t_\tau}}^2$$

$$\leq \eta(6\log(1 + (K+1)t) + 6)\left(\frac{17}{16}\lambda + 2\sqrt{\lambda}\log\left(2\sqrt{1+2t}T^2\right) + 16\left(\log(2\sqrt{1+2t}T^2)\right)^2\right) + 4\eta$$

$$+ 4\eta\sqrt{6}cd\log(1 + (t+1)/2\lambda) + \beta_\tau^2 = \beta_{\tau+1}^2.$$

Finally, we can conclude that, for $1 \leq t \leq t_2$, we have $\mathbb{P}(E_t) \geq 1 - \frac{1}{T^2}$, and for $\tau \geq 2$ and $t_\tau + 1 \leq t \leq t_{\tau+1}$, we have $\mathbb{P}(E_t|E_{t_\tau}) \geq 1 - \frac{1}{T^2}$.

### A.7 NECESSARY LEMMAS

**Lemma 12 (Lemma 12 in Abbasi-Yadkori et al. (2011))** *Let $A, B,$ and $C$ be positive semi-definite matrices such that $A = B + C$. Then we have*

$$\sup_{x \neq 0}\frac{x^\top A x}{x^\top B x} \leq \frac{\det(A)}{\det(B)}.$$

**Lemma 13 (Lemma 10 in Abbasi-Yadkori et al. (2011))** *Suppose $X_1, X_2, \ldots, X_t \in \mathbb{R}^d$ and for any $1 \leq s \leq t$, $\|X_s\|_2 \leq L$. Let $V_{t+1} = \lambda I + \sum_{s=1}^{t} X_s X_s^\top$ for some $\lambda > 0$. Then we have*

$$\det(V_{t+1}) \leq (\lambda + tL^2/d))^d.$$

We define $\sigma_t(z) : \mathbb{R}^{S_t} \to \mathbb{R}^{S_t}$ such that $[\sigma_t(z)]_i = \frac{\exp(z_i)}{1+\sum_{j=1}^{S_t}\exp(z_j)}$. We also denote the probability of choosing the outside option as $[\sigma_t(z)]_0 = \frac{1}{1+\sum_{j=1}^{S_t}\exp(z_j)}$ with $i_0 := 0$. We define a pseudo-inverse function of $\sigma_t(\cdot)$ such that $\sigma(\sigma^+(p)) = p$ for any $q \in \{p \in [0,1]^{S_t}|\|p\|_1 < 1\}$. We

can observe that $\sigma_t^+ : \mathbb{R}^{S_t} \to \mathbb{R}^{S_t}$ where $[\sigma_t^+(q)]_i = \log(q_i/(1 - \|q\|_1))$ for any $q \in \{p \in [0,1]^{S_t} | \|p\|_1 < 1\}$. We also define $\widetilde{z}_s = \sigma_s^+(\mathbb{E}_{w \sim P_s}[\sigma_s([z_{i,t}(p_{i,t})^\top w]_{i \in S_s})])$ and $P_s = \mathcal{N}(\widehat{\theta}_s, (1 + cH_s^{-1}))$ for a positive constant $c > 0$. We define $f_t(z, y) = \sum_{i=0}^{S_t} \mathbb{1}(y_{i,t}) \log(\frac{1}{[\sigma_t(z)]_i})$. Then we have the following lemmas.

**Lemma 14 (Lemma F.2 in Lee & Oh (2024))** *Let $\delta \in (0, 1]$ and $\lambda \geq 1$. For $\tau > 2$ and $t_\tau + 1 \leq t \leq t_{\tau+1}$, under $E_{t_\tau}$, with probability at least $1 - \delta$, we have*

$$\sum_{s=t_\tau}^{t-1} f_s(\theta^*) - \sum_{s=1}^{t} f_s(\widetilde{z}_s, y_s)$$

$$\leq (3\log(1 + (K+1)t) + 3)\left(\frac{17}{16}\lambda + 2\sqrt{\lambda}\log\left(\frac{2\sqrt{1+2t}}{\delta}\right) + 16\left(\log\left(\frac{2\sqrt{1+2t}}{\delta}\right)\right)^2\right) + 2.$$

**Lemma 15 (Lemma F.3 in Lee & Oh (2024))** *For any $c > 0$, let $\lambda \geq \max\{2, 72cd\}$. For $\tau > 2$ and $t_\tau + 1 \leq t \leq t_{\tau+1}$, under $E_{t_\tau}$, we have*

$$\sum_{s=t_\tau}^{t-1} f_s(\widetilde{z}_s, y_s) - f_s(\widehat{\theta}_{s+1}) \leq \frac{1}{2c}\sum_{s=t_\tau}^{t-1} \|\widehat{\theta}_s - \widehat{\theta}_{s+1}\|_{H_s}^2 + \sqrt{6}cd\log\left(1 + \frac{t+1}{2\lambda}\right).$$

