# OpenReview forum: "Dynamic Assortment Selection and Pricing with Censored Preference Feedback"
_ICLR.cc/2025/Conference — ICLR 2025 Poster_

### Official Review · Reviewer_Ues7 · 2024-10-26

**Soundness:** 2
**Presentation:** 3
**Contribution:** 3
**Rating:** 6
**Confidence:** 3

**Summary:**

The paper considers both multi-product selection and pricing under censorship. The authors use a new way to find sublinear regret that uses LCB to price items and UCB/TS to select assortment. They achieve optimal regret bounds with respect to $T$ and use some simulations to verify though lacking comparable benchmarks.

**Strengths:**

The way of applying LCB to pricing (actually exploration) is inspiring. The extension from MNL to C-MNL is practical in the real world. The usage of TS is quite innovative.

**Weaknesses:**

1. My main concern is the statement that $1/\kappa=O(K^2)$. Is it true? If $|S|=K$, since exp>0, we know that $P(i_0)\le1/K$. In the meantime, choosing $i$ with the minimum exp term, it seems that $P(i)\le 1/K$ as well. Therefore, $1/\kappa\gtrsim \Omega(K^2)$. Please let me know if I missed something. Otherwise, the results will become much weaker.

2. Since different items have different contexts, it's more reasonable if the buyer can buy more than one item at the same time. It'll be an interesting extension to consider this scenario.

3. I'm wondering if it's possible to get the logarithmic problem-dependent regret as literature in both dynamic pricing and MAB. Maybe you can do some experiments on this. For example, use regression to find the actual order of $T$ (regress log(Regret) on log(T)).

**Questions:**

For the TS, you use the maximum of m samples. Then, with high probability, $x^T\tilde\theta^{(m)}$ is larger than the mean. It's somewhat another kind of UCB rather than a "true" TS. Do you have any other intuition or motivation to use TS? Can it reduce computational complexity compared with UCB as it still contains a hard-to-compute argmax?

---

> ### Author Response · Authors · 2024-11-18
>
> We sincerely thank you for taking the time to review our paper and for providing thoughtful and valuable feedback. We greatly appreciate your recognition of our work and your constructive comments. Below, we address each of your comments and questions in detail:
>
>
> $\bullet$ **My main concern is the statement that $1/\kappa =O(K^2)$. Is it true?**
>
> **Response:** Thank you for your question. We confirm that it is indeed correct that $1/\kappa = O(K^2)$ in the worst case.
> According to our definition in Line 258, $\kappa=\inf\_{t,i,S,p,\theta\in \Theta}P\_{t,\theta}(i|S,p)P\_{t,\theta}(i\_0,S,p)$ where $\Theta=\\{\theta\in \mathbb{R}^{2d}:||\theta^{1:d}||\_2\le 1, ||\theta^{d+1:2d}||\_2\le 1\\}$ and $P\_{t,\theta}(i|S,p)=\frac{\exp(z\_{i,t}(p\_{i})^\top \theta))}{1+\sum\_{j\in S}\exp(z\_{j,i}(p\_j)^\top \theta)}$ (Line 191). Under Assumption 1, we observe that $|z\_{i,t}(p\_i)^\top \theta|\le C$ for constant $C>0$. This implies that, under the constraint $|S|\le K$, $P_{t,\theta}(i|S,p)\ge \exp(-C)/(1+K\exp(C))=\Omega(1/K)$, resulting in $\kappa = \Omega(1/K^2)$ from the definition, which in turn implies $1/\kappa = O(K^2)$. We will add this discussion in our final.
>
> $\bullet$ **Since different items have different contexts, it's more reasonable if the buyer can buy more than one item at the same time. It'll be an interesting extension to consider this scenario.**
>
> **Response:** Thank you for this suggestion.  In our current work, we use the multinomial logit (MNL) model for user choice, where the buyer purchases at most one item at each time step. This single-purchase constraint aligns with many real-world applications where buyers make selective decisions among a set of products presented at once, such as choosing one flight or booking one hotel room.
>
> A multiple-purchase extension could be approached by adapting the choice model to account for the cumulative utility of a subset of items rather than individual item utility alone. Additionally, this scenario could involve modifying the algorithms and regret analysis to reflect multiple decision points per time step. Extending the model to allow multiple purchases per time step would be an interesting direction for future research.
>
>
> $\bullet$ **I'm wondering if it's possible to get the logarithmic problem-dependent regret as literature in both dynamic pricing and MAB. Maybe you can do some experiments on this. For example, use regression to find the actual order of $T$ (regress log(Regret) on log(T)).**
>
> **Response:** To the best of our knowledge, for contextual bandit problems where the mean reward varies due to dynamic feature information, most existing work emphasizes problem-independent regret, as we do in this paper. This approach is widely used because the changing context leads to arbitrarily varying reward distributions over time, making it challenging to define a stable, problem-dependent regret bound. Given this variability, a worst-case regret bound is a reasonable and robust measure, providing guarantees across diverse and potentially unpredictable scenarios.

---

> ### Author Response · Authors · 2024-11-18
>
> $\bullet$ **For the TS, you use the maximum of m samples. Then, with high probability, $x^\top\tilde{\theta}^{(m)}$
>  is larger than the mean. It's somewhat another kind of UCB rather than a "true" TS. Do you have any other intuition or motivation to use TS? Can it reduce computational complexity compared with UCB as it still contains a hard-to-compute argmax?**
>
> **Response:** The need for multiple $M$ samples in our Thompson Sampling (TS) approach arises from the requirement to select multiple products for the assortment at each time step.  Note that the algorithm is still randomized and selects an assortment based on randomly sampled parameters. While the multiple samples are used to make this set of items a little more optimistic than single sample approach, yet they are far from UCB (that is, an assortment of these items does not form a high-probability optimism as done in UCB) Our TS is still a randomized algorithm, even can be pessimistic at times based on random samples. In contrast, UCB is a deterministic approach where upper confidence bound controls the tradeoff. Thus, there exists the fundamental difference between our TS and UCB algorithms. The nature of our TS algorithm aligns with the linear TS (Abeille \& Lazaric, 2017), and prior work on TS for MNL (Oh \& Iyengar, 2019) similarly uses multiple samples as we do.
>
>
>
> We also highlight that our experimental results demonstrate that TS performs comparably to UCB and sometimes even outperforms UCB when the number of products is sufficiently large (shown in Figure 2 of the revised paper).
>
> Moreover, TS offers computational advantages in assortment selection, especially in high-dimensional settings. Specifically, the computational complexity of the TS-based valuation index for each product,
> $\tilde{v}\_{i,t} = \arg\max\_{m \in [M]} x\_{i,t}^\top \tilde{\theta}\_{v,t}^{(m)}$,
> scales as $O(Md) = \tilde{O}(d)$, where $M = O(\log N)$. In contrast, the complexity of the UCB-based valuation index,
> $\overline{v}\_{i,t} = x_{i,t}^\top \widehat{\theta}\_{v,t} + \beta_\tau ||x\_{i,t}||\_{H\_{v,t}^{-1}}$,
> scales as $O(d^2)$. This difference makes TS computationally more efficient for calculating an index for valuation (and utility)  for assortment selection, particularly when the dimension is large.
>
>
>
> We will include this discussion in the final version of the paper.

---

> > ### Comment · Reviewer_Ues7 · 2024-11-26
> >
> > Thank you for your answer. I don't have more questions.

---

### Official Review · Reviewer_GGxS · 2024-11-02

**Soundness:** 4
**Presentation:** 3
**Contribution:** 3
**Rating:** 8
**Confidence:** 3

**Summary:**

This paper studies the problem of dynamic multi-product selection and pricing. In such a problem, the seller must not only set prices but also determine which products to offer. The authors introduce a LCB-based pricing strategy to set prices, which can promote exploration because buyers would be more likely to purchase, avoiding the censorship issue. As for product assortment selection, they employ two strategies, one based on UCB and the other based on Thompson Sampling. Each strategy corresponds to an algorithm with a guaranteed regret upper bound. Extensive experiments on synthetic datasets validate the performance of the algorithms and demonstrate their superiority over existing approaches.

**Strengths:**

1. The problem studied in this paper introduces product assortment selection in a multi-product sale scenario that has not been considered in previous work.

2. The theoretical results are very solid and accompanied by experimental validation. I really appreciate the proof sketch in the main text, which shows the authors' very clear proof ideas.

**Weaknesses:**

1. According to my understanding, proposing a framework implies proposing a class of solutions. The authors' real contribution is to find a problem - a problem that fits the actual application scenario - and formalize it, so it is inappropriate to describe it as "proposing a novel framework".

2. I know it is difficult to find and prove a regret lower bound in such a complicated problem that involves both price setting and product selection, but a complete paper should at least have a discussion about the regret lower bound. Perhaps it is to refer to the magnitude of the upper and lower bounds of other similar problems, or perhaps to propose a conjecture of a tight regret lower bound, so that readers can have a clearer understanding of the difficulty of this problem and the contribution of the authors.

3. No experiment on any real-world datsaet.

**Questions:**

This paper proposes two product selection strategies, which makes the theoretical results of this paper look more fruitful. Give the regret upper bound of the TS strategy is worse than that of the UCB strategy, is there any more benefit to studying this strategy besides being able to announce that "it is the first work to apply Thompson Sampling"? For example, is the TS strategy more convenient in code implementation? Or is there an advantage in computational complexity? ...

Please compare the two strategies in detail. The rating may be reduced if the answer is insufficient.

---

> ### Author Response · Authors · 2024-11-18
>
> We sincerely thank you for taking the time to review our paper and for providing thoughtful and valuable feedback. We greatly appreciate your recognition of our work and your constructive comments. Below, we address each of your comments and questions in detail:
>
>
> $\bullet$ **According to my understanding, proposing a framework implies proposing a class of solutions. The authors' real contribution is to find a problem - a problem that fits the actual application scenario - and formalize it, so it is inappropriate to describe it as "proposing a novel framework"**
>
> **Response:** Thank you for your feedback.  We understand the distinction you are making between proposing a framework and formalizing a new problem.  Our intention was to emphasize the structured approach we introduced to address the complexities of dynamic multi-product selection and pricing under preference feedback with censorship  (including hidden censorship feedback)—a scenario that closely aligns with real-world applications.
>
>
> By using the term "novel framework," we aimed to convey the structured model we developed to systematically capture these intricacies, including buyer censorship and dynamic learning of valuations and preferences.
>
> We will adjust the terminology to better reflect this in the revised manuscript.
>
>
> $\bullet$ **I know it is difficult to find and prove a regret lower bound in such a complicated problem that involves both price setting and product selection, but a complete paper should at least have a discussion about the regret lower bound. Perhaps it is to refer to the magnitude of the upper and lower bounds of other similar problems, or perhaps to propose a conjecture of a tight regret lower bound, so that readers can have a clearer understanding of the difficulty of this problem and the contribution of the authors.**
>
> **Response:** Thank you for this valuable suggestion. We agree that a discussion of regret lower bounds would offer further insight into the complexity of our problem, which involves both price setting and product selection.
>
> We note that previous UCB algorithms for the MNL bandit (without pricing aspect) achieve a regret bound of $d\sqrt{T}$.
> However, our problem setting extends the MNL bandits to a censored version of MNL integrated with pricing, which introduces additional complexity due to the activation function and hidden censorship feedback.
> Despite this increased challenge, our approach achieves a regret bound of $d^{3/2}\sqrt{T}$. While we conjecture that the regret lower bound for our problem likely lies between $d\sqrt{T}$ and $d^{3/2}\sqrt{T}$ inclusive, formally establishing this remains an open question.
>
> In our revision, we will include this discussion. We believe this addition will provide readers with a clearer understanding of the difficulty of our problem and the significance of our contributions in achieving this upper bound.
>
>
> $\bullet$ **No experiment on any real-world dataset.**
>
> **Response:** Conducting experiments in bandit settings using offline datasets is inherently challenging due to the nature of partial feedback in online learning. Bandit algorithms rely on feedback that is contingent upon the actions they take, whereas offline datasets typically lack counterfactual feedback—i.e., the outcomes of actions not taken during data collection. This fundamental limitation makes it difficult to directly evaluate bandit algorithms on fixed real-world datasets without significant adjustments.
>
> In most cases, offline datasets must be transformed into semi-synthetic datasets, where missing counterfactual feedback is either imputed or simulated based on assumptions. This transformation effectively reduces real-world data to synthetic, potentially limiting its practical interpretability.
>
> For these reasons, we adopted the approach commonly used in the bandit literature, focusing on theoretical results validated with synthetic datasets. This methodology allows for controlled experimentation and ensures that the theoretical properties of the algorithm are demonstrated under well-defined conditions. However, if the reviewer requests, we are open to performing additional experiments using a semi-synthetic approach with a real-world dataset.

---

> ### Author Response · Authors · 2024-11-18
>
> $\bullet$ **This paper proposes two product selection strategies, which makes the theoretical results of this paper look more fruitful. Give the regret upper bound of the TS strategy is worse than that of the UCB strategy, is there any more benefit to studying this strategy besides being able to announce that "it is the first work to apply Thompson Sampling"? For example, is the TS strategy more convenient in code implementation? Or is there an advantage in computational complexity? ...**
>
>
> **Response:**
> We thank the reviewer for their comment and would like to clarify the motivation for including the Thompson Sampling (TS) algorithm in our work. While TS exhibits slightly weaker theoretical regret bounds compared to the UCB-based approach (with an additional $\sqrt{d}$ term, which is consistent with the existing results in other TS algorithms (e.g., Oh \& Iyengar, 2019; Agrawal \& Goyal, 2013; Abeille \& Lazaric, 2017)), our experimental results demonstrate that TS performs comparably to UCB and sometimes even outperforms UCB when the number of products is sufficiently large (shown in Figure 2 of the revised paper).
>
>
> Moreover, it offers computational advantages in assortment selection, especially in high-dimensional settings. Specifically, the computational complexity of the TS-based valuation index for each product,
> $\tilde{v}\_{i,t} = \arg\max\_{m \in [M]} x\_{i,t}^\top \tilde{\theta}\_{v,t}^{(m)}$,
> scales as $O(Md) = \tilde{O}(d)$, where $M = O(\log N)$. In contrast, the complexity of the UCB-based valuation index,
> $\overline{v}\_{i,t} = x_{i,t}^\top \widehat{\theta}\_{v,t} + \beta_\tau ||x\_{i,t}||\_{H\_{v,t}^{-1}}$,
> scales as $O(d^2)$. This difference makes TS computationally more efficient for calculating an index for valuation for assortment selection, particularly when the dimension is large. This computational advantage also applies to calculating utility indices.
>
>
> Both UCB and TS are widely recognized as foundational approaches in bandit literature due to their theoretical and empirical efficiency. UCB represents a deterministic approach, while TS leverages randomized exploration. In our novel problem setting, we have adapted both frameworks to incorporate an integrated pricing strategy and provided rigorous regret analyses for each. Analyzing the regret of TS, in particular, is more challenging than UCB because it requires accounting for estimation errors in the sampled values and establishing optimism properties under the algorithm’s inherent randomness. For these reasons, achieving the regret bound for TS in this context, even after UCB, is widely considered an established contribution in bandit literature (Agrawal \& Goyal, 2013; Abeille \& Lazaric, 2017).
> This contribution is particularly significant in our problem setting because extending the analysis from UCB to TS is far less straightforward in our framework. The additional complexities introduced by the integration of pricing strategies and the randomness of TS make this a non-trivial extension, further highlighting the value of our contribution.
>
> Furthermore, as noted in Lines 467–475, this work is the first to apply TS to dynamic pricing under the MNL model, achieving a regret bound without $\kappa$ dependency in the main term and introducing computationally efficient online updates for the estimator. This contribution represents a meaningful step forward for the community, as TS-based algorithms have not been explored in this context before.
>
>
> For these reasons, we assert that TS is an important contribution to this work. We will include a discussion of these points in the final version of the paper to clarify the motivation and significance of including TS.

---

### Official Review · Reviewer_8pM9 · 2024-11-03

**Soundness:** 4
**Presentation:** 3
**Contribution:** 3
**Rating:** 6
**Confidence:** 4

**Summary:**

This paper studies the problem of online pricing and assortment optimization, where a seller engages in repeated interactions with a buyer over $ T $ discrete time steps. In this setting, the seller offers a selection of products from a set of size $ N $. At each time step $ t $, the seller chooses an assortment $ S_t \subseteq [N] $ of products to present to the buyer, along with a specific price $ p_{i,t} $ for each item $ i \in S_t $.

The buyer is characterized by latent parameters $ \theta_v $ and $ \theta_{\alpha} \in \mathbb{R}^d $, which influence their purchasing decisions but remain unknown the seller. Each product $ i \in S_t $ is associated with known feature vectors $ x_{i,t} $ and $ w_{i,t} \in \mathbb{R}^d $, representing various characteristics of the items. The buyer's valuation of a product is given by $v_{i,t} =  x_{i,t}^T \theta_v $, while their price sensitivity is described by the parameter $ \alpha_{i,t} = w_{i,t}^T \theta_{\alpha} $. Consequently, the buyer's utility for a product $ i $ is defined as $ v_{i,t} - \alpha_{i,t} p_{i,t} $. The buyer makes a purchasing decision based on a "censored multinomial logit choice function," which acts as a sigmoid function applied to all items $ i \in S_t $ for which the price $ p_{i,t} $ does not exceed the buyer’s value $ v_{i,t} $.

The goal is to minimize regret compared to the optimal assortment and pricing decisions in hindsight. A key challenge is that the seller lacks information on which products are "censored" by the buyer's multinomial logit choice function (i.e., $v_{i,t} < p_{i,t}$). This uncertainty makes it challenging for the seller to glean information from the buyer's purchasing behavior. To address this, the authors employ a combination of algorithms: a Lower Confidence Bound (LCB) approach for setting prices and an Upper Confidence Bound (UCB) method for selecting assortments. By using the LCB algorithm, prices are initially set low, reducing the likelihood that items will be censored in the early stages. This enables the algorithm to gather more accurate data on the buyer's latent parameters, $ \theta_v $ and $ \theta_{\alpha} $. This algorithm obtains a regret bound of $ O(d^{2/3}T^{1/2}) $.

**Strengths:**

-	The paper offers a natural extension of the multinomial logit (MNL) model, building on previous research that has explored online pricing and assortment optimization for the MNL model without incorporating censorship. Censorship seems natural and adds interesting additional, combinatorial challenges.
-	Additionally, the proposed approach, which combines the LCB and UCB algorithms, feels intuitive and well-aligned with the problem.
-	Finally, the paper well written.

**Weaknesses:**

-	Including the Thompson Sampling (TS) section should be more thoroughly motivated, especially since the algorithm’s regret is weaker than that of the LCB/UCB approach. From the paper, it’s unclear what advantages TS offers over the LCB/UCB algorithm. If none, this section might not be necessary and could be removed to streamline the paper.
-	Additionally, I would appreciate a more detailed explanation of why the latent price sensitivity parameter $ \alpha_{i,t} $ is essential to the model. It would be helpful to have some real-world examples that illustrate the importance of including $ \alpha_{i,t} $. Moreover, I find it counterintuitive that $ \alpha_{i,t} $ appears in the exponent of the MNL choice function $ \exp(v_{i,t} - \alpha_{i,t} p_{i,t}) $, yet it does not appear in the indicator $ \mathbb{1}(p_{i,t} \leq v_{i,t}) $. Intuitively, it might make more sense if the indicator were formulated as $ \mathbb{1}(\alpha_{i,t} p_{i,t} \leq v_{i,t}) $, as this would more consistently capture the influence of $ \alpha_{i,t} $ on the decision threshold.

**Questions:**

Please address the confusions I highlighted in the Weaknesses section.

---

> ### Author Response · Authors · 2024-11-18
>
> We sincerely thank you for taking the time to review our paper and for providing thoughtful and valuable feedback. We greatly appreciate your recognition of our work and your constructive comments. Below, we address each of your comments and questions in detail:
>
> $\bullet$ **Including the Thompson Sampling (TS) section should be more thoroughly motivated, especially since the algorithm’s regret is weaker than that of the LCB/UCB approach. From the paper, it’s unclear what advantages TS offers over the LCB/UCB algorithm. If none, this section might not be necessary and could be removed to streamline the paper.**
>
> **Response:**  We thank the reviewer for their comment and would like to clarify the motivation for including the Thompson Sampling (TS) algorithm in our work. While TS exhibits slightly weaker theoretical regret bounds compared to the UCB-based approach (with an additional $\sqrt{d}$ term, which is consistent with the existing results in other TS algorithms  (e.g., Oh \& Iyengar, 2019; Agrawal \& Goyal, 2013; Abeille \& Lazaric, 2017)), our experimental results demonstrate that TS performs comparably to UCB and sometimes even outperforms UCB when the number of products is sufficiently large (shown in Figure 2 of the revised paper).
>
> Moreover, TS offers computational advantages in assortment selection, especially in high-dimensional settings. Specifically, the computational complexity of the TS-based valuation index for each product,
> $\tilde{v}\_{i,t} = \arg\max\_{m \in [M]} x\_{i,t}^\top \tilde{\theta}\_{v,t}^{(m)}$,
> scales as $O(Md) = \tilde{O}(d)$, where $M = O(\log N)$. In contrast, the complexity of the UCB-based valuation index,
> $\overline{v}\_{i,t} = x_{i,t}^\top \widehat{\theta}\_{v,t} + \beta_\tau ||x\_{i,t}||\_{H\_{v,t}^{-1}}$,
> scales as $O(d^2)$. This difference makes TS computationally more efficient for calculating an index for valuation for assortment selection, particularly when the dimension is large. This computational advantage also applies to calculating utility indices.
>
>
>
>
> Both UCB and TS are widely recognized as foundational approaches in bandit literature due to their theoretical and empirical efficiency. UCB represents a deterministic approach, while TS leverages randomized exploration. In our novel problem setting, we have adapted both frameworks to incorporate an integrated pricing strategy and provided rigorous regret analyses for each. Analyzing the regret of TS, in particular, is more challenging than UCB because it requires accounting for estimation errors in the sampled values and establishing optimism properties under the algorithm’s inherent randomness. For these reasons, achieving the regret bound for TS in this context, even after UCB, is widely considered an established contribution in bandit literature (Agrawal \& Goyal, 2013; Abeille \& Lazaric, 2017).
> This contribution is particularly significant in our problem setting because extending the analysis from UCB to TS is far less straightforward in our framework. The additional complexities introduced by the integration of pricing strategies and the randomness of TS make this a non-trivial extension, further highlighting the value of our contribution.
>
> Furthermore, as noted in Lines 467–475, this work is the first to apply TS to dynamic pricing under the MNL model, achieving a regret bound without $\kappa$ dependency in the main term and introducing computationally efficient online updates for the estimator. This contribution represents a meaningful step forward for the community, as TS-based algorithms have not been explored in this context before.
>
> For these reasons, we assert that TS is an important contribution to this work. We will include a discussion of these points in the final version of the paper to clarify the motivation and significance of including TS.

---

> > ### Author Response · Authors · 2024-11-18
> >
> > $\bullet$ **Additionally, I would appreciate a more detailed explanation of why the latent price sensitivity parameter $\alpha_{i,t}$  is essential to the model. It would be helpful to have some real-world examples that illustrate the importance of including $\alpha_{i,t}$. Moreover, I find it counterintuitive that $\alpha_{i,t}$ appears in the exponent of the MNL choice function $\exp(v_{i,t}-\alpha_{i,t}p_{i,t})$, yet it does not appear in the indicator $1(p_{i,t}\le v_{i,t})$. Intuitively, it might make more sense if the indicator were formulated as $1(\alpha_{i,t}p_{i,t}\le v_{i,t})$, as this would more consistently capture the influence of $\alpha_{i,t}$ on the decision threshold.**
> >
> > **Response:**  Thank you for this thoughtful question. To clarify, we restate our C-MNL model as follows:
> >
> > $$
> > \mathbb{P}\_{t}(i | S_t, p_t) := \frac{\exp(v\_{i,t} - \alpha\_{i,t} p\_{i,t}) \cdot 1(p\_{i,t} \le v\_{i,t})}{1 + \sum\_{j \in S\_t} \exp(v\_{j,t} - \alpha\_{j,t} p\_{j,t}) \cdot 1(p\_{j,t} \le v\_{j,t})}.
> > $$
> >
> > In our model, the latent price sensitivity parameter $\alpha_{i,t}(\ge 0)$ is essential because it reflects how sensitive a buyer’s preference is to changes in price. When prices increase, $\alpha_{i,t}$ plays a role in reducing the perceived utility (preference value) of a product, following the expression $v_{i,t} - \alpha_{i,t} p_{i,t}$. This construction allows us to model refined buyer behavior, as buyers might respond differently to the same price changes depending on the products, which is important for real-world applications such as e-commerce. In online retail, different types of products elicit different levels of price sensitivity. For instance, a luxury item such as a high-end smartphone might have a low $\alpha_{i,t}$, indicating that buyers with a high preference are less deterred by price increases. In contrast, a basic household item like a coffee maker might have a higher $\alpha_{i,t}$, meaning buyers are more sensitive to price changes, and an increase could substantially reduce the likelihood of purchase. This price sensitivity parameter helps capture such differences in behavior across product categories.
> >
> >
> > The indicator function $1(p_{i,t} \le v_{i,t})$ is included to reflect the threshold of consideration: if the product’s price $p_{i,t}$ exceeds the buyer’s valuation $v_{i,t}$, they are unlikely to even consider it. The term $\alpha_{i,t}$ is not included in this threshold because $\alpha_{i,t}$ represents sensitivity in *preference utility* rather than the buyer’s absolute willingness to consider a product at a certain price. Including $\alpha_{i,t}$ in the exponent captures how sensitive buyers are in their *degree of preference* for the product when it is within a feasible price range, while the threshold $1(p_{i,t} \le v_{i,t})$ simply determines if the product is even a consideration. We believe this formulation better aligns with the intuitive buyer behavior, where the decision threshold is based on valuation and price, and the final preference is modulated by price sensitivity.
> >
> >
> > We hope this clarifies the role of $\alpha_{i,t}$, and we appreciate the chance to provide additional context. We will add these explanations and examples to the final version to aid reader understanding.

---

### Official Review · Reviewer_3kB4 · 2024-11-04

**Soundness:** 3
**Presentation:** 3
**Contribution:** 2
**Rating:** 6
**Confidence:** 3

**Summary:**

The authors study the dynamic multi-product selection problem. They introduce a new censored multinomial logit (C-MNL) choice model, capturing buyer behavior by filtering products based on price thresholds. They propose two algorithms that leverage a Lower Confidence Bound (LCB) pricing strategy, combined with either an Upper Confidence Bound (UCB) or Thompson Sampling (TS) approach for selecting product assortments. Both algorithms achieve provable regret upper bounds, and their performance is further validated through simulations.

**Strengths:**

1) The censored multinomial logit (C-MNL) model is a novel approach that effectively captures buyer behavior by incorporating a price threshold for product selection, reflecting realistic purchasing patterns.
2) The authors introduce an innovative Lower Confidence Bound (LCB)-based valuation strategy for pricing, which can be flexibly integrated with various product selection methods to support this new model.
3) The authors provide comprehensive theoretical analyses, deriving regret upper bounds for both proposed algorithms. Moreover, these algorithms are computationally more efficient than previous methods

**Weaknesses:**

1) The algorithms involve several input parameters, such as $\lambda$, $\eta$, and $C$. However, the paper lacks guidance on how to select appropriate values for these parameters and does not discuss how they impact the regret bounds.
2) There is no lower bound analysis on the dependency on $d$, making it uncertain whether the proposed algorithms are optimal.
3) In the TS-based algorithm, the prior distribution and prior knowledge used (e.g., Gaussian distributions) are not clearly discussed, leaving the assumptions about the prior unclear.
4) The experimental setup is relatively limited, with tests conducted only for $K=4$, $d=2$, and approximately 20 arms (products), which does not reflect the scale and complexity of practical applications.

**Questions:**

1) Line 174: "Then we define an oracle policy." Could you clarify if approximate regret is considered in cases where the oracle is based on an approximation algorithm?
2) Line 347: "Additionally, our regret bound does not contain $1/ \kappa$ in the leading term". Could you provide some intuition or high-level explanation as to why the term $1/ \kappa$ is absent in the leading term of our regret bound? This could help readers better understand the underlying reasons for this distinction.
3) It could be interesting to explore whether the LCB-based valuation strategy for pricing can be effectively integrated with product selection methods other than UCB and TS.
4) Section 6: Including examples of real-world applications would enhance the discussion on potential extensions.

---

> ### Author Response · Authors · 2024-11-18
>
> We sincerely thank you for taking the time to review our paper and for providing thoughtful and valuable feedback. We greatly appreciate your recognition of our work and your constructive comments. Below, we address each of your comments and questions in detail:
>
> $\bullet$ **The algorithms involve several input parameters, such as $\lambda$, $\eta$, $C$. However, the paper lacks guidance on how to select appropriate values for these parameters and does not discuss how they impact the regret bounds.**
>
> **Response:**
> We provide specific settings for $\eta$ and $\lambda$ in Lines 247 and 424, where we set $\eta = (1/2) \log(K+1) + 3$ and $\lambda = \max\\{84d\eta, 192\sqrt{2}\eta\\}$, which are derived from our regret analysis to ensure effective learning performance. For $C$, we consider any constant $C > 1$, as stated on Line 229. Importantly, $C$ influences the regret bound only by a constant factor of $\sqrt{C}$, which does not hurt the regret in order.
>
> To further clarify the role of each parameter. The parameter of $\eta$  serves as a step size in the online mirror descent updates of $\hat{\theta}_t$ (Line 5 in Algorithm 1), helping to control the convergence behavior in each iteration. The parameter of  $\lambda$ is used as a regularization parameter in constructing the Gram matrices $H\_t$ and $H\_{v,t}$ (Lines 2-4 in Algorithm 1). $\lambda$ mitigates variance in the estimation of $\hat{\theta}\_t$, especially under high-dimensional settings, ensuring stability in the updates. The parameter of $C$ impacts the frequency of estimator updates, specifically for $\hat{\theta}\_{v,(\tau)}$ in the LCB pricing strategy (Line 6 in Algorithm 1).
>
> We appreciate this opportunity to elaborate and will incorporate the explanation regarding the parameters in our final version.
>
>
> $\bullet$ **There is no lower bound analysis on the dependency $d$, making it uncertain whether the proposed algorithms are optimal.**
>
> **Response:**
> Thank you for your feedback. We agree that a discussion of regret lower bounds would offer further insight into the complexity of our problem, which involves both price setting and product selection.
>
> We note that previous UCB algorithms for the MNL bandit (without pricing aspect) achieve a regret bound of $d\sqrt{T}$.
> However, our problem setting extends the MNL bandits to a censored version of MNL integrated with pricing, which introduces additional complexity due to the activation function and hidden censorship feedback.
> Despite this increased challenge, our approach achieves a regret bound of $d^{3/2}\sqrt{T}$. While we conjecture that the regret lower bound for our problem likely lies between $d\sqrt{T}$ and $d^{3/2}\sqrt{T}$ inclusive, formally establishing this remains an open question.
> In our revision, we will include this discussion. We believe this addition will provide readers with a clearer understanding of the difficulty of our problem and the significance of our contributions in achieving this upper bound.
>
>
> $\bullet$ **In the TS-based algorithm, the prior distribution and prior knowledge used (e.g., Gaussian distributions) are not clearly discussed, leaving the assumptions about the prior unclear.**
>
> **Response:**
> To clarify, our approach does not rely on any explicit prior distribution or prior knowledge, consistent with the methodology outlined by Abeille and Lazaric (2016). As they note, “Thompson Sampling can be defined as a generic randomized algorithm constructed on the Regularized Least Squares estimate rather than as an algorithm sampling from a Bayesian posterior.” This principle underpins our implementation of the TS algorithm.
>
> Additionally, our regret analysis adopts a frequentist perspective, ensuring robustness in the worst-case setting without dependence on any assumed prior distribution. This frequentist approach provides regret bounds that are independent of specific priors, making them applicable across a broad range of scenarios.
>
> We will incorporate this discussion in the final version of our paper to prevent any potential misunderstandings about prior knowledge or distributional assumptions.
>
>
>
>
> $\bullet$  **The experimental setup is relatively limited, with tests conducted only for $K=4$, $d=2$, and approximately 20 arms (products), which does not reflect the scale and complexity of practical applications.**
>
> **Response:**
> To address this concern, we have expanded the experiments to include larger and more realistic scenarios that better reflect the scale and complexity of practical applications. Specifically, we have conducted additional experiments with the number of products $N=40, 60, 80$ with an increased assortment size $K=5$, and a higher feature dimensionality  $d=4$. The results of these experiments are included in Figure 2 in the experiment section of the revised version.

---

> > ### Author Response · Authors · 2024-11-18
> >
> > $\bullet$
> > **Line 174: "Then we define an oracle policy." Could you clarify if approximate regret is considered in cases where the oracle is based on an approximation algorithm?**
> >
> > **Response:**  We confirm that our framework considers *exact regret* without use of approximation, as the oracle is defined to maximize the true revenue $R_t(S, p)$ under the assumption of prior knowledge of $\theta^*$.  This definition ensures that the regret directly measures the performance gap between our proposed algorithms and the optimal policy.
> >
> > Furthermore, as mentioned in Line 370, the assortment optimization step can be solved by formulating it as a linear program (LP), as outlined in Davis et al. (2013). The LP formulation enables us to compute the exact optimal assortment. Consequently, there is no reliance on approximation algorithms.
> >
> >
> > $\bullet$ **Line 347: "Additionally, our regret bound does not contain $1/\kappa$ in the leading term". Could you provide some intuition or high-level explanation as to why the term $1/\kappa$ is absent in the leading term of our regret bound? This could help readers better understand the underlying reasons for this distinction.**
> >
> > **Response:**
> > The absence of the $1/\kappa$ term in the leading regret bound arises from our use of an adaptive Gram matrix (Eq. (3)) that incorporates the dynamic probability $ P_{t,\theta}(i | S_t, p_t) $ at each time step $ t $, rather than conservatively relying on the worst-case scenario of $ 1/\kappa $. This approach allows us to construct a tighter confidence bound, specifically $ ||\widehat{\theta}\_t - \theta^*||\_{H_t} \le \beta\_{\tau\_t} $, which effectively excludes the dependence on $ \kappa $ in $ \beta\_{\tau\_t} $.
> >
> > The intuition here is that by adapting the Gram matrix based on observed data and probabilities, we avoid the need to account for the worst-case $ 1/\kappa $ factor, which represents a highly conservative assumption that is not reflective of typical cases encountered in practice. Instead, our adaptive approach dynamically adjusts to the actual observed probabilities, allowing us to achieve a regret bound that scales more favorably without introducing unnecessary dependence on $ \kappa $. Additionally, our use of the online mirror descent method in updating $ \widehat{\theta}_t $ contributes to the computational efficiency of the algorithm. We will include the explanation in the final version.
> >
> >
> >
> >  $\bullet$ **It could be interesting to explore whether the LCB-based valuation strategy for pricing can be effectively integrated with product selection methods other than UCB and TS.**
> >
> >  **Response:** Thank you for your suggestion. Our LCB pricing strategy serves to encourage exploration by setting lower prices, which helps mitigate buyer censorship, allowing us to gather valuable information about buyer preferences and product valuations. This pricing approach effectively complements the UCB and TS methods for product selection by creating a synergy with exploration in product selection.
> >
> > In this work, we utilize UCB and TS-based approaches for product selection, achieving tight regret bounds. These methods are particularly suitable for our framework, as they strike a balance between exploration and exploitation, especially in scenarios where product selection is intertwined with dynamic pricing. However, exploring the integration of LCB-based pricing with alternative product selection strategies is an intriguing direction for future research.

---

> > > ### Author Response · Authors · 2024-11-18
> > >
> > > $\bullet$ **Section 6: Including examples of real-world applications would enhance the discussion on potential extensions.**
> > >
> > >
> > >  **Response:** Thank you for this suggestion. As noted on Lines 35–36, our model, which involves recommending multiple items at set prices from which the buyer selects one based on preference, has broad applicability across several real-world domains.
> > >
> > > (E-commerce) In online retail, platforms commonly present multiple products within a category, often with dynamic pricing that adjusts based on demand patterns, seasonal trends, and buyer behavior. Our model can be extended to improve recommendation and pricing algorithms by dynamically adjusting both assortment and pricing, learning buyer preferences through purchase data.
> > >
> > > (Hotel Reservations) Online travel agencies and hotel booking platforms typically display a selection of rooms, varying in price and amenities, to meet diverse customer preferences. In this context, our approach could help platforms determine optimal room assortments and price points to maximize revenue by dynamically adjusting for customer preferences over time.
> > >
> > > (Air Travel)  Aggregator platforms for air travel present travelers with a range of flight options varying in price, departure times, airlines, and seating classes. Using our framework, the platforms could optimize the assortment of flights displayed to each user by dynamically adjusting which flights and price points are shown based on observed user preferences. By learning from purchase behavior, the platforms could prioritize flights likely to match user needs while adjusting prices strategically to maximize platform revenue.
> > >
> > > We will incorporate this broader discussion into the final version.

---

### Meta-Review · Area_Chair_3ZDN · 2024-12-21

**Metareview:**

The paper addresses the dynamic multi-product selection problem and introduces the censored multinomial logit (C-MNL) model, which extends the traditional MNL model by incorporating price-based product censorship to better capture buyer behavior. The authors propose two algorithms that integrate a Lower Confidence Bound (LCB) pricing strategy with either Upper Confidence Bound (UCB) or Thompson Sampling (TS) approaches for assortment optimization. Both algorithms achieve theoretical regret bounds and are validated through simulations. The paper's strengths include a natural extension of the MNL model to address combinatorial challenges introduced by censorship, an intuitive algorithmic approach combining LCB and UCB, and clear and effective writing.

Considering the above novelties, I recommend acceptance of the paper once the authors incorporate all the reviewer's feedback in the final version.

**Additional Comments On Reviewer Discussion:**

See above.

---

### Decision · Program_Chairs · 2025-01-22

Accept (Poster)